# SAMPLING FROM MULTIMODAL DISTRIBUTIONS WITH WARM STARTS

## ABSTRACT

Sampling from multimodal distributions is a central challenge in Bayesian inference and machine learning. In light of hardness results for sampling—classical MCMC methods, even with tempering, can suffer from exponential mixing times—a natural question is how to leverage additional information, such as a warm start point for each mode, to enable faster mixing across modes. For this problem, we prove the first polynomial-time bound that works in a general setting, under a natural assumption that each component contains significant mass relative to the others when tilted towards the corresponding warm start point. For this, we introduce a modified version of the Annealed Leap-Point Sampler (ALPS) (Tawn et al., 2021; Roberts et al., 2022). Similarly to ALPS, we define distributions tilted towards a mixture centered at the warm start points, and at the coldest level, use teleportation between warm start points to enable efficient mixing across modes. In contrast to ALPS, our method does not require Hessian information at the modes, but instead estimates component partition functions via Monte Carlo. This additional estimation step is critical in allowing the algorithm to handle target distributions with more complex geometries besides approximate Gaussian. For the proof, we show convergence results for Markov processes when only part of the stationary distribution is well-mixing and estimation for partition functions for individual components of a mixture. We numerically evaluate our algorithm's mixing performance on a mixture of heavy-tailed distributions, comparing it against the ALPS algorithm on the same distribution.

## 1 INTRODUCTION

A key task in statistics and machine learning is sampling from a probability distribution known up to normalization, $\pi(x) \propto e^{-V(x)}$. The standard approach of Markov Chain Monte Carlo (MCMC) is to define a Markov chain with stationary distribution $\pi(x)$. The time it takes for MCMC methods to produce an approximate sample from $\pi(x)$ depends on the mixing time of the underlying Markov chain. Unfortunately, in many applications, the target distribution $\pi(x)$ is multimodal, which causes Markov chains with local moves to mix slowly, as transitions between different modes rarely occur; this is the general phenomenon of metastability (Bovier et al., 2002).

Modern MCMC methods such as simulated tempering (Marinari & Parisi, 1992), parallel tempering (also known as replica exchange) (Swendsen & Wang, 1986), Sequential Monte Carlo (also known as particle filtering) (Del Moral et al., 2006), and annealed importance sampling (Neal, 2001) attempt to speed up sampling by running a Markov chain with a sequence of interpolating distributions $p_\beta(x) \propto \pi(x)^\beta$ or $p_\beta(x) \propto \pi(x)^\beta p_0(x)^{1-\beta}$ at varying inverse temperatures $\beta$. The idea is that at high temperatures the Markov chain can more easily mix between modes of the target distribution.

These methods offer a powerful framework for sampling from multimodal distributions, and recent works have obtained non-asymptotic mixing time bounds for multimodal stationary distributions with polynomial dependence on parameters, but only under restrictive assumptions. Specifically, tempering methods are prone to bottlenecks, where the relative weight of a mode collapses at an intermediate temperature (Woodard et al., 2009b). This results in the sampler getting trapped in specific parts of the chain, and in fact can require any algorithm to make exponentially many queries (Ge et al., 2018a). Such issues motivate a search for algorithms which leverage more information, such as approximation of local modes, which we term *warm starts*.

A common strategy for utilizing warm starts is to employ mode jump samplers such as (Tjelmeland & Hegstad, 2001; Ibrahim, 2009; Lindsey et al., 2022). These algorithms address poor mixing due to multimodality by allowing samples to jump (teleport) between modes of the target distribution. However, in high dimensions the Markov process can have very low acceptance rates when jumping between modes, because arbitrary distributions will in general have exponentially small overlap even when superimposed.

One of the most effective approaches in the warm start setting is the Annealed Leap-Point Sampler (ALPS) (Tawn et al., 2021). The ALPS algorithm utilizes warm starts by combining mode-jumping with annealing to colder temperatures. At the coldest temperature, the distribution is peaked around the warm start points, and samples can leap from mode to mode of the peaked distributions with high acceptance probability. Note that annealing to cold temperatures is exactly the opposite of how tempering methods typically function. In order to prevent bottlenecks from forming, ALPS uses Hessian Adjusted Tempering (HAT) (Tawn et al., 2020b) to acquire an analytical estimate of the modal weights under the assumption that the modes are approximately Gaussian. While the ALPS algorithm benefits from fast run times and good mixing in this setting, it relies on local Gaussian approximations to overcome bottlenecks.

In this paper, we introduce Reweighted ALPS (Re-ALPS), a variant which dispenses with the Gaussian approximation assumption and addresses the challenge of vanishing modes by introducing a dynamic reweighting scheme. Our key contribution is to iteratively estimate the modal component weights at each temperature, which ensures relative weight balance throughout the temperature ladder. We are able to gain control of the modal weights by our choice of tempering scheme. Instead of using weight-preserving power tempering (Tawn et al., 2020b), we choose the intermediate distributions to be $\pi_\beta(x) \propto \pi(x) \cdot \sum_{k=1}^{M} w_{\beta,k} q_\beta(x - x_k)$ with $w_{\beta,k}$ dynamically chosen by the algorithm. We prove non-asymptotic bounds in total variation (TV) for our modified algorithm that have polynomial dependence on parameters.

Importantly, our results are free of functional inequalities that depend on the global geometry of the target density. Instead, we prove upper bounds on mixing time for the underlying Markov process in the algorithm in terms of local Poincaré constants (capturing local mixing) alone. Our analysis proceeds through a Markov chain decomposition theorem (Madras & Randall, 2002; Ge et al., 2018a), which requires us to bound the Poincaré constant of a certain projected chain (capturing mixing between components). This Poincaré constant is bounded by appropriate algorithmic choice of level and component weights $r_i, w_{i,k}$ and temperature ladder $\beta_i$.

We overcome two new technical challenges in the analysis. First, the tempering scheme can create bad components, so we develop new theoretical analyses for Markov chains that show mixing in the "good" part of the stationary distribution. Second, in addition to estimating the partition function of the tempered distributions $\pi_{\beta_i}$ for each level $i$ to balance the levels (via $r_i$), we also need to estimate the partition functions for the components $\pi_{\beta_i,k}$ of the mixture, in order to balance the modes (via $w_{i,k}$) and avoid a bottleneck in the projected chain. We show that the partition functions can be approximated using Monte Carlo; the proof requires a technically involved analysis due to possible interference between different components.

## 1.1 Sampling with different kinds of advice

We are interested in the problem of approximately sampling from a density $\pi(x) \propto e^{-V(x)}$ that is multimodal. A common way of formalizing the multimodality (Ge et al., 2018a; Koehler & Vuong, 2023)—strictly for the purpose of theoretical analysis—is to assume that $\pi = \sum_{i=1}^{m} w_i \pi_i$, where each component $\pi_i$ satisfies a functional inequality; that is, the natural Markov chain on the space mixes rapidly. Crucially, algorithms are required to operate in a black-box setting, without knowledge of the weights $w_i$, the component densities $\pi_i$, or the functional form of the decomposition. As heuristic support for the decomposition hypothesis, we note it is polynomially equivalent to having at most $k$ low-lying eigenvalues for discrete state space (Lee et al., 2014) or for Langevin (Miclo, 2015), and the latter holds for a distribution with $k$ modes in the limit of low temperature under mild regularity assumptions (Kolokoltsov, 2007, Chapter 8, Propositions 2.1–2.2).

We classify approaches to this problem depending on the strength of extra information, or *advice* that we are given. Here, we focus on approaches with theoretical guarantees.

**No advice.** Without extra information, guarantees are available only under strong conditions. Early work gives guarantees for simulated and parallel tempering assuming suitable decompositions (Madras & Randall, 2002; Woodard et al., 2009a). (Ge et al., 2018a) show that simulated tempering combined with Langevin dynamics works for a mixture of translates of distributions satisfying a log-Sobolev inequality (e.g. a mixture of Gaussians with equal covariance); this is generalized to other Markov processes by (Garg et al., 2025). For sequential Monte Carlo, (Paulin et al., 2018; Mathews & Schmidler, 2024) show guarantees for multimodal distributions but require separation between modes. (Lee & Santana-Gijzen, 2024) allow a general mixture but assume component weights do not change between temperatures, while (Han et al., 2025) consider general distributions with two-welled potentials.

An inherent challenge that leads to restrictive assumptions in the above results is the following: in general, a component can have smaller weights at higher temperatures, creating a "bottleneck" that prevents samples from moving into that mode. In simple terms, it is generally difficult to find a mode. A simple example is that of two Gaussians with different covariances. (Woodard et al., 2009b) observe exponential lower bounds for simulated and parallel tempering in this setting. More generally, considering a family of perturbations of such distributions, no algorithm can generate a sample within constant TV distance with sub-exponentially many queries to $\pi$ or $\nabla \ln \pi$ (Ge et al., 2018a). Reweighting is a possible solution (Tawn et al., 2020b) but relies on components being located and approximable by nice distributions such as gaussians.

**Strong advice.** Given strong advice in the form of a few samples from the target distribution, (Koehler & Vuong, 2023; Koehler et al., 2025; Gay et al., 2025) show that the problem is generically solvable: for a mixture with $m$ components, given $\tilde{O}(m/\epsilon^2)$ samples, a fresh sample within distance $\epsilon$ in TV can be generated by simply running the Markov chain starting from a random sample; this is termed data-based initialization. This framework works for both continuous and discrete settings. Although the assumption is strong, it is reasonable in the setting of generative modeling, when a dataset of samples is given and the task is to learn to generate new samples.

**Weak advice.** Given results on lower bounds in the no advice setting and the lack of strong advice in many problems, it is natural to try for general results given weaker information. As mode location is an inherent challenge, a natural assumption to isolate the search problem from the sampling problem is to assume we already have *warm starts* to the modes, e.g. obtained by multiple runs of optimization. (Tawn et al., 2021) introduce the annealed-leap point sampler, which combines tempering towards a mixture of peaked distributions, with teleportation, and gives asymptotic analysis in the limit as the modes become gaussian (Roberts et al., 2022). Another kind of information which can be considered as weak advice is that of a few *reaction coordinates* that are assumed to be the main obstacle to fast mixing; algorithms can take advantage of this by stratifying the landscape and forcing exploration in those directions. Examples include umbrella sampling (Torrie & Valleau, 1977; Thiede et al., 2016) (with analysis in (Dinner et al., 2020)), the Wang–Landau algorithm (Wang & Landau, 2001), and adaptive biasing force (Darve & Pohorille, 2001).

**Theoretical tools.** We highlight some theoretical tools that are useful for analyzing sampling for multimodal distributions. Firstly, Markov chain decomposition theorems (Madras & Randall, 2002; Woodard et al., 2009a; Ge et al., 2018a) or two-scale functional inequalities (Otto & Reznikoff, 2007; Grunewald et al., 2009; Lelièvre, 2009; Chen et al., 2021) show that functional inequalities for a Markov chain or process hold given that they hold locally (restricted to some component or coordinate) and that they hold for a projected process that tracks flow or closeness between the components. A number of works quantify and apply local mixing: Although Langevin diffusion does not generally converge quickly, (Balasubramanian et al., 2022) show it is efficient to sample with small relative Fisher information, which for a mixture, corresponds to local mixing within modes but not necessarily global mixing between modes. (Huang et al., 2025) show that sampling is possible under weak Poincaré inequalities (Andrieu et al., 2023) when a warm start can be maintained. Finally, partition function estimation using simulated annealing and Monte Carlo (Dyer et al., 1991; Štefankovič et al., 2009) is well-studied theoretically.

## 1.2 PROBLEM STATEMENT & ASSUMPTIONS

We address the problem of sampling from a target distribution $\pi(x) \propto e^{-V(x)}$ with oracle access to the target distribution $\pi(x)$ by utilizing a set of *warm start points* $\{x_1, \ldots, x_M\}$. (We will formally define this below.) These can be obtained as prior information or from multiple runs of optimization

algorithms. Another setting is when we have samples from a distribution but wish to sample from a modified version of it, for example, when fine-tuning a generative model or incentivizing certain features.

**Problem 1.1.** *Suppose we are given a set of "warm starts" $\{x_1, \ldots, x_M\}$ to the modes of a target distribution $\pi(x) = \sum_{k=1}^{M} \alpha_k \pi_k(x)$. Assume query access to $\pi(x)$ up to a normalization constant, and possibly $\nabla \ln \pi$ (in the case of $\mathbb{R}^d$). Produce a sample that is $\epsilon$-close in total variation distance to $\pi(x)$.*

Note that we only assume the existence of a decomposition of $\pi$, not that the $\pi_k$ are known. To introduce our algorithm, we fix a family of density functions $q_\beta$ on $X$, which have the property $q_\beta \to \delta_0$ weakly as $\beta \to \infty$ and $q_\beta = 1$ if $\beta = 0$. For example, the $q_\beta$ could be Gaussians in $\mathbb{R}^d$ or product distributions on the hypercube. We will apply simulated tempering to the sequence of distributions (for $\beta$ ranging from 0 to very large)

$$\tilde{p}_\beta(x) \propto \pi(x) \sum_{k=1}^{M} w_{\beta,k} q_\beta(x - x_k),$$

for some weights $w_{\beta,k}$ estimated by the algorithm. (On the hypercube, addition is understood in $\mathbb{Z}/2$.) Essentially, we tilt the target distribution towards the set of warm start points. At the coldest level, the distribution becomes approximately a mixture of point masses, and because of their similar shape, the teleportation step of our algorithm allows samples to move between the warm start points.

We will make the following assumptions on $\pi(x)$ and its components $\pi_k(x)$. The general idea of the warm start assumption (part 2 below) is that a significant portion of the mass should be located in the component that corresponds to the product between the component $\pi_k(x)$ and the $q_\beta$ centered at the corresponding warm start point $x_k$. We call this the *tilt* towards $x_k$ of the distribution $\pi_k$. In addition, we assume that each of these tilts satisfy a Poincaré inequality.

**Assumption 1.1.** *Suppose that $\pi(x)$ is a distribution on $\Omega$, $q_\beta : \Omega \to \mathbb{R}, \beta \geq 0$ are functions with $q_0 \equiv 1$. Fix a way of associating a distribution $p(x)$ on $\Omega$ with a Markov process with generator $\mathscr{L}_p$ that has $p$ as stationary distribution.*

1. *(Mixture distribution) The target distribution $\pi(x)$ is a mixture distribution $\pi(x) = \sum_{k=1}^{M} \alpha_k \pi_k(x)$, where $\alpha_k \geq 0$ and $\pi_k$ is a probability distribution.*

2. *($x_i$ are warm starts) For each $i \in [M]$ and for every $\beta \geq 0$,*

$$\int_X \alpha_i \pi_i(x) q_\beta(x - x_i) dx \geq c_{tilt} \int_X \pi(x) q_\beta(x - x_i) dx.$$

3. *(Local mixing) For all $i \in [M]$, $p_{\beta,i}(x) \propto \pi_i(x) q_\beta(x - x_i)$ satisfies a Poincaré inequality of the form*

$$\mathrm{Var}_{p_{\beta,i}}(f) \leq C_{\mathrm{P}} \mathscr{E}_{p_{\beta,i}}(f, f),$$

*where $\mathscr{E}_\pi(f, f) = -\langle f, \mathscr{L}_\pi f \rangle_\pi$ is the Dirichlet form and $\mathscr{L}_\pi$ is the generator of the Markov process with stationary distribution $\pi$.*

4. *(Markov chain decomposes) Whenever $p = \sum_{i=1}^{m} a_i p_i$, $a_i \geq 0$ is a mixture distribution on $\Omega$, the generators decompose: $\langle f, \mathscr{L}_p f \rangle_p \leq \sum_{i=1}^{m} a_i \langle f, \mathscr{L}_{p_i} f \rangle_{p_i}$.*

Many common Markov chains have generators which satisfy the last assumption, for example, Langevin diffusion or the Metropolis random walk on $\mathbb{R}^d$ and Glauber dynamics on product spaces. See (Lee & Santana-Gijzen, 2024) for a complete discussion with proofs. We work with continuous-time Markov processes for technical convenience; any discrete time Markov chain can be converted to a continuous-time by letting the waiting time between jumps be exponential random variables. For a discussion of the limitations of the warm start assumption, see Section 6. For the main theorem, we will make some additional assumptions on the tempering scheme in Assumptions 3.1.

## 2 ALGORITHMS

### 2.1 INGREDIENTS: SIMULATED TEMPERING AND TELEPORTATION

To introduce our main algorithm, we first define simulated tempering and the leap point process. These two algorithms will be the primary components of our main algorithm.

Simulated tempering is a classical approach to sampling from multimodal target distributions (Marinari & Parisi, 1992), where the target distribution is (typically) tempered to smoother (high-temperature) distributions that allow mixing between modes. Particles are allowed to transition between temperatures (with appropriate Metropolis-Hastings acceptance ratio) in addition to moving within their current temperature, and we take the samples that are at the desired temperature.

In our setting, the target measure is instead tempered to more peaked (colder temperature) distributions, and then at the coldest temperature the leap point process is applied to transition particles to different components of the mixture measure. This works particularly well given a set of warm starts $\{x_k\}_{k=1}^M$ since the mixture $\pi_0(x)$ can be tempered to peak around each $x_i$ and then a teleportation map $g_{jj'}$ can be defined to move points around $x_j$ to around $x_{j'}$. We define simulated tempering generally to apply for any specified sequence of distributions.

**Definition 2.1.** *Given a sequence of Markov processes $M_i$ with stationary distributions $p_i$, $1 \leq i \leq L$ on state space $\Omega$ and level weights $r_i$, we define* **simulated tempering** *to be the process on $\Omega \times [L]$ as follows.*

1. *Evolve $(x, i) \in \Omega$ according to $M_i$.*

2. *Propose jumps with rate $\lambda$. When a jump is proposed, change $i$ to $i'$ with probability*

$$\frac{1}{2} \min \left\{ \frac{r_{i'} p_{i'}(x)}{r_i p_i(x)}, 1 \right\}, \quad i' = i \pm 1; \tag{2.1}$$

*otherwise stay at $i \in [L]$.*

It is simple to check that the stationary distribution is $p(x, i) = \sum_{j=1}^L r_j p_j(x) I\{i = j\}$ (Marinari & Parisi, 1992; Neal, 1996). In our case, the $p_i$ will be chosen as $\tilde{p}_{\beta_i}$. For ease of notation, we will overload notation by replacing $\beta_i$ by $i$ in subscripts, e.g., $\tilde{p}_i := \tilde{p}_{\beta_i}$.

For the definition of the leap point process, we assume we are given a set of teleportation functions $g_{jj'}$; for $\mathbb{R}^d$, a simple choice is translation between modes, $g_{jj'}(x) = x + x_{j'} - x_j$.

**Definition 2.2.** *Given a set of teleportation functions $g_{ij}$, $i, j \in [M]$ satisfying Definition D.1 and a Markov process $P$ with stationary distribution $q$, define the* **leap point process** *on the state space $\Omega$ as follows.*

1. *Evolve $x \in \Omega$ according to $P$.*

2. *Propose leaps with rate $\gamma$. When a leap is proposed, choose $j$ and $j'$ uniformly and leap to $g_{jj'}(x)$ with probability*

$$\frac{1}{M} \min \left\{ \frac{g_{jj'}^{\#} q(x)}{q(x)}, 1 \right\}, \quad \forall j' \neq j; \tag{2.2}$$

*otherwise stay at $x$.*

Note that we define $j$ to be randomly sampled, which differs slightly from from (Roberts et al., 2022; Tawn et al., 2020a; 2021), where the current position $x$ of the Markov chain is assigned to a mode $j = \arg \min_k d(x, x_k)$; this is only for ease of analysis.

Our Markov chain uses simulated tempering to perform temperature swaps and employs the leap point process at the coldest temperature to mix between modes. Formally, we define our process on the level of the generators by adding together the original generator, the simulated tempering jumps, and the leaps at the coldest level ($i = 1$). We defer a formal treatment to Section D.1. See Algorithm 2 for pseudocode for simulation.

## 2.2 Weight Estimation & Level Balance

A challenge for tempering algorithms is the potential for bottlenecks to prevent the chain from exploring the entire state space. These bottlenecks form when the probability mass of specific modes becomes vanishingly small at certain levels. Our algorithm explicitly addresses this by iteratively estimating the weights—modal and level—to maintain the following balance condition.

**Definition 2.3.** *We define* **component balance** *with constant $C_1$ for partition functions $Z_{i,k} = \int_\Omega \alpha_k \pi_k(x) q_i(x - x_k) dx$ and weights $\{w_{i,k}\}_{i \in [L], k \in [M]}$ to be the condition*

$$\frac{w_{i,k} Z_{i,k}}{w_{i,k'} Z_{i,k'}} \in \left[\frac{1}{C_1}, C_1\right] \quad \text{for all } i \in [L], k, k' \in [M]. \quad (2.3)$$

*We define* **level balance** *with constant $C_2$ for partition functions $Z_i = \int_\Omega \pi(x) \cdot \sum_k w_{ik} q_i(x - x_k) dx$ and weights $\{r_i\}_{i=1}^L$ as*

$$\frac{r_i Z_i}{r_{i'} Z_{i'}} \in \left[\frac{1}{C_2}, C_2\right] \quad \text{for all } i, i' \in [L]. \quad (2.4)$$

Since level balance is enforced between each level, no exponentially bad bottlenecks can form between the coldest and warmest levels. Example B.2 illustrates a simple setting (a mixture of two Gaussians with different covariances) where exponential bottlenecks form between cold and warm levels.

The following figure is meant to provide some visual context for what poor mixing of the projected chain due to improper modal balance looks like (Figure 1) and how re-balancing by $w_{ik}$ alleviates the bottleneck (Figure 2). In the toy graphic example the partition functions are rescaled by modal weights $w_{ik}$, in practice, our algorithm also rescales each level weight by $r_i$. These weights can be thought of as tuning devices where for some fixed level $l$ the weights $w_{l,k}$ can tune the size of individual balls and the weight $r_l$ simultaneously tunes the size of all balls at one level.

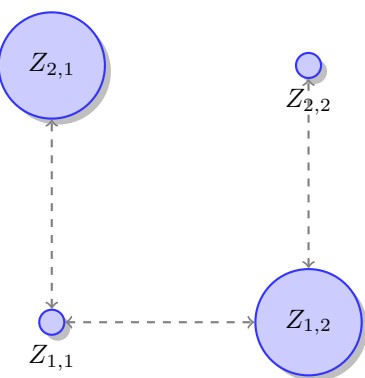

Figure 1: Unscaled modal weights ($Z_{ik}$). The different node sizes show the imbalance, creating bottlenecks that hinder sampling.

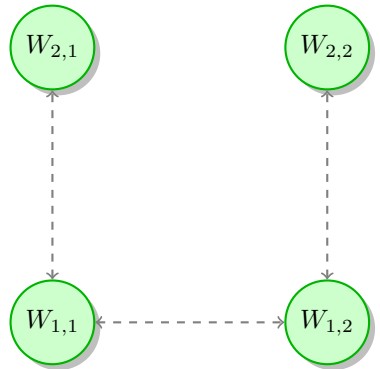

Figure 2: Scaled modal weights ($W_{ik} = w_{ik} Z_{ik}$). After re-weighting, all components are balanced, represented by the uniform node size.

## 2.3 Main Algorithm

Our algorithm is an inductive process which uses an auxiliary variable $\beta_1 > \cdots > \beta_L = 0$ to define a sequence of distributions $\{\sum_{k=1}^M w_{i,k} q_{\beta_i,k}(x)\}_{i=1}^L$ which temper peaked multimodal distributions to the target distribution. The main Algorithm 1 will inductively run Algorithm 2 (vanilla ALPS) to level $l$ and then approximate the weights of the component functions at $l + 1$ via Algorithm 3 (reweighting via partition function estimation). To start off, Algorithm 1 requires an estimation of the partition functions of $\tilde{\pi}_{1,k} = \pi_1(x) q_{1,k}(x)$. In Section I, we show that under appropriate conditions, for large enough $\beta_1$, these estimates can be obtained.

Algorithm 2 (with all levels) is run once all the weights are learned; this is akin to a vanilla version of ALPS (Roberts et al., 2022). As described in Section 2.1, it incorporates simulated tempering

to transition between adjacent temperature levels and the leap point process at the coldest level to transition between modes.

Algorithm 3 runs Algorithm 2 to level $l$ to acquire $N$ samples at the $l$-th level. Then the weights at level $l+1$ are approximated as $w_{l+1,k} = \left( \frac{1}{N} \sum_{j=1}^{N} \frac{\pi(x_j) q_{l+1}(x_j - x_k)}{\bar{p}(x_j, i_j)} I\{i_j = l\} \right)^{-1}$ and $r_{l+1}^{(l)} = \left( \frac{1}{N} \sum_{i=1}^{N} \frac{\pi(x_j) \cdot \left( \sum_k w_{l+1,k} q_{l+1}(x_j - x_k) \right)}{\bar{p}(x_j, i_j)} I\{i_j = l\} \right)^{-1}$. After these weights have been estimated, Algorithm 3 re-runs the chain, this time to level $l+1$, acquiring samples at levels. Then the level weights are adjusted via empirical occupancy, i.e., $r_i^{(l+1)} = r_i^{(l)} \left/ \frac{1}{N} \sum_{j=1}^{N} I\{i_j = i\} \right.$.

For clarity, we note our algorithm and analysis is akin to the vanilla version of ALPS (Roberts et al., 2022). This has the core algorithmic ideas of, but is different from the full version (Tawn et al., 2021), which is equipped with online mode location and parallel tempering.

---

**Algorithm 1** Main Algorithm: Resampled ALPS

---

**INPUT:** Temperature scale $\beta_1 > \beta_2 > \cdots > \beta_L = 0$ and weights $\{w_{1,k}\}_{k=1}^{M}$ satisfying level balance (2.3).
**OUTPUT:** A sample $x \in \mathbb{R}^d$
    **for** $l = 1 \to L$ **do**
2:    Input weights $\{w_{i,k}\}$, $\{r_i^{(l)}\}_{i=1}^{l}$ for $i \in [1, l]$ and $k \in [1, M]$ with temperature scale $\beta_1 > \beta_2 > \cdots > \beta_l$, time $T$ and rates $\lambda, \gamma$.
       **if** $l < L$ **then**
4:       Run Algorithm 3 (reweighting with partition function estimates) to obtain weights $\{w_{l+1,k}\}_{k=1}^{M}$ and $\{r_i^{(l+1)}\}_{i=1}^{l+1}$.
       **else if** $l = L$ **then**
6:       Run Algorithm 2 (vanilla ALPS) and return sample $x \in \mathbb{R}^d$.
       **end if**
8: **end for**

---

## 3   Main Result

We make some additional technical assumptions for the main theorem. We later show in Section I that these assumptions are satisfied in representative settings.

**Assumption 3.1.** *Defining* $\pi_{l,k}(x) \propto \alpha_k \pi_k(x) q_l(x - x_k)$, $\bar{\pi}_{l,k} = \pi(x) q_l(x - x_k)$, $Z_{l,k} = \int \bar{\pi}_{l,k}$ *for* $l \in [1, L]$, $k \in [1, M]$,

    1. *(Closeness at adjacent temperatures)* $\chi^2 \left( \pi_{l+1,k} || \pi_{l,k} \right) = O(1)$, $\chi^2 \left( \frac{\bar{\pi}_{l+1,k}}{Z_{l+1,k}} || \frac{\bar{\pi}_{l,k}}{Z_{l,k}} \right) = O(1)$.

    2. *(Closeness for components at lowest temperature)* $\chi^2 \left( \pi_{1,k} || \pi_{1,j} \right) = O(1)$.

    3. *(Warmness of initial distribution) The initial distribution* $\nu_0(x, i)$ *satisfies* $\left\| \frac{\nu_0(x,i)}{p(x,i)} \right\|_{\infty} \leq U$.

    4. *Component balance with constant* $O(1)$ *(Definition 2.3) is satisfied when* $L = 1$.

Given reasonable choices of tilting functions $q_\beta$, Assumption 1 requires the temperature ladder to be sufficiently closely spaced and Assumption 2 requires starting out at cold enough temperature so that teleportation is accepted with good probability. Assumption 3 requires the initialization of the samples to be close enough to the chain (this is possible by initializing at the lowest temperature, which is easily approximable). Assumption 4 is the base case of the inductive hypothesis and again depends on the lowest temperature distribution being approximable. As an example, we show these assumptions hold for Gaussian tilts on $\mathbb{R}^d$; we have not attempted to optimize the number of levels.

**Proposition 3.2.** *(Tempering by Gaussians) Let Assumptions 1.1 hold for* $\pi(x) = \sum_{k=1}^{M} \alpha_k \pi_k(x)$ *with* $\alpha_k \pi_k(x) = e^{-f_k(x)}$ *where* $f_k(x)$ *is L-smooth. In addition, assume that a log-Sobolev in-*

*equality holds with constant $C_{LS}$ for $\pi_{i,j,k} \propto \pi_j(x) \cdot q_i(x - x_k)$, for all $i \in [L], j, k \in [M]$. Define $q_i(x) = e^{-\beta_i \frac{\|x\|^2}{2}}$ and the teleportation map $g_{jj'}(x) = x - x_j + x_{j'}$. Lastly, choose $\Delta\beta = |\beta_i - \beta_{i+1}| = O(\frac{1}{C_{LS}d+r^2})$ and $\beta_1 = O(L^2 D^2 d)$, with $\|x_j - x_j^*\| \le D$, where $x_j^*$ is the true mode and $\|x_j - \mathbb{E}_{p_{i,j,k}} x\| \le r$ for all $j, k \in [M]$. Then Assumptions 3.1 hold with $U = O\left(\frac{1}{c_{tilt}^2}\right)$ and $w_{1k} \propto \frac{1}{\pi(x_k)}$ on a temperature schedule of $\Omega(d^2)$ levels.*

We now state our main theorem.

**Theorem 3.3.** *Suppose we are given a warm start of points $\{x_1, \ldots, x_M\}$ from a target distribution $p(x)$. Fix a family of density functions $q_\beta, \beta > 0$ on $X$, with $q_\beta = 1$ if $\beta = 0$. Suppose Assumptions 1.1 and 3.1 hold. Then Algorithm 1 with parameters[1]*

$$T = \Omega\left(\text{poly}\left(U, \tilde{C}, M, L, \frac{1}{c_{tilt}}, \frac{1}{\gamma}, \frac{1}{\lambda}, \frac{1}{\epsilon}\right)\right), \qquad N = \Omega\left(\text{poly}\left(L, M, U, \frac{1}{c_{tilt}}, \frac{1}{\delta}\right)\right)$$

*produces samples from $\hat{p}(x)$ such that with probability $1 - \delta$, $TV(\hat{p}(x), \pi(x)) \le \epsilon$.*

## 4  PROOF OVERVIEW

A standard approach to proving mixing time bounds for tempering Markov chains is to use a Markov decomposition theorem Ge et al. (2018b). Decomposition theorems allow for mixing to be quantified in terms of mixing within the components and mixing within the projected chain defined through probability flow between components. However, in our setting, a direct approach is complicated by our choice of tempered distributions $p_\beta(x) \propto \sum_j \alpha_j \pi_j(x) \cdot \sum_k w_{\beta,k} q_\beta(x - x_k)$. This formulation introduces cross terms of the form $\pi_j(x) q_\beta(x - x_k)$ for $j \ne k$, where the component $\pi_j(x)$ is tilted towards the incorrect mode. We refer to these cross terms as the bad portion of the target distribution and they prevent us from a clean analysis on the projected state. Our proof strategy is therefore to isolate the good components and perform the analysis on that portion.

The proof proceeds in four main steps:

1. **Quantifying Convergence on the "Good" Components:** To remedy the cross term issue, we formulate $\chi^2$ bounds that quantify the mixing of the whole chain in terms of the mixing on the good component, Lemma F.1. We accompany this with a lower bound on the portion of the mass from the Markov chain within the good component, Lemma F.2 and then use this to quantify the rate at which that portion converges to the good component itself. Since our Markov chain operates on the extended state space $\Omega \times [L]$, we generalize these results to the extended state space and quantify the amount of mass that mixes into the target level (which is entirely in the good part), Lemma F.3.

2. **Applying the Markov Decomposition Theorem:** Our previous analysis quantifies the mixing of the whole chain by the Poincaré constant corresponding to the good component, $C_{PI}(p_0(x, i))$. Now, working within the good component, we are able to bound this Poincaré constant using classical Markov decomposition theorems, Theorem E.1, which upper bound $C_{PI}(p_0(x, i))$ by the mixing within the modes and the mixing on the projected chain. The most difficult part is ensuring that there are no bottlenecks in the projected chain which would cause the Poincaré constant to explode.

3. **Controlling the Projected Chain:** The most significant challenge in the ST literature is ensuring that the projected chain mixes rapidly. Torpid mixing is primarily caused by bottlenecks in the projected chain which in turn causes the corresponding Poincaré constant to explode. Our work address this by controlling the probability flow between modes, which is ultimately determined by maintaining relative mass between modes.

4. **Maintaining Balance via Inductive Estimation:** To maintain proper modal and level balance (Definition 2.3), we define the good component as $p_0(x, i) \propto \sum_i r_i \sum_k \alpha_k w_{\beta_i,k} \pi_k(x) q_{\beta_i}(x - x_k)$. This provides us with control over both the level weights $\{r_i\}_{i \in [L]}$ and the modal weights $\{w_{i,k}\}_{i \in [L], k \in [M]}$—think of these as knobs that

---

[1]$L$ is the number of levels and $M$ the number of warm start points. $c_{tilt}$, $C_{ij}$ and $\tilde{C} = \max_{ij} C_{ij}$ are defined in Assumptions 1.1, and $\gamma, \lambda$ are hyper-parameters defined in D.5.

tune the probability flow between component measures. Good level balance is ensured by induction on the temperature levels, at each level showing that Definition 2.3 holds with $C_1 = \text{poly}(\frac{U}{c_{tilt}})$ and $C_2 = \text{poly}(\frac{U}{c_{tilt}})$ for weights $\{w_{i,k}\}_{i \in [L], k \in [M]}$ and $\{r_i\}_{i \in [L]}$ approximated by Monte Carlo averages is done in Theorem G.13, G.17. By dynamically re-weighting in this manner, the algorithm guarantees proper relative mass between modes, preventing bottlenecks in the projected chain.

## 5 EXPERIMENTS

We demonstrate that the Reweighted ALPS algorithm works well on a three mode mixture of a quartic exponential, Cauchy distribution and Gaussian distribution in $d = 5$. We compare the results of Reweighted ALPS to the original ALPS algorithm, described in Section **??**, using several different scaling factors and temperature ladders. We define the target distribution $\pi : \mathbb{R} \to \mathbb{R}^d$ as

$$\pi(x; \sigma_1, \sigma_2) = w_1 \text{Cauchy}(-\vec{15}, 1) + w_2 \frac{\exp\left(-\frac{||x||^4}{\sigma_1} - \frac{||x||^2}{\sigma_2}\right)}{Z} + w_3 \text{N}(\vec{15}, \mathbb{I}_d).$$

We ran both the ALPS and Reweighted ALPS algorithms with target distribution $\pi(x; \sigma_1 = .2, \sigma_2 = 20)$ with weights $\vec{w} = [.1, .8, .1]$. With this target distribution the true Hessian of the quartic mode is a poor estimate of its covariance which prevents the ALPS algorithm from jumping to the second mode. This is shown in Figure 3b where the first coordinate of the jump proposals are largely outside the support of the distribution itself, causing a low metropolis acceptance rate.

Several $\vec{\beta}$ ladders were tested, maintaining in the spirit of the ALPS results the suggested $d = 5$ temperature levels. The primary issue was in choosing $\beta_{max} = \beta_5$, too large a $\beta_5$ led to poor temperature swapping with no seen benefits to mode leap acceptance. Ultimately, in order to maintain good mixing between temperature levels, the ladder $\beta_{ALPS} = [1, 1.12, 1.25, 1.38, 1.5]$ was used in the results for ALPS. To keep an even playing field we chose $d = 5$ temperature levels for Reweighted ALPS of $\beta_{Re-ALPS} = [2.8, 1.25, .56, .25, 0]$.

Given these temperature ladders we ran both algorithms for $N = 25,000$ attempted temperature swaps. ALPS reported a mode leap acceptance rate of $.1$ (224/2236 attempts). However, it should be noted that the acceptance rate of $.1$ is due to swaps between modes 1 and 3, the mode acceptance rate to mode acceptance rate to mode 2 was $0/1791$ attempted swaps. The temperature swap rate was $.582$ (12452/25000 attempts). The poor acceptance rate to mode 2 can be visualized in Figure 3b. Reweighted ALPS reported a mode leap acceptance rate of $.435$ (1580/3630 attempts) and a swap rate of $.673$ (16824/25000 attempts).

Importantly, from the $N = 10,000$ samples, the Reweighted ALPS algorithm reported a modal occupancy of $[.095, .808, 0.96]$ which is close to the true weight of $[.1, .8, .1]$.

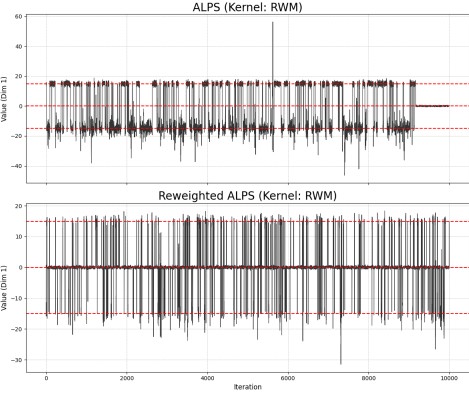

(a) Plots of the first coordinate of $N = 10,000$ samples acquired for both ALPS and Reweighted ALPS. The horizontal red lines indicate the modal points.

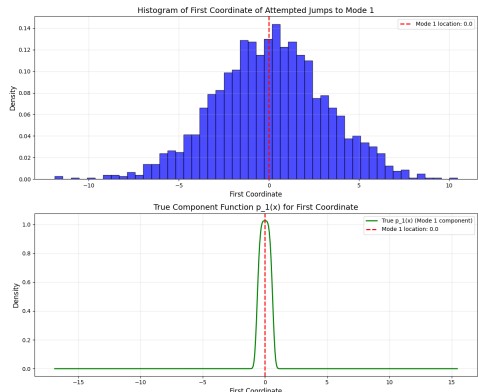

(b) A histogram of the first coordinate of $N = 1791$ proposal jumps to mode 2 and the true distribution of mode 2.

Figure 3: Samples were acquired from $\pi(x; \sigma_1 = .2, \sigma_2 = 20)$ with $d = 5$.

This experimental work provides us with two key insights:

1. **Hessian information can be misleading:** A Hessian can exist, be non-zero and can be a poor estimate of the covariance structure for a distribution $p(x)$. For example, choosing parameters $\sigma_1 = .2$ and $\sigma_2 = 20$ leads to $\pi(x; \sigma_1, \sigma_2)$ having extremely light tails around mode 2. The Hessian, however, is derived solely from the high-variance quadratic term, which incorrectly suggests the mode is broad. Using the Hessian information as a stand in for the covariance matrix in the ALPS algorithm leads to samples being drawn far from the support of mode 2, causing the original ALPS algorithm to rarely reach this mode.

   In practice, this introduces a significant risk to the ALPS algorithm, as it implies that for any unknown distribution, modes with a similar mismatch between their local curvature and tail behavior might be missed entirely. Reweighted ALPS, on the other hand, does not require any Hessian information and cannot be mislead in this fashion.

2. **Reweighted ALPS maintains modal weightings:** The primary benefit of the ALPS algorithm is its ability to leverage Hessian Adjusted Tempering to maintain modal weights. By maintaining modal weights it addresses the torpidly mixing example (B.1) for standard ST algorithms, introduced in (Woodard et al., 2009b). Our example shows with weights $w = [.1, .8, .1]$ and scaling $\sigma_1 = .2, \sigma_2 = 20$ that Reweighted ALPS can achieve rapid mixing in this setting.

## 6 CONCLUSION AND FURTHER WORK

We prove the first general polynomial-time bounds for sampling from multimodal distributions under the "weak advice" of warm start points to the different modes. Our algorithm is a modified version of the ALPS algorithm (Tawn et al., 2021) that is designed to work well on multi-modal target distributions with difficult geometries. The core innovation is a modified tempering schedule, $\pi_\beta(x) \propto \pi(x) \cdot \sum_k w_{\beta,k} q_\beta(x - x_k)$, where we estimate the weights via Monte Carlo simulation to keep components balanced. As our focus is on an initial theoretical analysis, there are several avenues for future theoretical and computational work towards making the algorithm practical.

Computationally speaking, our base algorithm has several computational inefficiencies, such as estimating the next level weights at every iteration from a fresh set of samples; it also requires warm start points to already be located. Hence, a beneficial modification would be to update weights and find additional modes in an online manner. As our algorithm is tailored for ease of theoretical analysis, we do not recommend it in its current form as a replacement for ALPS, and believe that a hybrid algorithm incorporating our approach to weight rebalancing may be ultimately more practical. We leave to future work the design of a more versatile and efficient algorithm which works on practical problems in high dimensions and with complex geometries.

**Warm start assumption.** An important limitation is our definition of a warm start point, in terms of the tilt having significant mass. Applied to components of different shapes (e.g., Gaussians in Proposition I.8), this may require separation conditions between the components. In order to loosen the definition of warm start point, we may need adaptive tilting schemes, e.g. Gaussians with covariances chosen adaptively. As a concrete theoretical problem to guide algorithm design, we propose this open question: *Is there a polynomial-time sampler for $\pi = \sum_{i=1}^{M} w_i \pi_i$ where each $\pi_i$ is a log-concave distribution, given one sample $x_i \sim \pi_i$ from each component?*

Some more technical limitations of our warm start assumption is that (1) we currently assume a 1-to-1 correspondence between warm starts and modes, and (2) taking $\beta = 0$, each component is required to have mass that is lower-bounded. We can hope that this can be relaxed to making sure that all modes are covered (and allowing spurious points), and that modes having small mass can be disregarded.

Finally, theoretical analysis is also highly desirable for other algorithms in the weak advice setting, such as those based on stratification. Another promising direction is to combine information on warm starts with neural network flow-based methods for sampling (Albergo et al., 2023; Vargas et al., 2023; Albergo & Vanden-Eijnden, 2024), as well as learning the interpolation (Máté & Fleuret, 2023).

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

# A    ORGANIZATION OF PAPER

We provide a brief overview of each section. In Section 2, we define the simulated tempering and Re-ALPS algorithms. In Section 3, we present the main results of our paper, and we provide an overview of the proof in **??**.

In Section D, we cast Algorithm 2 as a continuous time process with Markov generator $\mathscr{L}$. We then show that the Markov generator $\mathscr{L}$ satisfies the decomposition assumptions of Theorem E.1 in Section E.

The analysis in Section E is similar to the work done in Ge et al. (2018c). We show that under basic assumptions the Poincaré constant corresponding to a stationary distribution $p_i(x) = \sum_{j=1}^{M} w_{ij} p_{ij}(x)$ of the Markov process $P$ can be bounded by a function of Poincaré constants corresponding to the component measures and the Poincaré constant of the projected chain capturing transitions between components (defined in the assumptions of Theorem E.1). Since we've shown in Section D that the continuous time process corresponding to Algorithm 1 satisfies the assumptions of Theorem 3.3, the decomposition theorem allows us to proceed after finding a bound on the Poincaré constant of the projected chain.

A major obstacle in our analysis is the cross terms that appear by defining $\tilde{p}_\beta \propto \sum_k \alpha_k \pi_k(x) \cdot \sum_k w_{\beta,k} q_\beta(x - x_k)$. Ideally our algorithm is able to mix well into the aligned components $\pi_k(x) q_\beta(x - x_k)$ while ignoring the cross terms which will naturally have negligible weight (see Fig. 4). Under reasonable assumptions, as we will later show, it makes sense to refer to the portion of the product where $j = k$, the aligned components, as the good portion. In Section F, we prove convergence in chi-squared divergence for the good portion of the stationary distribution. We show that convergence can be bounded by a function which depends on the Poincaré constant corresponding to the good set as well as an expected value that describes the "flow" into the good set.

Our main algorithm, Algorithm 1, inductively runs Algorithm 3 to estimate the partition functions at the subsequent level. To estimate the partition functions, Algorithm 3 runs Re-ALPS to the current level and collects $N$ samples. The samples at level $l$ are used to obtain a Monte Carlo estimate of the partition functions at the next level yielding weights $\{w_{l+1,k}\}_{k=1}^{M}$ and $r_{l+1}$. With these weights, the algorithm is then run one more time to level $l + 1$ obtaining another $N$ samples. This time the samples are used to get an estimate of the level weights by empirical occupancy and this occupancy is used to adjust the level weightings $\{r_i\}_{i=1}^{l+1}$.

In Section G.3, we show that under the inductive hypothesis, Assumptions G.1, and Assumptions 3.1, weights $\{r_l\}_{l=1}^{L}$ and $\{w_{l,k}\}_{l \in [1,L], k \in [1,M]}$ can be chosen to maintain level and component balance between the partition functions. More precisely, the weights are chosen so that there exists a constant $C_1$ so that $\frac{1}{C_1} \le \frac{w_{l,k} Z_{l,k}}{w_{l,k'} Z_{l,k'}} \le C_1$ for $k, k' \in [1, M]$ and a constant $C_2$ so that $\frac{1}{C_2} \le \frac{r_l Z_l}{r_{l'} Z_{l'}} \le C_2$ for $l, l' \in [1, L]$. Maintaining this level balance prevents bottlenecks—a mode having low weight at a level so that is is hard to obtains samples at subsequent temperatures—and so allows for bounds on the Poincaré constant of the projected chain; see Lemma G.10.

In Section H, we prove the main results, which follow from the results in Section G.3.

Section I provides a general setting under which the assumptions of the main theorem hold. Assumptions 3.1 focus on the initial distribution and tempering scheme used to run Algorithm 1. In Section I, we show that Assumptions 3.1 hold in $\mathbb{R}^d$ for $q_i(x)$ chosen to be Gaussians and in the general case where the component measure of the target function is specified as $p_k(x) = e^{-f_k(x)}$, where $f_k(x)$ is $L$-smooth. Assumptions 1.1 focus on the target measure $p(x) = \sum_{k=1}^{M} \alpha_k p_k(x)$. In Section I.2, we show families of target measures where Assumptions 1.1 hold.

# B    BACKGROUND

## B.1    NOTATION

We denote the target distribution on $\Omega$ by

$$\pi(x) \propto e^{-V(x)},$$

and assume it decomposes as a mixture

$$\pi(x) = \sum_{k=1}^{M} \alpha_k \pi_k(x)$$

where $\pi_k(x)$ are normalized component measures with corresponding weights $\alpha_k$. The set of warm start points is given by $\{x_1, \ldots, x_M\} \subset \Omega$. The tempering functions are denoted $q_\beta(\cdot)$ and are unnormalized distributions satisfiying $q_\beta \to \delta_0$, where $\delta_0$ is the Dirac delta measure, as $\beta \to \infty$ and $q_\beta = 1$ for $\beta = 0$. The unnormalized tempered distributions are given by

$$\tilde{p}_\beta(x) = \pi(x) \sum_{k=1}^{M} w_{\beta,k} q_\beta(x - x_k) \tag{B.1}$$

where $w_{\beta,k}$ are learned weights. In Section G.3 we define the following for ease of computation. The target measure tilted by $q_\beta(x - x_k)$ on level $l$ is given by

$$\bar{\pi}_{l,k}(x) = \pi(x) \cdot q_l(x - x_k)$$

and the component measure aligned with its correcting tempering function is denoted by

$$\tilde{\pi}_{l,k}(x) = \pi_k(x) \cdot q_l(x - x_k).$$

The partition functions corresponding to these measures are denoted $\bar{Z}_{l,k}$ and $Z_{l,k}$ respectively. In Section G.3, we make use of the unnormalized joint distribution over the temperature levels which is defined to be

$$\tilde{p}(x, i) = \sum_{j=1}^{l} r_j^{(l)} \tilde{p}_j(x) I\{i = j\}. \tag{B.2}$$

Similarly, we define the normalized version to be

$$p(x, i) = \sum_{j=1}^{l} \omega^j p_j(x) I\{i = j\}$$

where $\omega^j = \frac{r_j^{(l)}}{\bar{Z}_j}$. In the context of Section F, we will occasionally refer to the "good" part of the distribution and will denote this unnormalized portion by

$$\tilde{p}_0(x, i) = \sum_{j=1}^{l} r_j^{(l)} \sum_{k=1}^{M} \alpha_k w_{j,k} \pi_k(x) q_j(x - x_k) I\{i = j\}.$$

The good portion conditioned on the level is then denoted by

$$p_{i0}(x) \propto \sum_{k=1}^{M} \alpha_k w_{i,k} \pi_k(x) q_i(x - x_k). \tag{B.3}$$

We can then express the normalized joint distribution as a mixture of the good and bad portions by

$$p(x, i) = \alpha_0 p_0(x, i) + (1 - \alpha_0) p_1(x, i)$$

where $\alpha_0 = \frac{\sum_i \int_\Omega \tilde{p}_0(x,i) dx}{\sum_i \int_\Omega \tilde{p}(x,i) dx}$ is the component weight of the good portion.

## B.2 MOTIVATING EXAMPLES

To motivate tempering to colder temperatures, corresponding to more peaked distributions, we show that in high dimensions, flat components of the mixture distribution can cause the teleport process to have low acceptance probabilities. This leads to poor mixing of the projected chain—mixing between modes—which in turn leads to long run times. In our algorithm, the projected chain at the coldest level has probability flow given by overlap between component measures after a translation which maps a warm start point to another.

**Example B.1** (Low acceptance for teleporting). *Let $\pi(x) = \frac{1}{2}N(0, I_d) + \frac{1}{2}N(\mu_1, \sigma I_d)$ be the mixture of two Gaussians in $\mathbb{R}^d$ and define the teleport function to be the translation $g_{ij}(x) = x - \mu_i + \mu_j$. Then the probability of transitioning from $\mu_0 = 0$ to $\mu_1$, denoted $P(\{0\}, \{1\})$, is given by*

$$P(\{0\}, \{1\}) = \min\left\{ \frac{(1/2\pi)^{\frac{d}{2}} \det(\sigma I)^{-\frac{1}{2}} \exp\left(-\frac{1}{2\sigma}\|x + \mu_1 - \mu_1\|_2^2\right)}{(1/2\pi)^{\frac{d}{2}} \det(I)^{-\frac{1}{2}} \exp\left(-\frac{1}{2}\|x\|_2^2\right)}, 1 \right\}$$

$$= \min\left\{ \frac{\left(\frac{1}{\sigma}\right)^{\frac{d}{2}} \exp\left(-\frac{1}{2\sigma}\|x\|_2^2\right)}{\exp\left(-\frac{1}{2}\|x\|_2^2\right)}, 1 \right\}$$

$$= \min\left\{ \left(\frac{1}{\sigma}\right)^{\frac{d}{2}} \exp\left(\frac{\sigma - 1}{2\sigma}\|x\|_2^2\right), 1 \right\}$$

*It is clear that for $\sigma > 1$ as $d \to \infty$ and $\|x\|$ on the order of $\sqrt{d}\sigma$ we have that $P(\{0\}, \{1\}) \to 0$.*

The following example shows that in high dimensions a bimodal mixture of Gaussians with different variances can have exponentially bad weight distortion when power tempering is applied. Power tempering is one of the most standard tempering methods that takes the target distribution $\pi(x) \propto e^{-V(x)}$ and raises it to the inverse temperature $\beta$ so that at each level $\pi_\beta(x) \propto \pi(x)^\beta$. The same issue arises when tempering towards a prior, $\pi_\beta(x) \propto \pi(x)^\beta q(x)^{1-\beta}$.

**Example B.2** (Bottlenecks with tempering). *(Roberts et al. (2022)) Given target density $\pi(x) = \frac{1}{2}N(x; 0, I_d) + \frac{1}{2}N(x; \mu_1, \sigma I_d)$ and assuming the power tempered target can be given by the mixture Woodard et al. (2009a)*

$$\pi(x) = W_{0,\beta} N\left(0, \frac{I_d}{\beta}\right) + W_{1,\beta} N\left(\mu_1, \frac{\sigma I_d}{\beta}\right)$$

*where the weights are given by $W_{i,\beta} \propto \left(\frac{1}{2}\right)^\beta \det(\sigma I_d)^{\frac{1-\beta}{2}}$. In our case this yields the ratio*

$$\frac{W_{1,\beta}}{W_{0,\beta}} = \sigma^{d(1-\beta)},$$

*which is exponential in the dimension.*

## C  ALGORITHMS

### C.1  COMPARISON OF ALGORITHMS

All three ALPS variants—Full ALPS, Vanilla ALPS, and Reweighted ALPS (Re-ALPS)—share the fundamental approach of annealing to colder temperatures combined with teleportation to enable efficient mode jumping using weak information (warm starts). This core structure is exactly the simulated tempering and teleportation process detailed in the previous section. This annealing process makes the modes peaked around the warm start points at the coldest temperature. This feature enables all three algorithms to perform efficient mode jumping at the coldest level, allowing for inter-modal exploration using only weak information.

All three algorithms seek to approximate modal weights that address the potential for bottlenecks between temperature swaps, but diverge in how they approach the approximation of modal weights.

1. Vanilla ALPS (Roberts et al., 2022) and Full ALPS (Tawn et al., 2021) address the issue of modal weights by applying weight-preserving simulated tempering (Tawn et al., 2020b). This method relies on taking a local Gaussian approximation to the modes, which requires an approximation to the Hessian at the modal points. Then the analytic approximation to the partition function is used as the modal weight. This approach enables both algorithms to be lightweight and computationally efficient. The downside is that these analytical approximates can be poor estimates of the true modal weights when the components are far from Gaussian.

2. Reweighted ALPS is designed for cases where modes have more complex geometries. Instead of an analytical approximation of the modal weights, our algorithm inductively estimates component partition functions via Monte Carlo. These estimates are used to adjust

both the modal weights and the level weights, directly preventing bottlenecks from forming.

It should be noted that ALPS and Vanilla ALPS share the core power tempering structure with analytical stand-ins via Gaussian approximation but differ for sake of analysis. The full version of ALPS includes, online modal and Hessian updates, parallel tempering and a more adaptive HAT tempering scheme.

Table 1: Comparison of Three ALPS Algorithm Versions

| Feature | Full ALPS (Tawn et al., 2021) | Vanilla ALPS (Roberts et al., 2022) | Resampled ALPS |
|---|---|---|---|
| **Core Idea** | A practical, robust algorithm combining parallel tempering, online mode finding, and annealing to sample from complex multimodal distributions. | A simplified version designed for theoretical analysis to study its computational complexity and prove diffusion limits to skew Brownian motion. | A modified version providing the first general polynomial-time bounds for sampling with "warm starts", specifically designed to avoid requiring Hessian information. |
| **Tempering Method** | Uses "Hessian Adjusted Tempered" (HAT) targets (Tawn et al., 2020b), which apply a weight-preserving transformation: $\pi_\beta(x) \propto \pi(x)^\beta \pi(\mu_{\hat{A}})^{1-\beta}$. | For theoretical analysis, assumes a weight-preserving transformation where intermediate distributions are normalized powers of modal components: $\pi_\beta(x) \propto \sum_j w_j g_j^\beta(x)$. | Tilts the target distribution toward warm starts with dynamically estimated weights, avoiding power tempering: $\pi_\beta(x) \propto \pi(x) \sum_k w_{\beta,k} q_\beta(x - x_k)$. |
| **Mode Information** | Finds modes and their Hessians online. | Assumes mode locations are known and the state is pre-allocated to a mode for theoretical analysis. | Requires a set of "warm starts" (approximate mode locations) as input. |
| **Mode Jumping** | At the coldest level ($\beta_{max}$), it uses a Gaussian mixture independence Metropolis-Hastings sampler built from the learned mode locations and Hessians. | For analysis, it assumes perfect, immediate mixing between modes at the coldest temperature ($\beta_{max}$) and no inter-mode mixing otherwise. | At the coldest level, it uses a "leap-point process" that proposes teleportation moves between the provided warm start points to enable mixing across modes. |
| **Weight Handling** | Aims to preserve the relative mass of modes across different temperatures using the HAT formulation, based on a Laplace approximation at the modes. | Assumes for its analysis that a weight-preserving transformation is used, such that the component weights $w_j$ remain constant for all inverse-temperatures $\beta$. | Dynamic estimation of modal weights ($w_{i,k}$) and level weights ($r_i$) via Monte Carlo to explicitly maintain component balance and level balance to prevent bottlenecks. |
| **Implementation** | Implemented using **Parallel Tempering**, where separate chains run at each temperature and swap states. | Analyzes a **Simulated Tempering** process, where a single state moves between temperatures. | Based on a **Simulated Tempering** framework where a single chain moves between different temperature levels. |

## C.2 ALGORITHM DETAILS

---

**Algorithm 2** Vanilla ALPS Main Algorithm

---

**INPUT:** Temperature scale $\beta_1 > \beta_2 > \cdots > \beta_l$, weights $\{w_{i,k}\}$ for $i \in [1, l], k \in [1, M]$ and $\{r_i\}_{i=1}^L$, time $T$ and rates $\lambda, \gamma$.

1: Sample $(x, 1) \sim \sum_{k=1}^M w_{1,k} \pi_{1,k}$
2: **while** $T_n < T$ **do**
3:     Set $T_{n+1} = T_n + \xi_{n+1}$ with $\xi_{n+1} \sim \exp(\gamma)$
4:     **if** $i = 1$ (base level) **then**
5:         Set $T'_{n+1} = T'_n + \xi'_{n+1}$ with $\xi'_{n+1} \sim \exp(\lambda)$
6:         **if** $T'_{n+1} < T_{n+1}$ **then**
7:             Run $K_1$ for $\xi'_{n+1}$ time (discretized)
8:             Choose $j, j' \in [1, M]$ and accept transition to $(g_{jj'}(x), 1)$ with pr. $\min\left\{\frac{g_{jj'}^\# p_1(x)}{p_1(x)}, 1\right\}$
9:         **else**
10:            Run $K_1$ for $\xi_{n+1}$ time (discretized)
11:            Transition to $(x, 2)$ with pr. $\min\left\{\frac{r_2 p_2(x)}{r_1 p_1(x)}, 1\right\}$
12:         **end if**
13:     **else**
14:         Run $K_i$ for $\xi_{n+1}$ time (discretized)
15:         Choose $i' = i \pm 1$ with pr. $\frac{1}{2}$ transition to $(x, i')$ with pr. $\min\left\{\frac{r_{i'} p_{i'}(x)}{r_i p_i(x)}, 1\right\}$.
16:     **end if**
17:     Let $\tilde{T} = \min\left\{T_{n+1}, T'_{n+1}\right\}$ then set $T_{n+1} = \tilde{T}$ and $T'_{n+1} = \tilde{T}$
18: **end while**
19: if final state is $(l, x)$, return sample $x$. Otherwise, re-run the chain.

---

---

**Algorithm 3** Reweighting via Partition Function Estimation

---

1: *Part 1: Estimate component weights for level $l + 1$*
2: Run $\hat{p}_t(x, i)$ from Algorithm 2 to the $l$-th level and obtain samples $\{(x_j, i_j)\}_{j=1}^N \sim p(x, i)$.
3: **for** $k = 1, \ldots, M$ **do**
4:     Set

$$w_{l+1,k} = \frac{1}{\frac{1}{N} \sum_{j=1}^N \frac{\bar{\pi}_{l+1,k}(x_j)}{\hat{p}(x_j, i_j)} I\{i_j = l\}}$$

5: **end for**
6: *Part 2: Estimate weight for level $l + 1$*
7: Run $\hat{p}_t(x, i)$ from Algorithm 2 again to the $l$-th level and obtain samples $\{(x_j, i_j)\}_{j=1}^N \sim p(x, i)$.

8: Set

$$r_{l+1}^{(l)} = \frac{1}{\frac{1}{N} \sum_{i=1}^N \frac{\left( \sum_j \alpha_j \pi_j(x_j) \right) \left( \sum_k w_{l+1,k} q_{l+1}(x_j - x_k) \right)}{\hat{p}(x_j, i_j)} I\{i_j = l\}}$$

9: *Part 3: Re-estimate level weights*
10: Run $\hat{p}_t(x, i)$ from Algorithm 2 again to the $l + 1$-th level and obtain samples $\{(x_j, i_j)\}_{j=1}^N \sim p(x, i)$
11: **for** $i = 1, \ldots, l + 1$ **do**
12:     Set

$$r_i^{(l+1)} = r_i^{(l)} \bigg/ \frac{1}{N} \sum_{j=1}^N I\{i_j = i\}$$

13: **end for**
14: Scale $r_1^{(l+1)}$ by $C_2$
15: Return $\{w_{l+1,k}\}_{k=1}^M$ and $\{r_i^{(l+1)}\}_{i=1}^{l+1}$

---

# D    CONTINUOUS TIME PROCESS

## D.1    LEAP-POINT PROCESS

We define a continuous version of the leap-point process at the coldest temperature. In this setting, the process is defined on the mixture distribution $\sum_{k=1}^M w_k q_k(x)$ and can freely jump from any $q_i$ to $q_j$. Jumps are made according to the Poisson point process at time intervals $T_n - T_{n-1} \sim$ Exponential($\gamma$). This specifies the projected chain as a continuous time process on the state space given by the modes, where the probability flow between the modes is compared using the pushforward $g_{ij}^{\#}$. This allows us to express the generator of the process $\mathscr{L}_{tel}$ on $\Omega$ as the sum of the Markov processes on the continuous state space and the transitions between the modes.

**Definition D.1.** *For $i, j \in [n]$, we define the function $g_{ij}(x)$ to be a function that "teleports" $x \in \Omega$ from mode $i$ to mode $j$ and satisfies the following properties:*

    *1. $g_{ii}$ is the identity,*
$$g_{ii}(x) = x.$$

    *2. $g_{ij}$ is the inverse of $g_{ji}$,*
$$g_{ij}(g_{ji}(x)) = x.$$

    *3. $g_{ij}$ is transitive,*
$$g_{kj}(g_{ik}(x)) = g_{ij}(x).$$

**Definition D.2.** *We define **the continuous leap-point Markov process** $K_{leap}$ with rate $\gamma$ on $\Omega$ as follows:*

    *1. Let $T_n$ be a Poisson point process with rate $\gamma$ so that,*
$$T_n - T_{n-1} \sim Exp(\gamma)$$

2. *The Markov process with state $x \in \Omega$ evolves according to $K$ on the time interval $[T_{n-1}, T_n)$.*

3. *At time $T_n$, with randomly chosen $i, j \in [1, M]$, the Markov process takes a jump to $x \mapsto g_{ij}(x)$ with probability*

$$\frac{1}{M} \min \left\{ \frac{g_{ij}^{\#} q(x)}{q(x)}, 1 \right\}, \quad \forall j \neq i$$

*and stays at $x$ otherwise.*

**Lemma D.3.** *Let $K_{leap}$ be the leap-point Markov process with rate $\gamma$ with stationary distribution $q(x) = \sum_{i=1}^{M} w_i q_i(x)$ on $\Omega$. Then the continuous process has the generator $\mathscr{L}_{leap}$ given by,*

$$\mathscr{L}_{leap} f(x) = \mathscr{L}_{ld} f(x) + \frac{\gamma}{M} \sum_{i=1}^{M} \sum_{j=1}^{M} \min \left\{ \frac{g_{ij}^{\#} q(x)}{q(x)}, 1 \right\} \left( f(g_{ij}(x)) - f(x) \right).$$

*Proof.* Let $P_t f(x) = \mathbb{E}_K[f(x_t)|x_0 = x]$ be the expected value after running the chain for time $t$. Then we decompose the conditional expectation by considering the number of jumps the Poisson process takes. Here, we let $H$ be the kernel of the jump process and calculate

$$P_t f(x) = \mathbb{P}(N_t = 0) \cdot P_t f(x) + \int_0^t P_s H P_{t-s} f(x) \mathbb{P}(t_1 = ds, N_t = 1) + \mathbb{P}(N_t = 2) h$$

$$= (1 - \gamma t + O(t^2)) P_t f(x) + \int_0^t P_s H P_{t-s} f(x)(\gamma + O(s)) ds + O(t^2) h$$

$$\frac{\partial}{\partial t}(P_t f)|_{t=0} = -\gamma f(x) + \mathscr{L}_{ld} f(x) + \gamma H f + O(t)$$

By specifying

$$H f(x) = f(x) + \frac{1}{M} \sum_{i=1}^{M} \sum_{j=1}^{M} \min \left\{ \frac{g_{ij}^{\#} q(x)}{q(x)}, 1 \right\} \left( f(g_{ij}(x)) - f(x) \right)$$

we get the desired operator $\mathscr{L}$.

$\square$

**Corollary D.4.** *The corresponding Dirichlet form for the process $\mathscr{L}_{leap}$ is given by*

$$\mathcal{E}(f, f) = -\langle f, \mathscr{L}_{ld} f \rangle_q + \frac{\gamma}{2M} \sum_{i=1}^{M} \sum_{j=1}^{M} \int_{\Omega} \min \left\{ g_{ij}^{\#} q(x), q(x) \right\} \left( f(g_{ij}(x)) - f(x) \right)^2.$$

*Proof.* Using reversibility, we compute

$$\mathcal{E}(f, f) = -\langle f, \mathscr{L}_{leap} f \rangle_q$$

$$= -\langle f, \mathscr{L}_{ld} \rangle_q + \frac{\gamma}{M} \sum_{i=1}^{M} \sum_{j=1}^{M} \int_{\Omega} \min \left\{ \frac{g_{ij}^{\#} q(x)}{q(x)}, 1 \right\} \left( f(g_{ij}(x)) - f(x) \right) f(x) q(x) dx$$

$$= -\langle f, \mathscr{L}_{ld} \rangle_q + \frac{\gamma}{2M} \sum_{i=1}^{M} \sum_{j=1}^{M} \int_{\Omega} \min \left\{ g_{ij}^{\#} q(x), q(x) \right\} \left( f(g_{ij}(x)) - f(x) \right)^2.$$

$\square$

## D.2 SIMULATED TEMPERING TELEPORT PROCESS

We now decompose the simulated tempering version of the Markov process. In this setting, the process is defined to have stationary distribution $p(x,i) = \sum_j r_j p_j(x) I\{i = j\}$ on $\Omega \times [L]$, where $\sum_i r_i = 1$ and $p_j$ can be expressed as the mixture $p_j = \sum_{k=1}^M w_k p_{kj}$. The Markov process moves between temperatures according to the simulated tempering chain, at each temperature running Langevin diffusion till the next jump. At the coldest temperature, as in Section D.1, the Markov process leaps between modes of the distribution corresponding to a Poisson point process. In Section D.1 we found the generator $\mathscr{L}_{leap}$ at the coldest level. By applying $\mathscr{L}_{leap}$ to the simulated tempering results in Ge et al. (2018c) we are able to compute the generator $\mathscr{L}_{Tel}$ for the Simulated Tempering with Teleporting Sampler.

**Definition D.5.** *We define **the continuous Simulated Tempering with Teleporting Markov process** $K_{Tel}$ on $\Omega \times [L]$ with jump rate $\lambda$ (between temperatures) and leap rate $\gamma$ (at coldest temperature between modes) as follows:*

1. *Let $T_n$ be a Poisson point process with rate $\lambda$ so that,*

$$T_n - T_{n-1} \sim Exp(\lambda).$$

2. *If $i \neq 1$, the Markov process with state $(x,i)$ evolves according to $K_i$ on the interval $[T_{n-1}, T_n)$.*

   *At time $T_n$, the Markov process jumps to $(x,i')$ with probability*

$$\frac{1}{2} \min\left\{\frac{r_{i'} p_{i'}(x)}{r_i p_i(x)}, 1\right\}, \quad for\ i' = i \pm 1$$

   *and stays at $(x,i)$ otherwise.*

3. *Let $T'_{n'}$ be a Poisson point process with rate $\gamma$ so that,*

$$T'_{n'} - T'_{n'-1} \sim Exp(\gamma).$$

4. *If $i = 1$, Let $\tilde{T} = \min(T_n, T'_{n'})$, the Markov process with state $(x,1)$ evolves according to $K_1$ on the interval $[T'_{n'-1}, \tilde{T})$.*

   *If $\tilde{T} = T'_{n'}$, the Markov process leaps to $(x,1) \mapsto (g^\#_{jj'}(x), 1)$ with probability*

$$\frac{1}{M} \min\left\{\frac{g^\#_{jj'} p_1(x)}{p_1(x)}, 1\right\}, \quad \forall j' \neq j$$

   *and stays at $(x,1)$ otherwise.*

   *If $\tilde{T} = T_n$, the Markov process jumps to $(x,2)$ with probability*

$$\frac{1}{2} \min\left\{\frac{r_2 p_2(x)}{r_1 p_1(x)}, 1\right\}$$

   *and stays at $(x,1)$ otherwise.*

**Lemma D.6.** *(Lemma 5.1 Ge et al. (2018c)) Let $M_i, i \in [L]$ be a sequence of continuous Markov proceses with state space $\Omega$, generators $\mathscr{L}_i$, and unique stationary distributions $p_i$. Then the continuous simulated tempering Markov process $M_{st}$ with rate $\lambda$ and relative probabilities $r_i$ has generator $\mathscr{L}_{st}$ defined by the following equation, where $f = (f_1, \ldots, f_L) \in \prod_{i=1}^L \mathcal{D}(\mathscr{L}_i)$:*

$$(\mathscr{L}f)(i,y) = (\mathscr{L}_i f_i)(y) + \frac{\lambda}{2}\left(\sum_{1 \leq j \leq L, j=i\pm 1} \min\left\{\frac{r_j p_j(x)}{r_i p_i(x)}, 1\right\}(f_j(x) - f_i(x))\right).$$

*The corresponding Dirichlet form is given by,*

$$\mathscr{E}(f,f) = -\sum_{i=1}^{L} r_i \langle f_i, \mathscr{L}_i f_i \rangle_{p_i} + \frac{\lambda}{4} \left( \sum_{1 \le i \le L, j=i\pm 1} \int_\Omega \min\left\{ r_j p_j(x), r_i p_i(x) \right\} (f_j(x) - f_i(x))^2 \right) dx.$$

In Ge et al. (2018c), the authors determine the Dirichlet form of the generator $\mathscr{L}$ for the simulated tempering Markov process. In their setting, $\mathscr{L}_i$ for all $1 \le i \le L$ is the Langevin diffusion generator. In our setting, the generator for the teleport sampler, $\mathscr{L}_{leap}$ in Lemma D.3, takes the place of the generator at the coldest temperature $\mathscr{L}_1$. We will maintain the notation of $\mathscr{L}_i$, $1 \le i \le L$ as the Langevin diffusion generator and replace $\mathscr{L}_{leap}$ with $\mathscr{L}_1$ in the previous Lemma.

**Corollary D.7.** *Let all assumptions and notation hold from Lemma D.6, then the Dirichlet form for the continuous time annealed leap-point Markov process $\mathscr{L}_{Tel}$ is given by*

$$\mathscr{E}(f,f) = \sum_{i=1}^{L} r_i \mathscr{E}_i(f_i, f_i) + \frac{\gamma \cdot r_1}{2M} \sum_{j=1}^{M} \sum_{i=1}^{M} \int_\Omega \min\left\{ g_{ij}^{\#} p_1(x), p_1(x) \right\} \left( f_1(g_{ij}(x)) - f_1(x) \right)^2 dx$$

$$+ \frac{\lambda}{4} \sum_{i \le i \le L, j=i\pm 1} \int_\Omega \min\left\{ r_j p_j(x), r_i p_i(x) \right\} \left( f_j(x) - f_i(x) \right)^2 dx$$

*Proof.*

$$\mathscr{E}(f,f) = -\langle f_1, \mathscr{L}_{leap} f_1 \rangle_{p_1} - \sum_{i=2}^{L} r_i \langle f_i, \mathscr{L}_i f_i \rangle_{p_i} + \frac{\lambda}{4} \left( \sum_{1 \le i \le L, j=i\pm 1} \int_\Omega \min\left\{ r_j p_j(x), r_i p_i(x) \right\} (f_j(x) - f_i(x))^2 \right) dx$$

by Corollary D.4

$$= -r_1 \langle f_1, \mathscr{L}_1 f_1 \rangle_{p_1} + \frac{\gamma \cdot r_1}{2M} \sum_{j=1}^{M} \sum_{i=1}^{M} \int_\Omega \min\left\{ g_{ij}^{\#} p_1(x), p_1(x) \right\} \left( f_1(g_{ij}(x)) - f_1(x) \right)^2 dx$$

$$- \sum_{i=2}^{L} r_i \langle f_i, \mathscr{L}_i f_i \rangle_{p_i} + \frac{\lambda}{4} \sum_{i,j \le L, j=i\pm 1} \int_\Omega \min\left\{ r_j p_j(x), r_i p_i(x) \right\} \left( f_j(x) - f_i(x) \right)^2 dx$$

$$= -\sum_{i=1}^{L} r_i \langle f_i, \mathscr{L}_i f_i \rangle_{p_i} + \frac{\gamma \cdot r_1}{2M} \sum_{j=1}^{M} \sum_{i=1}^{M} \int_\Omega \min\left\{ g_{ij}^{\#} p_1(x), p_1(x) \right\} \left( f_1(g_{ij}(x)) - f_1(x) \right)^2 dx$$

$$+ \frac{\lambda}{4} \sum_{i,j \le L, j=i\pm 1} \int_\Omega \min\left\{ r_j p_j(x), r_i p_i(x) \right\} \left( f_j(x) - f_i(x) \right)^2 dx$$

$\square$

**Lemma D.8.** *Let $K_{Tel}$ be the annealed leap-point Markov process with generator $\mathscr{L}_{ALPS}$ on $\Omega \times [L]$ and stationary distribution $p(x,i) = \sum_j r_j p_j(x) I\{i = j\}$. We also make the following assumptions,*

1. *Each $p_i(x) = \alpha_i \pi_{0,i} + (1 - \alpha_i) \pi_{1,i}$ where each $\pi_{j,i} = \sum_k w_{ji}^{(k)} \pi_{j,i}^{(k)}$.*

2. *For each Markov process $M_i$ there exists a decomposition*

   $$\langle f_i, \mathscr{L}_i f \rangle_{p_i} \le \sum_k w_{ik} \langle f_i, \mathscr{L}_{ik} f_i \rangle_{p_{ik}},$$

   *where $\mathscr{L}_{ik}$ is the generator of some Markov process $M_{ik}$ with stationary distribution $p_{ik}(x)$.*

*Then for some weight $\alpha$ the following decomposition holds*

$$\langle f, \mathscr{L}_{Tel} f \rangle_p \le \alpha \langle f, \mathscr{L}_{Tel,0} f \rangle_{\pi_0} + (1 - \alpha) \langle f, \mathscr{L}_{Tel,1} f \rangle_{\pi_1},$$

*where $\mathscr{L}_{ALPS,k}$ is the continuous time annealed leap-point process with stationary distribution $\pi_k \propto \sum_i \alpha_i r_i \pi_{ki}$.*

*Proof.* We consider the following,

$$\langle f, \mathscr{L}_{tel} \rangle_p = \underbrace{\sum_{i=1}^{L} r_i \langle f_i, \mathscr{L}_i f_i \rangle_{p_i}}_{A} - \underbrace{\frac{\gamma \cdot r_1}{2M} \sum_{j=1}^{M} \sum_{i=1}^{M} \int_{\Omega} \min \left\{ g_{ij}^{\#} p_1(x), p_1(x) \right\} \left( f_1(g_{ij}(x)) - f_1(x) \right)^2 dx}_{B}$$

$$\underbrace{- \frac{\lambda}{4} \sum_{i \leq i \leq L, j = i \pm 1} \int_{\Omega} \min \left\{ r_j p_j(x), r_i p_i(x) \right\} \left( f_j(x) - f_i(x) \right)^2 dx}_{C}$$

We proceed by finding an upper bound on each part, starting with $A$.

$$A = \sum_{i=1}^{L} r_i \langle f_i, \mathscr{L}_i f_i \rangle_{p_i}$$

By assumption 2 we can decompose the generator $\mathscr{L}_i$,

$$\leq \sum_{i=1}^{L} r_i \big( \alpha_i \langle f_i, \mathscr{L}_{i0} f_i \rangle_{\pi_0} + (1 - \alpha_i) \langle f_i, \mathscr{L}_{i1} f_i \rangle_{\pi_1} \big).$$

To find an upper bound on **B** it is worth noting by assumption 1 we have that

$$p_1(x) = \alpha_1 p_{1,good} + (1 - \alpha_1) p_{1,bad}.$$

Which by change of notation we let $p_{1,good} = \pi_{1,0}$ and $p_{1,bad} = \pi_{1,1}$.

$$-B = -\frac{\gamma \cdot r_1}{2M} \sum_{j=1}^{M} \sum_{i=1}^{M} \int_{\Omega} \min \left\{ g_{ij}^{\#} p_1(x), p_1(x) \right\} \left( f_1(g_{ij}(x)) - f_1(x) \right)^2 dx$$

$$= -\frac{\gamma \cdot r_1}{2M} \sum_{j=1}^{M} \sum_{i=1}^{M} \int_{\Omega} \min \left\{ g_{ij}^{\#} (\alpha_1 \pi_{1,0} + (1 - \alpha_1) \pi_{1,1}), (\alpha_1 \pi_{1,0} + (1 - \alpha_1) \pi_{1,1}) \right\} \left( f_1(g_{ij}(x)) - f_1(x) \right)^2 dx$$

$$\leq -\frac{\gamma \cdot r_1}{2M} \sum_{j=1}^{M} \sum_{i=1}^{M} \int_{\Omega} \min \left\{ \alpha_1 g_{ij}^{\#} \pi_{1,0}, \alpha_1 \pi_{1,0} \right\} \left( f_1(g_{ij}(x)) - f_1(x) \right)^2 dx$$

$$- \frac{\gamma \cdot r_1}{2M} \sum_{j=1}^{M} \sum_{i=1}^{M} \int_{\Omega} \min \left\{ (1 - \alpha_1) g_{ij}^{\#} \pi_{1,1}, (1 - \alpha_1) \pi_{1,1} \right\} \left( f_1(g_{ij}(x)) - f_1(x) \right)^2 dx.$$

Lastly we have that

$$-C = -\frac{\lambda}{4} \sum_{i \leq i \leq L, j = i \pm 1} \int_{\Omega} \min \left\{ r_j p_j(x), r_i p_i(x) \right\} \left( f_j(x) - f_i(x) \right)^2 dx$$

$$= -\frac{\lambda}{4} \sum_{i \leq i \leq L, j = i \pm 1} \int_{\Omega} \min \left\{ r_j (\alpha_j \pi_{0,j} + (1 - \alpha_j) \pi_{1,j}), r_i (\alpha_i \pi_{0,i} + (1 - \alpha_i) \pi_{1,i}) \right\} \left( f_j(x) - f_i(x) \right)^2 dx$$

$$\leq -\frac{\lambda}{4} \sum_{i \leq i \leq L, j = i \pm 1} \int_{\Omega} \min \left\{ r_j \alpha_j \pi_{0,j}, r_i \alpha_i \pi_{0,i} \right\} \left( f_j(x) - f_i(x) \right)^2 dx$$

$$- \frac{\lambda}{4} \sum_{i \leq i \leq L, j = i \pm 1} \int_{\Omega} \min \left\{ r_j (1 - \alpha_j) \pi_{1,j}, r_i (1 - \alpha_i) \pi_{1,i} \right\} \left( f_j(x) - f_i(x) \right)^2 dx.$$

By our bounds on A, B and C and choosing normalizing constant $\alpha = \sum_i r_i \alpha_i$ we can express

$$\langle f, \mathscr{L}_{Tel} f \rangle_p \leq \alpha \langle f, \mathscr{L}_{Tel,0} f \rangle_{\pi_0} + (1 - \alpha) \langle f, \mathscr{L}_{Tel,1} f \rangle_{\pi_1}.$$

$\square$

# E  MARKOV PROCESS DECOMPOSITION

In this section we bound the Poincaré constant corresponding to the continuous time Markov process defined in Section D. The analysis in this section is similar to that of the analysis found in (Ge et al., 2018c), where the Poincaré constant corresponding to the whole chain is bounded by the Poincaré constants corresponding to local components and the Poincaré constant corresponding to the projected chain. This decomposition reduces the mixing time analysis to finding a bound on the Poincaré constant of the projected.

**Theorem E.1.** *Let $K_{Tel}$ be the Markov process in definition D.5 with stationary distribution $p(x,k) = \sum_{i=1}^{L} r_i \sum_{j=1}^{M} w_{ij} P_{ij}(x) I\{k = i\}$. Let $K_i$ $1 \leq i \leq L$ be the Markov process on $\Omega$ with generator $\mathscr{L}_i$ with stationary distribution $P_i(x) = \sum_{j=1}^{M} w_{ij} P_{ij}(x)$. More specifically, $K_1 = K_{leap}$ as in definition D.2 with generator $\mathscr{L}_1 = \mathscr{L}_{leap}$ (Lemma D.3). The function $f = (f_1, \ldots, f_L) \in [L] \times \Omega$ and the Dirichlet form is $\mathscr{E}_p(f,f) = \langle f, \mathscr{L}_{ALPS}f \rangle_P$. Assume the following hold.*

1. *There exists a decomposition*

$$\langle f, \mathscr{L}_i f \rangle_{P_i} \leq \sum_{j=1}^{M} \langle f, \mathscr{L}_{ij} f \rangle_{P_{ij}}$$

   *where $\mathscr{L}_{ij}$ is the generator of some Markov process with $K_{ij}$ with stationary distribution $P_{ij}$.*

2. *Each distribution $P_{ik}$ satisfies a Poincaré inequality*

$$Var_{P_{ij}}(f) \leq C\mathscr{E}_{ij}(f,f).$$

3. *We define the projected chain as*

$$\bar{T}\big((i,j),(i',j')\big) = \begin{cases} \int_\Omega \min\left\{ \frac{w_{1j'} \cdot g_{jj'}^\# \cdot P_{1j'}(x)}{w_{1j} \cdot P_{1j}(x)}, 1 \right\} P_{1j}(x)dx, & i = i' = 1 \\ \int_\Omega \min\left\{ \frac{r'_i w_{i'j} \cdot P_{i'j}(x)}{r_i w_{ij} \cdot P_{ij}(x)}, 1 \right\} P_{ij}(x)dx, & i' = i \pm 1 \text{ and } j = j' \\ 0 & \text{otherwise} \end{cases}$$

   (E.1)

   *Let $\bar{P}(\{i,j\}) = r_i w_{ij}$ be the stationary distribution of $\bar{T}$. Where $\bar{T}$ satisfies the Poincaré inequality*

$$Var_{\bar{P}}(\bar{f}) \leq \bar{C} \cdot \mathscr{E}_{\bar{P}}(\bar{f}, \bar{f})$$

   *with $\bar{f}(\{i,j\}) = \mathbb{E}_{P_{ij}}(f_i)$. Then $K_{ALPS}$ satisfies the Poincaré inequality*

$$Var_P(f) \leq \max\left\{ C(1 + (6M + 12)\bar{C}, \frac{6M\bar{C}}{\gamma}, \frac{12\bar{C}}{\lambda} \right\} \mathscr{E}(f,f)$$

.

*Proof.* We begin by considering the following,

$$\begin{aligned}
\mathrm{Var}_P(f) &= \sum_{i=1}^{L} \sum_{j=1}^{M} r_i w_{ij} \int_\Omega \left( f_i - \mathbb{E}_P(f) \right)^2 P_{ij}(dx) \\
&= \sum_{i=1}^{L} \sum_{j=1}^{M} r_i w_{ij} \int_\Omega \left( f_i - \mathbb{E}_{P_{ij}}(f_i) + \mathbb{E}_{P_{ij}}(f_i) - \mathbb{E}_P(f) \right)^2 P_{ij}(dx) \\
&= \sum_{i=1}^{L} \sum_{j=1}^{M} r_i w_{ij} \int_\Omega \left( f_i - \mathbb{E}_{P_{ij}}(f_i) \right)^2 P_{ij}(dx) + \sum_{i=1}^{L} \sum_{j=1}^{M} r_i w_{ij} \left( \mathbb{E}_{P_{ij}}(f_i) - \mathbb{E}_P(f) \right)^2 \\
&= \sum_{i=1}^{L} \sum_{j=1}^{M} r_i w_{ij} \mathrm{Var}_{P_{ij}}(f_i) + \mathrm{Var}_{\bar{P}}(\bar{g})
\end{aligned}$$

$$\leq C \sum_{i=1}^{L} \sum_{j=1}^{M} r_i w_{ij} \mathscr{E}_{ij}(f_i, f_i) + \bar{C} \mathscr{E}_{\bar{P}}(\bar{f}, \bar{f})$$

$$\leq C \sum_{i=1}^{L} r_i \mathscr{E}_i(f_i, f_i) + \bar{C} \mathscr{E}_{\bar{P}}(\bar{f}, \bar{f})$$

We now decompose the Dirichlet form $\mathscr{E}_{\bar{P}}(\bar{f}, \bar{f})$ per Lemma J.1.

$$\mathscr{E}_{\bar{P}}(\bar{f}, \bar{f}) = \sum_{(i,j) \in [L] \times [M]} \sum_{(i',j') \in [L] \times [M]} \left( \mathbb{E}_{P_{i'j'}}(f_{i'}) - \mathbb{E}_{P_{ij}}(f_i) \right)^2 \bar{P}(\{i,j\}) \bar{T}(\{i,j\}, \{i',j'\})$$

By applying the definition of $\bar{T}$

$$= \underbrace{r_1 \sum_{j=1}^{M} \sum_{j'=1}^{M} w_{1j} \left( \mathbb{E}_{P_{1j'}}(f_1) - \mathbb{E}_{P_{1j}}(f_1) \right)^2 \int_{\Omega} \min \left\{ \frac{w_{1j'} \cdot g_{jj'}^{\#} P_{1j'}(x)}{w_{1j} \cdot P_{1j}(x)}, 1 \right\} P_{1j}(x) dx}_{A}$$

$$+ \underbrace{\sum_{j=1}^{M} \sum_{\substack{1 \leq i \leq L \\ i'=i\pm 1}} r_i w_{ij} \left( \mathbb{E}_{P_{i'j}}(f_{i'}) - \mathbb{E}_{P_{ij}}(f_i) \right)^2 \int_{\Omega} \min \left\{ \frac{r'_i w_{i'j} \cdot P_{i'j}(x)}{r_i w_{ij} \cdot P_{ij}(x)}, 1 \right\} P_{ij}(x) dx}_{B}$$

To simplify the above expressions we let $\delta_{(1,j),(1,j')}^{g} = \int_{\Omega} \min \left\{ w_{1j'} \cdot g_{jj'}^{\#} P_{1j'}(x), w_{1j} \cdot P_{1j}(x) \right\} dx$ and let $Q_{(1,j),(1,j')}^{g}(x) = \frac{1}{\delta_{(1,j),(1,j')}^{g}} \min \left\{ w_{1j'} \cdot g_{jj'}^{\#} P_{1j'}(x), w_{1j} \cdot P_{1j}(x) \right\}$ be the normalized distribution. Then we consider the following,

$$A = r_1 \sum_{j=1}^{M} \sum_{j'=1}^{M} w_{1j} \left( \mathbb{E}_{P_{1j'}}(f_1) - \mathbb{E}_{P_{1j}}(f_1) \right)^2 \int_{\Omega} \min \left\{ \frac{w_{1j'} \cdot g_{jj'}^{\#} P_{1j'}(x)}{w_{1j} \cdot P_{1j}(x)}, 1 \right\} P_{1j}(x) dx$$

$$= r_1 \sum_{j=1}^{M} \sum_{j'=1}^{M} \left( \mathbb{E}_{P_{1j'}}(f_1) - \mathbb{E}_{P_{1j}}(f_1) \right)^2 \int_{\Omega} \min \left\{ w_{1j'} \cdot g_{jj'}^{\#} P_{1j'}(x), w_{1j} P_{1j}(x) \right\} dx$$

by a change of measure on the first term,

$$= r_1 \sum_{j=1}^{M} \sum_{j'=1}^{M} \left( \int_{\Omega} f_1 \circ g_{jj'}(x) g_{jj'}^{\#} P_{1j'}(dx) - \int_{\Omega} f_1(x) P_{1j}(dx) \right)^2 \delta_{(1,j),(1,j')}^{g}$$

$$= 3 r_1 \sum_{j=1}^{M} \sum_{j'=1}^{M} \left[ \left( \int_{\Omega} f_1 \circ g_{jj'}(x) \left( Q_{(1,j),(1,j')}^{g}(dx) - g_{jj'}^{\#} P_{1j'}(dx) \right) \right)^2 \right.$$

$$+ \left( \int_{\Omega} (f_1 \circ g_{jj'}(x) - f_1(x)) Q_{(1,j),(1,j')}^{g}(dx) \right)^2 + \left. \left( \int_{\Omega} f_1(x) \left( P_{1j}(dx) - Q_{(1,j),(1,j')}^{g}(dx) \right) \right)^2 \right] \delta_{(1,j),(1,j')}^{g}$$

By Lemma J.2

$$\leq 3 r_1 \sum_{j=1}^{M} \sum_{j'=1}^{M} \left[ \mathrm{Var}_{g_{jj'}^{\#} P_{1j'}}(f_1 \circ g_{jj'}) \chi^2 (Q_{(1,j),(1,j')}^{g} || g_{jj'}^{\#} P_{1j'}) + \mathrm{Var}_{P_{1j}}(f_1) \chi^2 (Q_{(1,j),(1,j')}^{g} || P_{1j}) \right.$$

$$+ \left. \int_{\Omega} \left( f_1 \circ g_{jj'}(x) - f_1(x) \right)^2 Q_{(1,j),(1,j')}^{g}(dx) \right] \delta_{(1,j),(1,j')}^{g}$$

By applying the definition of $Q^g_{(1,j),(1,j')}$,

$$\leq 3r_1 \sum_{j=1}^{M} \sum_{j'=1}^{M} \left[ \text{Var}_{g^{\#}_{jj'},P_{1j'}}(f_1 \circ g_{jj'})\chi^2(Q^g_{(1,j),(1,j')}||g^{\#}_{jj'}P_{1j'}) + \text{Var}_{P_{1j}}(f_1)\chi^2(Q^g_{(1,j),(1,j')}||P_{1j}) \right] \delta^g_{(1,j),(1,j')}$$

$$+ 3r_1 \sum_{j=1}^{M} \sum_{j'=1}^{M} \int_{\Omega} \left( f_1 \circ g_{jj'}(x) - f_1(x) \right)^2 \min\left\{ w_{1j'} \cdot g^{\#}_{jj'}P_{1j'}(x), w_{1j} \cdot P_{1j}(x) \right\} dx$$

By Lemma G3 (Ge et al., 2018c)

$$\leq 3r_1 \sum_{j=1}^{M} \sum_{j'=1}^{M} \left[ w_{1j'} \text{Var}_{g^{\#}_{jj'},P_{1j'}}(f_1 \circ g_{jj'}) + w_{1j} \text{Var}_{P_{1j}}(f_1) \right]$$

$$+ 3r_1 \sum_{j=1}^{M} \sum_{j'=1}^{M} \int_{\Omega} \left( f_1 \circ g_{jj'}(x) - f_1(x) \right)^2 \min\left\{ w_{1j'} \cdot g^{\#}_{jj'}P_{1j'}(x), w_{1j} \cdot P_{1j}(x) \right\} dx$$

$$= 3r_1 \sum_{j=1}^{M} \sum_{j'=1}^{M} \left[ w_{1j'} \text{Var}_{P_{1j'}}(f_1) + w_{1j} \text{Var}_{P_{1j}}(f_1) \right]$$

$$+ 3r_1 \sum_{j=1}^{M} \sum_{j'=1}^{M} \int_{\Omega} \left( f_1 \circ g_{jj'}(x) - f_1(x) \right)^2 \min\left\{ w_{1j'} \cdot g^{\#}_{jj'}P_{1j'}(x), w_{1j} \cdot P_{1j}(x) \right\} dx$$

$$\leq 6r_1 M \cdot C\mathcal{E}_1(f_1, f_1) + 3r_1 \sum_{j=1}^{M} \sum_{j'=1}^{M} \int_{\Omega} \left( f_1 \circ g_{jj'}(x) - f_1(x) \right)^2 \min\left\{ w_{1j'} \cdot g^{\#}_{jj'}P_{1j'}(x), w_{1j} \cdot P_{1j}(x) \right\} dx$$

$$\leq 6r_1 M \cdot C\mathcal{E}_1(f_1, f_1) + 3r_1 \sum_{j=1}^{M} \sum_{j'=1}^{M} \int_{\Omega} \left( f_1 \circ g_{jj'}(x) - f_1(x) \right)^2 \min\left\{ P_1 \circ g_{jj'}(x), P_1(x) \right\} dx$$

The bound for (B) should mimic the bound for (B) in Theorem 6.3 of (Ge et al., 2018c). The proof is included for sake of completeness. Denote by $\delta_{(i,j),(i',j')} = \int_{\Omega} \min\left\{ \frac{r'_i w_{i'j'}}{r_i w_{ij}} \cdot P_{i'j'}(x), P_{ij}(x) \right\} dx$ and $Q_{(i,j),(i',j')}(x) = \frac{1}{\delta_{(i,j),(i',j')}} \min\left\{ \frac{r'_i w_{i'j'}}{r_i w_{ij}} \cdot P_{i'j'}(x), P_{ij}(x) \right\}$ be the normalized distribution. Then by applying Lemma 6.4 in (Ge et al., 2018c) yields,

$$B = \sum_{j=1}^{M} \sum_{\substack{1 \leq i \leq L \\ i' = i \pm 1}} r_i w_{ij} \left( \mathbb{E}_{P_{i'j}}(f_{i'}) - \mathbb{E}_{P_{ij}}(f_i) \right)^2 \int_{\Omega} \min\left\{ \frac{r'_i w_{i'j} \cdot P_{i'j}(x)}{r_i w_{ij} \cdot P_{ij}(x)}, 1 \right\} P_{ij}(x) dx$$

$$\leq 3 \sum_{j=1}^{M} \sum_{\substack{1 \leq i \leq L \\ i' = i \pm 1}} \left[ \text{Var}_{P_{ij}}(f_i)\chi^2(Q_{(i,j),(i',j)}||P_{ij}) + \text{Var}_{P_{i'j}}(f_{i'})\chi^2(Q_{(i,j),(i',j)}||P_{i'j}) \right.$$

$$\left. + \int_{\Omega} (f_i - f_{i'})^2 Q_{(i,j),(i',j)}(dx) \right] \cdot r_i w_{ij} \delta_{(i,j),(i',j)}.$$

By Lemma G3 (Ge et al., 2018c)

$$, \leq 3 \sum_{j=1}^{M} \sum_{\substack{1 \leq i \leq L \\ i' = i \pm 1}} \text{Var}_{P_{ij}}(f_i) \frac{r_i w_{ij} \delta_{(i,j),(i',j)}}{\delta_{(i,j),(i',j)}} + \text{Var}_{P_{i'j}}(f_{i'}) \frac{r_i w_{ij} \delta_{(i,j),(i',j)}}{\delta_{(i',j),(i,j)}}$$

$$+ 3 \int_{\Omega} (f_i - f_{i'})^2 \min\left\{ r_{i'} w_{i'j} \cdot P_{i'j}(x), r_i w_{ij} P_{ij}(x) \right\} dx$$

$$\leq 3 \sum_{j=1}^{M} \sum_{\substack{1 \leq i \leq L \\ i' = i \pm 1}} r_i w_{ij} \text{Var}_{P_{ij}}(f_i) + r_{i'} w_{i'j} \text{Var}_{P_{i'j}}(f_{i'})$$

$$+ 3 \sum_{\substack{1 \leq i \leq L \\ i'=i\pm1}} \int_{\Omega} (f_i - f_{i'})^2 \min \left\{ r_{i'} \sum_{j=1}^{M} w_{i'j} \cdot P_{i'j}(x), r_i \sum_{j=1}^{M} w_{ij} P_{ij}(x) \right\} dx$$

$$\leq 12C \sum_{j=1}^{M} \sum_{\substack{1 \leq i \leq L \\ i'=i\pm1}} r_i \mathscr{E}_i(f_i, f_i) + 3 \sum_{\substack{1 \leq i \leq L \\ i'=i\pm1}} \int_{\Omega} (f_i - f_{i'})^2 \min \left\{ r_{i'} P_{i'}(x), r_i P_i(x) \right\} dx$$

Combining terms $A$ and $B$ with the bound on the intra-mode variance we have that,

$$Var_P(f) \leq C \sum_{i=1}^{L} r_i \mathscr{E}_i(f_i, f_i)$$

$$+ \bar{C} \left[ 6 r_1 M \cdot C \mathscr{E}_1(f_1, f_1) + 3 r_1 \sum_{j=1}^{M} \sum_{j'=1}^{M} \int_{\Omega} \left( f_1 \circ g_{jj'}(x) - f_1(x) \right)^2 \min \left\{ P_1 \circ g_{jj'}(x), P_1(x) \right\} dx \right.$$

$$+ 12C \sum_{i=1}^{L} r_i \mathscr{E}_i(f_i, f_i) + 3 \sum_{\substack{1 \leq i \leq L \\ i'=i\pm1}} r_i w_{ij} \left( \mathbb{E}_{P_{i'j}}(f_{i'}) - \mathbb{E}_{P_{ij}}(f_i) \right)^2 \int_{\Omega} \min \left\{ r_i' P_{i'}(x), r_i P_i(x) \right\} dx \left. \right]$$

Grouping like terms and comparing to the Dirichlet form in Corollary D.7

$$\leq C(1 + \bar{C}(6M + 12)) \sum_{i=1}^{L} r_i \mathscr{E}_i(f_i, f_i)$$

$$+ \frac{6\bar{C}M}{\gamma} \frac{\gamma \cdot r_1}{2M} \sum_{j=1}^{M} \sum_{j'=1}^{M} \int_{\Omega} \min \left\{ P_1 \circ g_{jj'}(x), P_1(x) \right\} \left( f_1 \circ g_{jj'}(x) - f_1(x) \right)^2 dx$$

$$+ \frac{12\bar{C}}{\lambda} \frac{\lambda}{4} \sum_{\substack{1 \leq i \leq L \\ i'=i\pm1}} r_i w_{ij} \left( f_{i'}(x) - f_i(x) \right)^2 \int_{\Omega} \min \left\{ r_i' P_{i'}(x), r_i P_i(x) \right\} dx \left. \right]$$

By applying the dirichlet form in Corollary D.7

$$\leq \max \left\{ C(1 + (6M + 12)\bar{C}, \frac{6M\bar{C}}{\gamma}, \frac{12\bar{C}}{\lambda} \right\} \mathscr{E}(f, f)$$

$$\square$$

## F  LOCAL CONVERGENCE FOR A MARKOV PROCESS

In this section we show that for a Markov chain $P_t$, with stationary mixture distribution $\pi = \sum_k w_k \pi_k$, the weight adjusted distribution $p_{T,0} = p_T \frac{\pi_0}{\pi} / \int_{\Omega} p_T \frac{\pi_0}{\pi}$ converges to the component $\pi_0$ in chi-squared divergence, where $p_T = \nu_0 P_t$ for some initial probability measure $\nu_0$. This can be applied to the mixture measure $\tilde{p}_{\beta}(x) \propto \pi(x) \cdot \sum_{k=1}^{M} w_{\beta,k} q_{\beta}(x - x_k)$, on the extended state space $\Omega \times [L]$. Since on the target level $\beta_L = 0$, there is no bad component, the divergence $\chi^2(p_{T,good} || p_{good})$ provides us with a good indication of how close we are at the target level.

The key challenge in our analysis arising from our tempering schedule, $\tilde{p}_{\beta}(x) \propto \pi(x) \cdot \sum_{k=1}^{M} w_{\beta,k} q_{\beta}(x - x_k)$, is the creation of unwanted cross-terms, which can manifest as pseudo-modes in the state space. As illustrated in Figure 4, the stationary distribution at any given temperature is a mix of good components, where the target modes are correctly aligned with the tilting functions (solid lines), and these bad components (dashed lines). The analysis in this section provides the theoretical groundwork to formally disregard these bad components, effectively proving that the time spent exploring the pseudo-modes is negligible. This allows us to analyze the mixing time of the overall process by focusing only on the simpler dynamics within and between the well-defined good components.

Figure 4: An illustration of good (solid) and bad (dashed) mixture components. The good components are sharply peaked around the warm-start locations, while the bad pseudo-modes from cross-terms have negligible mass.

The following Lemma provides us with an upper bound on $\chi^2(p_{T,good}\|p_{good})$ in terms of the Poincaré constant on $p_{good}$.

**Lemma F.1.** *Suppose that for a Markov generator $\mathscr{L}$, with stationary distribution $\pi = \sum_{k=0}^{\ell} \alpha_k \pi_k$, $\langle f, \mathscr{L} f \rangle_\pi \leq \sum_k \alpha_k \langle f, \mathscr{L}_k f \rangle_{\pi_0}$ for all $f$, where $\mathscr{L}_0$ has stationary measure $\pi_0$ and Poincaré constant $C$. Let $\bar{p}_T$ be the distribution of $X_t$ where $t \sim \mathsf{Unif}(0,T)$, $K_0 = \chi^2(p_0\|\pi)$, and $\bar{p}_{T,0} = \bar{p}_T \frac{\pi_0}{\pi} / \int_\Omega \bar{p}_T \frac{\pi_0}{\pi}$. Then*

$$\mathrm{Var}_{\pi_0}\left(\frac{\bar{p}_T}{\pi}\right) \leq \frac{K_0 C}{\alpha_0 T}$$

*or equivalently,*

$$\chi^2(\bar{p}_{T,0}\|\pi_0) \leq \frac{K_0 C}{\alpha_0 \left[\mathbb{E}_{\bar{p}_T}\left(\frac{\pi_0}{\pi}\right)\right]^2 T}.$$

*Proof.* We have that

$$\frac{d}{dt}\chi^2(p_t\|\pi) = -\left\langle \frac{p_t}{\pi}, \mathscr{L}\frac{p_t}{\pi} \right\rangle,$$

so

$$K_0 = \chi^2(p_0\|\pi) \geq \int_0^T \left\langle \frac{p_s}{\pi}, \mathscr{L}\frac{p_s}{\pi} \right\rangle ds$$

$$\geq \int_0^T \sum_{k=0}^{\ell} \alpha_k \left\langle \frac{p_s}{\pi}, \mathscr{L}_k\frac{p_s}{\pi} \right\rangle ds$$

$$\geq \frac{1}{C} \int_0^T \sum_{k=0}^{\ell} \alpha_k \mathrm{Var}_{\pi_k}\left(\frac{p_s}{\pi}\right) ds$$

$$\geq \frac{\alpha_0}{C} \int_0^T \mathrm{Var}_{\pi_0}\left(\frac{p_s}{\pi}\right) ds$$

$$\geq \frac{\alpha_0 T}{C} \mathbb{E}_{s\sim\mathsf{Unif}(0,T)} \mathrm{Var}_{\pi_0}\left(\frac{p_s}{\pi}\right)$$

$$\geq \frac{\alpha_0 T}{C} \mathrm{Var}_{\pi_0}\left(\frac{\bar{p}_T}{\pi}\right). \tag{F.1}$$

Now

$$\mathrm{Var}_{\pi_0}\left(\frac{\bar{p}_T}{\pi}\right) = \mathrm{Var}_{\pi_0}\left(\frac{\bar{p}_T \frac{\pi_0}{\pi}}{\pi_0}\right) = \left[\mathbb{E}_{\bar{p}_T}\left(\frac{\pi_0}{\pi}\right)\right]^2 \mathrm{Var}_{\pi_0}\left(\frac{\bar{p}_{T,0}}{\pi_0}\right) = \left[\mathbb{E}_{\bar{p}_T}\left(\frac{\pi_0}{\pi}\right)\right]^2 \chi^2(\bar{p}_{T,0}\|\pi_0).$$
$$\tag{F.2}$$

$\square$

**Lemma F.2.** *Consider an ergodic Markov process on $\Omega$ with stationary distribution $\pi$. Suppose $\pi = \alpha_0 \pi_0 + (1 - \alpha_0)\pi_1$ for measures $\pi_0$, $\pi_1$. For any measure $\nu_0$ on $\Omega$,*

$$\mathbb{E}_{x_0 \sim \nu_0, t \sim \mathsf{Unif}(0,T)} \left[ \alpha_0 \frac{d\pi_0}{d\pi}(x_t) \right] \geq \frac{\alpha_0}{2 \left\| \frac{d\nu_0}{d\pi_0} \right\|_{L^\infty}}$$

$$\mathbb{E}_{x_0 \sim \nu_0, t \sim \mathsf{Unif}(0,T)} \left[ \alpha_0 \frac{d\pi_0}{d\pi}(x_t) \right] \geq \frac{\alpha_0}{12(\chi^2(\nu_0 \| \pi_0) + 1)}.$$

For example, if $\pi_0 = \pi|_A$, then $\alpha_0 = \pi(A)$ and

$$\mathbb{E}_{x_0 \sim \nu_0, t \sim \mathsf{Unif}(0,T)} \left[ \alpha_0 \frac{d\pi_0}{d\pi}(x_t) \right] = \mathbb{P}_{x_0 \sim \nu_0, t \sim \mathsf{Unif}(0,T)}(x_t \in A).$$

*Proof.* For a trajectory $x : \mathbb{R}_{\geq 0} \to \Omega$ of the Markov process, define

$$F_x(t) = \int_0^t \alpha_0 \frac{d\pi_0}{d\pi}(x_s) \, ds$$

(which we can interpret as the proportion of time it is in the component $\pi_0$). Note that this is a continuous, differentiable, non-decreasing function. We will write $F^{-1}(r)$ to mean $\min F^{-1}(\{r\})$. Define the random variables

$$T_r := F_x^{-1}(r) = \min \left\{ u : \int_0^u \alpha_0 \frac{d\pi_0}{d\pi}(x_s) \, ds \geq r \right\}$$

$$T_{t,r} := F_x^{-1}(F_x(t) + r) - t = \min \left\{ u : \int_t^{t+u} \alpha_0 \frac{d\pi_0}{d\pi}(x_s) \, ds \geq r \right\}.$$

Then

$$\mathbb{E}_{x_0 \sim \nu_0, t \sim \mathsf{Unif}(0,T)} \left[ \alpha_0 \frac{d\pi_0}{d\pi}(x_t) \right] = \frac{1}{T} \int_0^T \mathbb{E}_{x_0 \sim \nu_0} \alpha_0 \frac{d\pi_0}{d\pi}(x_t) \, dt$$

$$= \frac{1}{T} \int_0^T \mathbb{P}_{x_0 \sim \nu_0} \left( \int_0^T \alpha_0 \frac{d\pi_0}{d\pi}(x_t) \geq r \right) \, dr$$

$$= \frac{1}{T} \int_0^T \mathbb{P}_{x_0 \sim \nu_0}(T_r \leq T) \, dr$$

$$= \frac{1}{T} \int_0^T (1 - \mathbb{P}_{x_0 \sim \nu_0}(T_r > T)) \, dr$$

$$\geq \frac{1}{T} \int_0^T \left( 1 - \frac{\mathbb{E}_{x_0 \sim \nu_0} T_r}{T} \right) \vee 0 \, dr$$

where the last inequality follows from Markov's inequality. We now calculate $\mathbb{E}_{x_0 \sim \pi_0} T_r$ by a counting-in-two-ways argument; then we will use a change-of-measure inequality. Note that for $x_0 \sim \pi$, $x_t$ also has distribution $\pi$, so

$$\mathbb{E}_{x_0 \sim \pi_0} T_r = \int_\Omega \mathbb{E}[T_r | x_0 = x] d\pi_0(x)$$

$$= \int_\Omega \mathbb{E}[T_r | x_0 = x] \frac{d\pi_0}{d\pi}(x) \, d\pi(x)$$

$$= \frac{1}{T} \int_\Omega \int_0^T \mathbb{E}\left[ T_{t,r} \frac{d\pi_0}{d\pi}(x_t) | x_0 = x \right] dr \, d\pi(x) \qquad (\text{F.3})$$

$$\leq \frac{1}{T} \mathbb{E} \int_0^T ((F_x^{-1}(F_x(t) + r) \vee T) - t) \frac{d\pi_0}{d\pi}(x_t) \, dt + \frac{1}{\alpha_0 T} \int_0^T \mathbb{E}[T_{t,r} \mathbb{1}_{T_{t,r} \geq T-t}] \, dt$$

$$= \frac{1}{\alpha_0 T} \mathbb{E} \int_0^T ((F_x^{-1}(F_x(t) + r) \vee T) - t) F_x'(t) \, dt + \frac{1}{\alpha_0 T} \int_0^T \mathbb{E}[T_{t,r} \mathbb{1}_{T_r \geq t}] \, dt.$$

where (F.3) uses the fact that for any $t$, the distribution of $x_t$ is still $\pi$. Because $\mathbb{E}T_{t,r} < \infty$, $g(t) := \mathbb{E}[T_{t,r}\mathbb{1}_{T_r \geq t}] \to 0$ by the Dominated Convergence Theorem. Now if $g(t)$ is bounded and $\lim_{t\to\infty} g(t) = 0$, then $\lim_{T\to\infty} \frac{1}{T}\int_0^T g(t)\,dt = 0$, so the second term converges to 0 as $T \to \infty$. We focus on the first term. Change of variable gives

$$\int_0^T ((F_x^{-1}(F_x(t) + r) \vee T) - t)F_x'(t)\,dt = \int_0^{F_x(T)} (F_x^{-1}(y + r) \vee T) - F_x^{-1}(y)\,dy$$

$$= \int_0^{F_x(T)} \int_{F_x^{-1}(y)}^{F_x^{-1}(y+r)\vee T} dz\,dy.$$

This is the measure of

$$\{(y, z) : 0 \leq y \leq F_x(T), F_x^{-1}(y) \leq z \leq F_x^{-1}(y + r) \vee T\}$$
$$\subseteq \{(y, z) : y \leq F_x(z) \leq y + r, z \leq T\}$$
$$= \{(y, z) : F_x(z) - r \leq y \leq F_x(z), 0 \leq z \leq T\}$$

which evidently has measure $Tr$. Hence taking $T \to \infty$,

$$\mathbb{E}_{x_0 \sim \pi_0} T_r \leq \frac{1}{\alpha_0 T} \cdot Tr = \frac{r}{\alpha_0}.$$

Let $K_\infty = \left\|\frac{d\nu_0}{d\pi_0}\right\|_\infty$ and $K_2^2 = \chi^2(\nu_0 \| \pi_0) + 1$. For the first bound,

$$\mathbb{E}_{x_0 \sim \nu_0} T_r \leq K_\infty \mathbb{E}_{x_0 \sim \pi_0} T_r \leq \frac{K_\infty r}{\alpha_0}$$

$$\implies \mathbb{E}_{x_0 \sim \nu_0, t \sim \mathsf{Unif}(0,T)} \left[\alpha_0 \frac{d\pi_0}{d\pi}(x_t)\right] \geq \frac{1}{T}\int_0^T \left(1 - \frac{\mathbb{E}_{x_0 \sim \nu_0} T_r}{T}\right) \vee 0\,dr$$

$$\geq \frac{1}{T}\int_0^T \left(1 - \frac{K_\infty r}{\alpha_0 T}\right) \vee 0\,dr = \frac{\alpha_0}{2K_\infty}.$$

For the second bound, let

$$G = \left\{x : \frac{d\nu_0}{d\pi_0}(x) \leq \frac{K_2^2}{\varepsilon}\right\}$$

and note by Markov's inequality that

$$\mathbb{P}_{x_0 \sim \nu_0}(T_r > T) \leq P(G^c) + \mathbb{P}_{x_0 \sim P_0}(T_r > T \wedge x_0 \in G)$$

$$\leq \varepsilon + \frac{K_2^2}{\varepsilon} \frac{\mathbb{E}_{x_0 \sim \pi_0} T_r}{T}$$

$$\leq \varepsilon + \frac{K_2^2}{\varepsilon} \frac{r}{\alpha_0 T}$$

$$= 2K_2 \sqrt{\frac{r}{\alpha_0 T}} \qquad\qquad \text{taking } \varepsilon = K_2\sqrt{\frac{r}{\alpha_0 T}}.$$

Then

$$\mathbb{E}_{x_0 \sim \nu_0, t \sim \mathsf{Unif}(0,T)} \left[\alpha_0 \frac{d\pi_0}{d\pi}(x_t)\right] = \frac{1}{T}\int_0^T (1 - \mathbb{P}_{x_0 \sim \nu_0}(T_r > T))\,dr$$

$$\leq \frac{1}{T}\int_0^T \left(1 - 2K_2\sqrt{\frac{r}{\alpha_0 T}}\right) \vee 0\,dr = \frac{\alpha_0 T}{12K_2^2}.$$

$\square$

**Lemma F.3.** *On the state space $\Omega \times [L]$ we define the measures*

$$\pi(x, i) := \sum_{j=1}^L \omega^j \pi^j(x) I\{j = i\}$$

$$\pi_0(x, i) := \sum_{j=1}^L \omega_0^j \pi_0^j(x) I\{j = i\}$$

where $\pi^i(x)$ is a p.m. on $\Omega$ with component p.m. $\pi_0^i(x)$. Consider running the Markov chain $P$ on $\Omega \times [L]$ with stationary measure $\pi(x,i)$ from initial measure $\nu_0(x,i)$. Let $\overline{p}_T(x,i)$ be the distribution of $X_t$ where $t \sim \mathsf{Unif}(0,T)$ and $X_0 \sim \nu_0(x,i)$. Then

$$
\int_\Omega \left( \frac{\overline{p}_T(x,L)\frac{\pi_0^L(x)}{\omega^L \pi^L(x)} \Big/ \int_\Omega \overline{p}_T(x,i)\frac{\pi_0^L(x)}{\omega^L \pi^L(x)}dx}{\pi_0^L(x)} - 1 \right)^2 \pi_0^L(x)dx \le \frac{\chi^2\big(\nu_0(x,i)||\pi(x,i)\big) \cdot C_{PI}\big(\pi_0(x,i)\big)}{\alpha_0 \omega_0^L \cdot \left( \int_\Omega \overline{p}_T(x,L)\frac{\pi_0^L(x)}{\omega^L \pi^L(x)}dx \right)^2 \cdot T},
$$

where $\alpha_0$ is the component weight of $\pi_0(x,i)$ in $\pi(x,i) = \alpha_0 \pi_0(x,i) + (1-\alpha_0)\pi_1(x,i)$.

*Proof.* We consider the following,

$$
\mathrm{Var}_{\pi_0(x,i)}\left( \overline{p}_T(x,i)\frac{\pi_0(x,i)}{\pi(x,i)} \Big/ \pi_0(x,i) \right) = \sum_{i=1}^L \int_\Omega \left( \frac{\overline{p}_T(x,i)\frac{\pi_0(x,i)}{\pi(x,i)}}{\pi_0(x,i)} - \mathbb{E}_{\pi_0(x,i)}\left[ \overline{p}_T(x,i)\frac{\pi_0(x,i)}{\pi(x,i)} \Big/ \pi_0(x,i) \right] \right)^2 \pi_0(x,i)dx
$$

$$
= \sum_{i=1}^L \int_\Omega \left( \frac{\overline{p}_T(x,i)\frac{\pi_0(x,i)}{\pi(x,i)}}{\pi_0(x,i)} - \mathbb{E}_{\pi_0(x,i)}\left[ \frac{\overline{p}_T(x,i)}{\pi(x,i)} \right] \right)^2 \pi_0(x,i)dx
$$

$$
= \sum_{i=1}^L \omega_0^i \int_\Omega \left( \frac{\overline{p}_T(x,i)\frac{\pi_0(x,i)}{\pi(x,i)}}{\pi_0(x,i)} - \int_\Omega \frac{\pi_0(x,i)}{\omega_0^i} \frac{\overline{p}_T(x,i)}{\pi(x,i)} + \int_\Omega \frac{\pi_0(x,i)}{\omega_0^i} \frac{\overline{p}_T(x,i)}{\pi(x,i)} - \mathbb{E}_{\pi_0(x,i)}\left[ \frac{\overline{p}_T(x,i)}{\pi(x,i)} \right] \right)^2 \pi_0^i(x)dx
$$

$$
\ge \omega_0^L \int_\Omega \left( \frac{\overline{p}_T(x,L)\frac{\pi_0^L(x)}{\omega^L \pi^L(x)}}{\pi_0^L(x)} - \int_\Omega \pi_0^L(x)\frac{\overline{p}_T(x,L)}{\omega^L \pi^L(x)} \right)^2 \pi_0^L(x)dx
$$

$$
= \omega_0^L \int_\Omega \left( \frac{\overline{p}_T(x,L)\frac{\pi_0^L(x)}{\omega^L \pi^L(x)} \Big/ \int_\Omega \overline{p}_T(x,L)\frac{\pi_0^L(x)}{\omega^L \pi^L(x)}dx}{\pi_0^L(x)} - 1 \right)^2 \pi_0^L(x)dx \cdot \left( \int_\Omega \overline{p}_T(x,L)\frac{\pi_0^L(x)}{\omega^L \pi^L(x)}dx \right)^2
$$

Note that $\overline{p}_T(x,L)$ is the non-normalized component of the p.m. at the $L$-th level. However, this is homogeneous in the numerator of the previous expression therefore we have that this equals

$$
= \omega_0^L \int_\Omega \left( \frac{\overline{p}_T(x,L)\frac{\pi_0^L(x)}{\omega^L \pi^L(x)} \Big/ \int_\Omega \overline{p}_T(x,L)\frac{\pi_0^L(x)}{\omega^L \pi^L(x)}dx}{\pi_0^L(x)} - 1 \right)^2 \pi_0^L(x)dx \cdot \left( \int_\Omega \overline{p}_T(x,L)\frac{\pi_0^L(x)}{\omega^L \pi^L(x)}dx \right)^2,
$$

To finish the proof, we apply Lemma F.1 which says

$$
\mathrm{Var}_{\pi_0(x,i)}\left( \overline{p}_T(x,i)\frac{\pi_0(x,i)}{\pi(x,i)} \Big/ \pi_0(x,i) \right) \le \frac{\chi^2\big(\nu_0(x,i)||\pi(x,i)\big) \cdot C_{PI}\big(\pi_0(x,i)\big)}{\alpha_0 \cdot T}.
$$

Therefore we have that

$$
\int_\Omega \left( \frac{\overline{p}_T(x,L)\frac{\pi_0^L(x)}{\omega^L \pi^L(x)} \Big/ \int_\Omega \overline{p}_T(x,L)\frac{\pi_0^L(x)}{\omega^L \pi^L(x)}dx}{\pi_0^L(x)} - 1 \right)^2 \pi_0^L(x)dx \le \frac{\chi^2\big(\nu_0(x,i)||\pi(x,i)\big) \cdot C_{PI}\big(\pi_0(x,i)\big)}{\alpha_0 \omega_0^L \cdot \left( \int_\Omega \overline{p}_T(x,L)\frac{\pi_0^L(x)}{\omega^L \pi^L(x)}dx \right)^2 \cdot T}.
$$

$\square$

**Lemma F.4.** *Let* $\tilde{X}_t = (X,i) \in \Omega \times [L]$ *and* $\tilde{Y} = (Y,i) \in \Omega \times [L]$ *with* $\tilde{X}_t$ *drawn from the density* $\overline{p}_T(x,i)\frac{\pi_0(x,i)}{\pi(x,i)} \Big/ Z$, *where* $\overline{p}_T$ *is the distribution of* $X_t$ *with* $t \sim \mathsf{Unif}(0,T)$, $Z = \sum_{i=1}^L \int_\Omega \overline{p}_T(x,i)\frac{\pi_0(x,i)}{\pi(x,i)}dx$, *and* $\tilde{Y} \sim \pi_0(x,i)$. *On the state space* $\Omega \times [L]$ *we define the measures*

$$
\pi(x,i) := \sum_{j=1}^L \omega^j \pi^j(x) I\{j=i\}
$$

$$
\pi_0(x,i) := \sum_{j=1}^L \omega_0^j \pi_0^j(x) I\{j=i\}
$$

*and the relation $\pi(x,i) = \alpha_0 \pi_0(x,i) + (1 - \alpha_0)\pi_1(x,i)$. Then*

$$\int_\Omega \overline{p}_T(x,L) \frac{\pi_0^L(x)}{\omega^L \pi^L(x)} dx \geq \frac{1}{2 \left\| \frac{\nu_0}{\pi_0} \right\|_\infty} - \left( \frac{\chi^2\big(\nu_0(x,i) \| \pi(x,i)\big) \cdot C_{PI}\big(\pi_0(x,i)\big)}{\alpha_0 (\omega_0^L)^2 \cdot T} \right)^{\frac{1}{2}}.$$

*Proof.* By Lemma F.1, for $\epsilon = \frac{K_0 C}{\alpha_0 \big[\mathbb{E}_{\overline{p}_T}\big(\frac{\pi_0}{\pi}\big)\big]^2 T}$, we have that

$$\chi^2\left( \overline{p}_T(x,i) \frac{\pi_0(x,i)}{\pi(x,i)} \middle/ Z \, \middle\| \, \pi_0(x,i) \right) \leq \epsilon.$$

The data processing inequality for random variables with $f(x) = I(x = L)$ yields for any random variables $\tilde{X}, \tilde{Y}$ that

$$\chi^2\left( \tilde{X}_t \, \middle\| \, \tilde{Y} \right) \geq \chi^2\left( I\{\tilde{X}_t = L\} \, \middle\| \, I\{\tilde{Y} = L\} \right)$$

$$\implies \chi^2\left( \overline{p}_T(x,i) \frac{\pi_0(x,i)}{\pi(x,i)} \middle/ Z \, \middle\| \, \pi_0(x,i) \right) \geq \chi^2\left( \text{Bernoulli}\left( \frac{\int_\Omega \overline{p}_T(x,L) \frac{\omega_0^L \pi_0^L(x)}{\omega^L \pi^L(x)} dx}{\sum_i \int_\Omega \overline{p}_T(x,i) \frac{\pi_0(x,i)}{\pi(x,i)} dx} \right) \, \middle\| \, \text{Bernoulli}(\omega_0^L) \right)$$

The chi-squared divergence of two Bernoulli random variables is lower bounded by

$$\geq \left| \frac{\int_\Omega \overline{p}_T(x,L) \frac{\omega_0^L \pi_0^L(x)}{\omega^L \pi^L(x)} dx}{\sum_i \int_\Omega \overline{p}_T(x,i) \frac{\pi_0(x,i)}{\pi(x,i)} dx} - \omega_0^L \right|^2.$$

This yields a lower bound of

$$\int_\Omega \overline{p}_T(x,L) \frac{\omega_0^L \pi_0^L(x)}{\omega^L \pi^L(x)} dx \geq \left( \omega_0^L - \epsilon^{\frac{1}{2}} \right) \sum_i \int_\Omega \overline{p}_T(x,i) \frac{\pi_0(x,i)}{\pi(x,i)} dx$$

Applying Lemma F.1 yields

$$\geq \omega_0^L \int_\Omega \overline{p}_T(x,i) \frac{\pi_0(x,i)}{\pi(x,i)} dx - \left( \frac{\chi^2\big(\nu_0(x,i) \| \pi(x,i)\big) \cdot C_{PI}\big(\pi_0(x,i)\big)}{\alpha_0 \cdot T} \right)^{\frac{1}{2}}$$

which by Lemma F.2 is

$$\geq \frac{\omega_0^L}{2\|\frac{\nu_0}{\pi_0}\|_\infty} - \left( \frac{\chi^2\big(\nu_0(x,i) \| \pi(x,i)\big) \cdot C_{PI}\big(\pi_0(x,i)\big)}{\alpha_0 \cdot T} \right)^{\frac{1}{2}}$$

This yields

$$\int_\Omega \overline{p}_T(x,L) \frac{\pi_0^L(x)}{\omega^L \pi^L(x)} dx \geq \frac{1}{2\|\frac{\nu_0}{\pi_0}\|_\infty} - \left( \frac{\chi^2\big(\nu_0(x,i) \| \pi(x,i)\big) \cdot C_{PI}\big(\pi_0(x,i)\big)}{\alpha_0 (\omega_0^L)^2 \cdot T} \right)^{\frac{1}{2}}.$$

$\square$

# G   ESTIMATING PARTITION FUNCTIONS

In this section we show how to approximate the weights $w_{i,k}$ and $r_i$. Partition function approximation is standard for stimulated tempering on non-normalized distributions. One approach is to run the ST algorithm to the $l$th level and then acquire a Monte Carlo estimate of the partition function at the next level, see Ge et al. (2018c). However, in our setting, we also require an estimate of $Z_{i,k} = \int_\Omega \alpha_k \pi_k(x) q_i(x - x_k) dx$. Without access to the component functions $\pi_k(x)$ of the target measure $\pi(x) = \sum_k \alpha_k \pi_k(x)$, it is not possible to directly estimate $Z_{i,k}$ via Monte Carlo. Fortunately, we only require an estimate up to polynomial factors, so we can use the assumption that after tilting towards the warm start point, a significant chunk of the mass of $\pi(x) q_i(x - x_k)$ comes from $\pi_k(x) q_i(x - x_k)$ (Definition 1.1(2)). Hence, it will suffice to estimate $\overline{Z}_{i,k} = \int_\Omega \pi(x) q_i(x - x_k) dx$.

To obtain an estimate of $Z_{i,k}$, we define

$$\overline{\pi}_{l,k}(x) = p(x) \cdot q_l(x - x_k),$$

where $p(x) = \sum_k \alpha_k p_k(x)$ is the target function. Since we assume oracle access to the target $p(x)$ up to normalization and $q_l(x)$ is chosen, we can freely evaluate $\bar{\pi}_{l,k}(x)$. Next we define $\hat{p}_t(x, i) = \nu_0 P_{ST,tel}^t$ to be the distribution of a sample at the $i$-th level after running the ALPS process for time $t$ from an initial distribution $\nu_0$. This Markov process converges to the joint distribution of $p_i(x) = p(x) \cdot \sum_{k=1}^M w_{i,k} q_i(x - x_k)$ over the levels $i \in [1, l]$. Below we state the inductive hypothesis which assumes component and level balance (def. 2.3) is maintained through level $l$.

**Assumption G.1. (Inductive Hypothesis)** *Let $Z_l = \int_\Omega \tilde{p}_l(x)dx$ and $Z_{l,k} = \int_\Omega \alpha_k \pi_k(x) q_l(x - x_k)dx$, and $U$ be a given parameter. We make the following assumptions at the $l$-th level:*

$\boxed{\textbf{H1}(l)}$ *(Component balance)*

$$\frac{w_{i,k} Z_{i,k}}{w_{i,k'} Z_{i,k'}} \in \left[\frac{1}{C_1}, C_1\right] \quad \text{for all } k, k' \in [1, M] \text{ and } i \in [1, l],$$

*where $C_1 = poly(\frac{U}{c_{tilt}})$.*

$\boxed{\textbf{H2}(l)}$ *(Level balance)*

$$\frac{r_j^{(l)} Z_j}{r_{j'}^{(l)} Z_{j'}} \in \left[\frac{1}{C_2}, C_2\right] \text{ for all } j, j' \in [1, l],$$

*where $C_2 = poly(\frac{U}{c_{tilt}^2})$.*

The following lemma follows directly from the inductive hypothesis.

**Lemma G.2.** *Let Assumptions 1.1 and G.1 hold and let $\bar{Z}_{i,k} = \int_\Omega \pi(x) q_i(x - x_k)dx$. Then*

$$\frac{w_{i,k} \bar{Z}_{i,k}}{w_{i,k'} \bar{Z}_{i,k'}} \in \left[\frac{c_{tilt}}{C_1}, \frac{C_1}{c_{tilt}}\right] \quad \text{for all } k, k' \in [1, M] \text{ and } i \in [1, l].$$

The following two lemmas make clear how the inductive hypothesis is used to bound the weight component of modes at varying levels. Lemma G.3 shows that in the context of Algorithm 3, re-weighting the level weight $r_1^{(l)}$ at the initial level yields weights which still satisfies the inductive hypothesis **H2**($l$) but with a different constant. By placing more mass on the first level, the level re-weighting allows for a good portion of our target distribution to be aligned with our initialization. Therefore, in this section, the level weights $\{r_i^{(l)}\}_{i=1}^l$ will be replaced with $\{\hat{r}_i^{(l)}\}_{i=1}^l$ in the following section. This will allow us to consider the practical scaling where the initial level is up-weighted. Note that this pushes the work of the inductive hypothesis H2(l) onto the following lemma.

**Lemma G.3.** *Let **H2**($l$) hold and let $\hat{r}_1^{(l)} = l \cdot C_2 r_1^{(l)}$ and $\hat{r}_j^{(l)} = r_j^{(l)}$ for all $j = 2, \ldots, l$. Then*

$$\frac{1}{l \cdot C_2^2} \leq \frac{\hat{r}_j^{(l)} Z_j}{\hat{r}_k^{(l)} Z_k} \leq l \cdot C_2^2$$

*for all $k, j \in [1, l]$.*

*Proof.* The conclusion is clear for $j, k \neq 1$, which remain unscaled, by the inductive hypothesis $\frac{1}{C_2} \leq \frac{\hat{r}_j^{(l)} Z_j}{\hat{r}_k^{(l)} Z_k} \leq C_2$.

It suffices to show $\frac{\hat{r}_j^{(l)} Z_j}{\hat{r}_1^{(l)} Z_1} \in \left[\frac{1}{l \cdot C_2^2}, l \cdot C_2^2\right]$; then the same bound follows for the reciprocal. By **H2**($l$) applied to $r_j^{(l)}$,

$$\frac{\hat{r}_j^{(l)} Z_j}{\hat{r}_1^{(l)} Z_1} = \frac{1}{l \cdot C_2} \frac{r_j^{(l)} Z_j}{r_1^{(l)} Z_1} \in \left[\frac{1}{l \cdot C_2} \cdot \frac{1}{C_2}, \frac{1}{l \cdot C_2} \cdot C_2\right] \subset \left[\frac{1}{l \cdot C_2^2}, l \cdot C_2^2\right],$$

as needed.

$\square$

Note that H1($l$) says that components at the same level are approximately balanced, while H2($l$) says that different levels as a whole are approximately balanced. Putting these together, we obtain that components at different levels are also approximately balanced.

**Lemma G.4** (Balancing between all components at all levels). *Given Assumptions 1.1 and Assumptions G.1, we have*

$$\frac{\hat{r}_i^{(l)} w_{i,k} Z_{i,k}}{\hat{r}_{i'}^{(l)} w_{i',k'} Z_{i',k'}} \in \left[\frac{1}{C}, C\right] \quad \text{for all } i, i' \in [1, l] \text{ and } k, k' \in [1, M],$$

*where*

$$C = \frac{l C_2^2 C_1^2}{c_{tilt}}.$$

*Proof.* We start by using the tilting assumption and Lemma G.3,

$$\frac{1}{l C_2^2} \le \frac{\hat{r}_i^{(l)} Z_i}{\hat{r}_{i'}^{(l)} Z_{i'}} \le \frac{\frac{1}{c_{tilt}} \hat{r}_i^{(l)} \sum_j w_{i,j} Z_{i,j}}{\hat{r}_{i'}^{(l)} \sum_j w_{i',j} Z_{i',j}}$$

$$= \frac{\frac{1}{c_{tilt}} \hat{r}_i^{(l)} w_{i,k} Z_{i,k} \sum_j \frac{w_{i,j} Z_{i,j}}{w_{i,k} Z_{i,k}}}{\hat{r}_{i'}^{(l)} w_{i',k'} Z_{i',k'} \sum_j \frac{w_{i',j} Z_{i',j}}{w_{i',k'} Z_{i',k'}}}$$

By inductive assumption **H1**($l$),

$$\le \frac{\frac{1}{c_{tilt}} \hat{r}_i^{(l)} w_{i,k} Z_{i,k} M \cdot C_1}{\hat{r}_{i'}^{(l)} w_{i',k'} Z_{i',k'} M \cdot \frac{1}{C_1}}$$

$$= \frac{C_1^2}{c_{tilt}} \frac{\hat{r}_i^{(l)} w_{i,k} Z_{i,k}}{\hat{r}_{i'}^{(l)} w_{i',k'} Z_{i',k'}}$$

$$\implies \frac{c_{tilt}}{C_2 \cdot C_1^2} \le \frac{\hat{r}_i^{(l)} w_{i,k} Z_{i,k}}{\hat{r}_{i'}^{(l)} w_{i',k'} Z_{i',k'}}.$$

Since the above lower bound holds for all $i, i' \in [1, l]$ and $k, k' \in [1, M]$ the reciprocal holds as an upper bound. $\square$

**Proof Overview:** In the context of Algorithm 3:

1. We show that $\boxed{\textbf{H1}(l+1)}$ by showing the following,

$$\frac{1}{N} \sum_{j=1}^N \frac{\bar{\pi}_{l+1,k}(x_j)}{\tilde{p}(x_j, i_j)} I\{i_j = l\} \underbrace{\approx}_{\textbf{(A)}} \mathbb{E}_{\hat{p}_t}\left[\frac{\bar{\pi}_{l+1,k}}{\tilde{p}(x, i)} I\{i = l\}\right] \underbrace{\asymp}_{\textbf{(B)}} \mathbb{E}_p\left[\frac{\bar{\pi}_{l+1,k}}{\tilde{p}(x, i)} I\{i = l\}\right] \underbrace{\asymp}_{\textbf{(C)}} \mathbb{E}_p\left[\frac{\tilde{\pi}_{l+1,k}}{\tilde{p}(x, i)} I\{i = l\}\right] = \frac{Z_{l+}}{Z}$$

   where in the **(B)** and **(C)** steps we show that the two terms are within a constant factor.

   (a) In Lemma G.5, we prove (A) using Chebyshev's inequality.
   (b) In Lemma G.6, we prove (B) by utilizing the work in Section F, which shows convergence of the Markov process to the "good" part of the stationary distribution.
   (c) In Lemma G.7, we show (C) by using the tilting coefficient $c_{tilt}$ to compare $\bar{\pi}_{l,k}$ and $\tilde{\pi}_{l,k}$.

2. We obtain an estimate for the partition function of $\tilde{p}_{l+1}(x) = \left(\sum_j \alpha_j \pi_j(x)\right)\left(\sum_k w_{l+1,k} q_{l+1}(x - x_k)\right)$ by again showing that

$$\frac{1}{N} \sum_{j=1}^N \frac{\tilde{p}_{l+1}(x_j)}{\tilde{p}(x_j, i_j)} I\{i_j = l\} \underbrace{\approx}_{\textbf{(A)}} \mathbb{E}_{\hat{p}_t}\left[\frac{\tilde{p}_{l+1}(x)}{\tilde{p}(x, i)} I\{i = l\}\right] \underbrace{\asymp}_{\textbf{(B')}} \mathbb{E}_p\left[\frac{\tilde{p}_{l+1}(x)}{\tilde{p}(x, i)} I\{i = l\}\right] = \frac{Z_{l+1}}{Z}.$$

   (a) In Lemma G.5, we prove (A) as an application of Chebyshev's inequality.

(b) In Lemma G.8, we prove (B') in a similar fashion to Lemma G.6 for (B).

3. We then show $\boxed{\textbf{H2}(l+1)}$ by level rebalance.

We split the work of this section into three subsections. The first, Subsection G.1, finds bounds between the ratios of the expectations terms from the proof overview. These bounds contain several constants which depend on the spectral gap and the mixing time of the projected chain. In Subsection G.2, we give an upper bound on the spectral and analyze the mixing time of the projected chain. Lastly, Subsection G.3 combines the results from the previous two subsections to show that running Algorithm 3 mains the level balance in the inductive hypothesis.

## G.1 BOUNDING THE APPROXIMATIONS

**Lemma G.5 (A: Chebyshev).** *Given i.i.d. samples $x_i \sim \pi$ for $1 \le i \le N$ with $\frac{\mathbb{E}_\pi\left[f^2\right]}{\mathbb{E}_\pi\left[f\right]^2} \le R$, then with probability at least $1 - \delta$,*

$$1 - \epsilon \le \frac{\frac{1}{N}\sum_{i=1}^N f(x_i)}{\mathbb{E}_\pi[f]} \le 1 + \epsilon$$

*where $\epsilon = \sqrt{\frac{R}{N \cdot \delta}}$.*

*Proof.* This is a simple application of Chebyshev's inequality,

$$\mathbb{P}\left(\left|\frac{\frac{1}{N}\sum_{i=1}^N f(x_i)}{\mathbb{E}_\pi[f]} - 1\right| > \epsilon\right) = \mathbb{P}\left(\left|\frac{1}{N}\sum_{i=1}^N f(x_i) - \mathbb{E}_\pi[f]\right| > \epsilon \cdot \mathbb{E}_\pi[f]\right) \le \frac{\mathbb{E}_\pi[f^2]}{n\epsilon^2\mathbb{E}_\pi[f]^2} \le \frac{R}{n\epsilon^2}.$$

Letting $\epsilon = \sqrt{\frac{R}{N \cdot \delta}}$ yields the desired result. $\qquad\square$

In the following Lemma G.6 we are able to show that $\mathbb{E}_{\hat{p}_t}\left[\frac{\bar{\pi}_{l+1,k}}{\tilde{p}(x,i)}I\{i=l\}\right] \asymp \mathbb{E}_p\left[\frac{\bar{\pi}_{l+1,k}}{\tilde{p}(x,i)}I\{i=l\}\right]$. This is a consequence of our results in Section F that show $\hat{p}^t$ converges to the good part of $p(x)$.

**Lemma G.6 (B).** *Given Assumptions 1.1 and Assumptions 3.1*

$$c_{tilt}\left(\frac{1}{2||\frac{\nu_0(x,i)}{p_0(x,i)}||_\infty} - \left(1 + \chi^2\left(\pi_{l+1,k} \,||\, \pi_{l,k}\right)^{\frac{1}{2}}\right) \cdot \Delta\right) \le \frac{\mathbb{E}_{\hat{p}_T}\left[\frac{\bar{\pi}_{l+1,k}(x)}{\tilde{p}(x,i)}I\{i=l\}\right]}{\mathbb{E}_p\left[\frac{\bar{\pi}_{l+1,k}(x)}{\tilde{p}(x,i)}I\{i=l\}\right]} \le \left\|\frac{\nu_0(x,i)}{p(x,i)}\right\|_{L^\infty},$$

*where $\Delta = \left(\frac{\chi^2\left(\nu_0(x,i)||p(x,i)\right) \cdot C_{PI}\left(p_0(x,i)\right)}{\alpha_0(\omega_0^L)^2 \cdot T}\right)^{\frac{1}{2}}$.*

*Proof.* Upper bound. Note that for any $f \ge 0$ and $p \ll q$ that $\frac{\mathbb{E}_p f}{\mathbb{E}_q f} = \frac{\mathbb{E}_q \frac{dp}{dq} f}{\mathbb{E}_q f} \le \left\|\frac{dp}{dq}\right\|_{L^\infty}$. Applying this here and then using contraction gives

$$\frac{\mathbb{E}_{\hat{p}_T}\left[\frac{\bar{\pi}_{l+1,k}(x)}{\tilde{p}(x,i)}I\{i=l\}\right]}{\mathbb{E}_p\left[\frac{\bar{\pi}_{l+1,k}(x)}{\tilde{p}(x,i)}I\{i=l\}\right]} = \left\|\frac{\hat{p}_T(x,i)}{p(x,i)}\right\|_{L^\infty} \le \left\|\frac{\nu_0(x,i)}{p(x,i)}\right\|_{L^\infty}.$$

Lower bound. For the lower bound, we first compare the denominator to just the integral of the $k$th component, i.e. $Z_{l,k}$, using the tilting assumption. Noting that $\tilde{p}(x,i)I\{i=l\} = r_l\tilde{p}_l(x)$ and denoting $\hat{p}_{l,T}(x) = \hat{p}_T(x,l)$. Also note that in order to apply the lemmas in Section F we define $\pi_0(x,i) = \omega_0^l \pi_{l,k}(x)I\{i=l\} + \sum_{j=1}^{l-1} w_0^j p_j(x)I\{j=i\}$. Then we obtain the lower bound

$$\frac{\mathbb{E}_{\hat{p}_T}\left[\frac{\bar{\pi}_{l+1,k}(x)}{\tilde{p}(x,i)}I\{i=l\}\right]}{\mathbb{E}_p\left[\frac{\bar{\pi}_{l+1,k}(x)}{\tilde{p}(x,i)}I\{i=l\}\right]} = \frac{\int_\Omega \hat{p}_{l,T}(x)\frac{\bar{\pi}_{l+1,k}(x)}{r_l\tilde{p}_l(x)}dx}{\int_\Omega \omega^l p_l(x)\frac{\bar{\pi}_{l+1,k}(x)}{r_l\tilde{p}_l(x)}dx} = \frac{\int_\Omega \hat{p}_{l,T}(x)\frac{\bar{\pi}_{l+1,k}(x)}{r_l\tilde{p}_l(x)\frac{1}{Z_l}}dx}{\int_\Omega \omega^l p_l(x)\frac{\bar{\pi}_{l+1,k}(x)}{r_l\tilde{p}_l(x)\frac{1}{Z_l}}dx}$$

replacing $r_l^{(l)}$ are constants, so canceling terms yields

$$= \frac{\int_\Omega \hat{p}_{l,T}(x) \frac{\bar{\pi}_{l+1,k}(x)}{p_l(x)} dx}{\int_\Omega \omega^l \bar{\pi}_{l+1,k}(x)}.$$

Next we apply the definition of $\bar{\pi}_{l+1,k}$ and $\pi(x) = \sum_k \alpha_k \pi_k(x)$

$$= \frac{\int_\Omega \hat{p}_{l,T}(x) \left[ \frac{\pi(x) \cdot q_{l+1}(x - x_k)}{\omega^l p_l(x)} \right] dx}{\int_\Omega \pi(x) \cdot q_{l+1}(x - x_k) dx} \geq \frac{\int_\Omega \hat{p}_{l,T}(x) \left[ \frac{\alpha_k \pi_k(x) \cdot q_{l+1}(x - x_k)}{\omega^l p_l(x)} \right] dx}{\int_\Omega \pi(x) \cdot q_{l+1}(x - x_k) dx}$$

by tilting, Assumption 1.1(2)

$$\geq c_{tilt} \frac{\int_\Omega \hat{p}_{l,T}(x) \left[ \frac{\alpha_k \pi_k(x) \cdot q_{l+1}(x - x_k)}{\omega^l p_l(x)} \right] dx}{\int_\Omega \alpha_k \pi_k \cdot q_{l+1}(x - x_k) dx} = c_{tilt} \int_\Omega \hat{p}_{l,T}(x) \left[ \frac{\alpha_k \pi_k(x) \cdot q_{l+1}(x - x_k) \big/ Z_{l+1,k}}{\omega^l p_l(x)} \right] dx$$

$$= c_{tilt} \mathbb{E}_{\pi_{l,k}} \left[ \frac{\hat{p}_{l,T}}{\pi_{l,k}} \frac{\pi_{l+1,k}}{\omega^l p_l} \right] = c_{tilt} \cdot \mathbb{E}_{\pi_{l,k}} \left[ \frac{\pi_{l+1,k}}{\pi_{l,k}} \cdot \frac{\hat{p}_{l,T} \frac{\pi_{l,k}}{\omega^l p_l}}{\pi_{l,k}} \right],$$

where we write it in this way so we can use the closeness of $\pi_{l+1,k}$ to $\pi_{l,k}$ and the convergence of $\hat{p}_{l,T}$ to the good part $\pi_{l,k}$ of $p$ on the $l$th level. Let $\Delta = \left( \frac{\chi^2 \left( \nu_0(x,i) || p(x,i) \right) \cdot C_{PI} \left( p_0(x,i) \right)}{\alpha_0 (\omega_0^l)^2 \cdot T} \right)^{\frac{1}{2}}$. We first bound the expectation when $\hat{p}_{l,T} \frac{\pi_{l,T}}{\omega^l p_l}$ is normalized to a probability distribution:

$$\left| \mathbb{E}_{\pi_{l,k}} \left[ \frac{\pi_{l+1,k}}{\pi_{l,k}} \cdot \frac{\hat{p}_{l,T} \frac{\pi_{l,k}}{p_l} \Big/ \int_\Omega \hat{p}_{l,T} \frac{\pi_{l,k}}{p_l} dx}{\pi_{l,k}} \right] - 1 \right|$$

$$\leq \mathbb{E}_{\pi_{l,k}} \left| \left( \frac{\pi_{l+1,k}}{\pi_{l,k}} - 1 \right) \cdot \left( \frac{\hat{p}_{l,T} \frac{\pi_{l,k}}{p_l} \Big/ \int_\Omega \hat{p}_{l,T} \frac{\pi_{l,k}}{p_l} dx}{\pi_{l,k}} - 1 \right) \right|$$

by Cauchy-Schwarz,

$$\leq \chi^2 \left( \pi_{l+1,k} \,||\, \pi_{l,k} \right)^{\frac{1}{2}} \cdot \chi^2 \left( \frac{\hat{p}_{l,T} \frac{\pi_{l,k}}{p_l}}{\int_\Omega \hat{p}_{l,T} \frac{\pi_{l,k}}{p_l} dx} \,||\, \pi_{l,k} \right)^{\frac{1}{2}}$$

by Lemma F.3,

$$\leq \chi^2 \left( \pi_{l+1,k} \,||\, \pi_{l,k} \right)^{\frac{1}{2}} \cdot \frac{\Delta}{\int_\Omega \nu_0 P^T(x,i) \frac{\pi_{l,k}}{\omega^l p_l(x)} dx}$$

Multiplying by $\int_\Omega \hat{p}_{l,T} \frac{\pi_{l,k}}{\omega^l p_l} dx$, we have that

$$\mathbb{E}_{\pi_{l,k}} \left[ \frac{\pi_{l+1,k}}{\pi_{l,k}} \cdot \frac{\hat{p}_{l,T} \frac{\pi_{l,k}}{\omega^l p_l}}{\pi_{l,k}} \right] \geq \int_\Omega \hat{p}_{l,T} \frac{\pi_{l,k}}{\omega^l p_l} dx - \chi^2 \left( \pi_{l+1,k} \,||\, \pi_{l,k} \right)^{\frac{1}{2}} \cdot \Delta.$$

It remains to lower-bound this first term, which is the fraction of mass that is considered to "belong" to the $k$th component after running for time $T$, compared to the fraction for the stationary distribution. This is lower-bounded by Lemma F.4,

$$\mathbb{E}_{\pi_{l,k}} \left[ \frac{\pi_{l+1,k}}{\pi_{l,k}} \cdot \frac{\hat{p}_{l,T} \frac{\pi_{l,k}}{\omega^l p_l}}{\pi_{l,k}} \right] \geq \frac{1}{2 \left\| \frac{\nu_0(x,i)}{p_0(x,i)} \right\|_\infty} - \Delta - \chi^2 \left( \pi_{l+1,k} \,||\, \pi_{l,k} \right)^{\frac{1}{2}} \cdot \Delta.$$

$\square$

**Lemma G.7.** *(C) Given assumptions 1.1 then*

$$1 \leq \frac{\mathbb{E}_{p(x,i)} \left[ \frac{\bar{\pi}_{l+1,k}}{\bar{p}(x,i)} I\{i = l\} \right]}{\mathbb{E}_{p(x,i)} \left[ \frac{\bar{\pi}_{l+1,k}}{\bar{p}(x,i)} I\{i = l\} \right]} \leq \frac{1}{c_0}.$$

*Proof.* By assumptions 1.1,

$$\frac{\mathbb{E}_p\left[\frac{\bar{\pi}_{l+1,k}}{\tilde{p}(x,i)}I\{i=l\}\right]}{\mathbb{E}_p\left[\frac{\tilde{\pi}_{l+1,k}}{\tilde{p}(x,i)}I\{i=l\}\right]} = \frac{\int_\Omega p(x)q_{l+1}(x-x_k)dx}{\mathbb{E}_p\left[\frac{\bar{\pi}_{l+1,k}}{p(x,i)}I\{i=l\}\right]} \leq \frac{1}{c_0}.$$

and we also have,

$$\frac{\mathbb{E}_p\left[\frac{\bar{\pi}_{l+1,k}}{\tilde{p}(x,i)}I\{i=l\}\right]}{\mathbb{E}_p\left[\frac{\tilde{\pi}_{l+1,k}}{\tilde{p}(x,i)}I\{i=l\}\right]} = \frac{\int_\Omega \sum_k \alpha_k p_k(x)q_{l+1}(x-x_k)dx}{\int_\Omega \alpha_k p_k(x)q_{l+1}(x-x_k)dx} \geq 1.$$

$\square$

In the following Lemma G.8 we show that $\mathbb{E}_{\hat{p}_t}\left[\frac{\tilde{p}_{l+1}(x)}{\tilde{p}(x,i)}I\{i=l\}\right] \asymp \mathbb{E}_p\left[\frac{\tilde{p}_{l+1}(x)}{\tilde{p}(x,i)}I\{i=l\}\right]$. Conceptually and proof-wise this is the same as Lemma G.6. The only difference is that now we are showing the closeness in the importance estimate holds for the entire next level $\tilde{p}_{l+1}(x)$ with the learned weights $w_{l+1,k}$.

**Lemma G.8.** *(B') Given Assumptions 1.1 and Assumptions 3.1. Then*

$$c_{tilt} \cdot \left(\frac{1}{2\|\frac{\nu_0(x,i)}{p_0(x,i)}\|_\infty} - \left(1 + \chi^2\left(\pi_{l+1,k} \,\|\, \pi_{l,k}\right)^{\frac{1}{2}}\right) \cdot \Delta_{C_1}\right) \leq \frac{\mathbb{E}_{\hat{p}_T}\left[\frac{\tilde{p}_{l+1}(x)}{\tilde{p}(x,i)}I\{i=l\}\right]}{\mathbb{E}_p\left[\frac{\tilde{p}_{l+1}(x)}{\tilde{p}(x,i)}I\{i=l\}\right]} \leq \left\|\frac{\nu_0(x,i)}{p(x,i)}\right\|_{L^\infty},$$

*where* $\Delta_{C_1} = \left(\frac{C_1^2 \cdot \chi^2\left(\nu_0(x,i)\|\pi(x,i)\right) \cdot C_{PI}\left(\pi_0(x,i)\right)}{\alpha_0(\omega_0^L)^2 \cdot T}\right)^{\frac{1}{2}}.$

*Proof.* We show that by normalizing $\tilde{p}$,

$$\frac{\mathbb{E}_{\hat{p}_T}\left[\frac{\tilde{p}_{l+1}(x)}{\tilde{p}(x,i)}\right]}{\mathbb{E}_p\left[\frac{\tilde{p}_{l+1}(x)}{\tilde{p}(x,i)}\right]} = \frac{\sum_i \int_\Omega \frac{\hat{p}_T(x,i)}{p(x,i)}p(x,i)\frac{\tilde{p}_{l+1}(x)}{\tilde{p}(x,i)}I\{i=l\}dx}{\mathbb{E}_p\left[\frac{\tilde{p}_{l+1}(x)}{\tilde{p}(x,i)}I\{i=l\}\right]}$$

$$\leq \frac{\left\|\frac{\hat{p}_T(x,i)}{p(x,i)}\right\|_{L^\infty}\sum_i \int_\Omega |\frac{\tilde{p}_{l+1}(x)}{\tilde{p}(x,i)}|p(x,i)I\{i=l\}dx}{\mathbb{E}_p\left[\frac{\tilde{p}_{l+1}(x)}{\tilde{p}(x,i)}I\{i=l\}\right]}$$

$$= \left\|\frac{\hat{p}_T(x,i)}{p(x,i)}\right\|_{L^\infty}$$

By contraction,

$$\leq \left\|\frac{\nu_0(x,i)}{p(x,i)}\right\|_{L^\infty}$$

Denote $\hat{p}_{l,T}(x) = \hat{p}_T(x,l)$. Then we obtain the lower bound

$$\frac{\mathbb{E}_{\hat{p}_T}\left[\frac{\tilde{p}_{l+1}(x)}{\tilde{p}(x,i)}I\{i=l\}\right]}{\mathbb{E}_{p_l}\left[\frac{\tilde{p}_{l+1}(x)}{\tilde{p}(x,i)}I\{i=l\}\right]} = \frac{\mathbb{E}_{\hat{p}_T(x,l)}\left[\frac{\tilde{p}_{l+1}(x)}{\omega^l p_l(x)}\right]}{\int_\Omega \tilde{p}_{l+1}(x)dx} \geq \frac{\mathbb{E}_{\hat{p}_{l,T}}\left[\frac{\sum_k w_{l+1,k}\alpha_k \pi_k(x) \cdot q_l(x-x_k)}{\omega^l p_l(x)}\right]}{\int_\Omega \tilde{p}_{l+1}(x)dx}$$

by tilting assumptions 1.1

$$\geq c_{tilt} \frac{\mathbb{E}_{\hat{p}_{l,T}}\left[\frac{\sum_k w_{l+1,k}\alpha_k \pi_k(x)\cdot q_l(x-x_k)}{\omega^l p_l(x)}\right]}{\int_\Omega \sum_k w_{l+1,k}\alpha_k \pi_k(x)\cdot q_l(x-x_k)dx} = \mathbb{E}_{\hat{p}_{l,T}}\left[\frac{\frac{1}{Z_{l+1,0}}\sum_k w_{l+1,k}\alpha_k \pi_k(x)\cdot q_l(x-x_k)}{\omega^l p_l(x)}\right]$$

Let $p_{i,0}(x) = \frac{1}{Z_{l+1,0}}\sum_k w_{i,k}\alpha_k \pi_k(x)\cdot q_i(x-x_k)$, where $Z_{l+1,0} = \sum_k w_{l+1,k}Z_{l+1,k}$

$$= c_{tilt}\cdot \mathbb{E}_{p_{l,0}}\left[\frac{p_{l+1,0}}{p_{l,0}}\cdot \frac{\hat{p}_{l,T}\frac{p_{l,0}}{\omega^l p_l}}{p_{l,0}}\right].$$

Let $\Delta_{C_1} = \left(\frac{C_1^2\cdot \chi^2\big(\nu_0(x,i)||\pi(x,i)\big)\cdot C_{PI}\big(\pi_0(x,i)\big)}{\alpha_0(\omega_0^L)^2\cdot T}\right)^{\frac{1}{2}}$ then we can show

$$\mathbb{E}_{p_{l,0}}\left[\frac{p_{l+1,0}}{p_{l,0}}\cdot \frac{\hat{p}_{l,T}\frac{p_{l,0}}{\omega^l p_l}}{p_{l,0}}\right] \geq \frac{1}{2||\frac{\nu_0(x,i)}{p_0(x,i)}||_\infty} - \left(1 + \chi^2\Big(\pi_{l+1,k}\,||\,\pi_{l,k}\Big)^{\frac{1}{2}}\right)\cdot \Delta_{C_1}.$$

Consider the following

$$\left|\mathbb{E}_{p_{l,0}}\left[\frac{p_{l+1,0}}{p_{l,0}}\cdot \frac{\hat{p}_{l,T}\frac{p_{l,0}}{\omega^l p_l}\Big/\int_\Omega \hat{p}_{l,T}\frac{p_{l,0}}{\omega^l p_l}dx}{p_{l,0}}\right] - 1\right|$$

$$\leq \mathbb{E}_{p_{l,0}}\left|\left(\frac{p_{l+1,0}}{p_{l,0}} - 1\right)\cdot \left(\frac{\hat{p}_{l,T}\frac{p_{l,0}}{p_l}\Big/\int_\Omega \hat{p}_{l,T}\frac{p_{l,0}}{p_l}dx}{p_{l,0}} - 1\right)\right|.$$

By Cauchy-Shwarz,

$$\leq \chi^2\Big(p_{l+1,0}\,||\,p_{l,0}\Big)^{\frac{1}{2}}\cdot \chi^2\left(\frac{\hat{p}_{l,T}\frac{p_{l,0}}{p_l}}{\int_\Omega \hat{p}_{l,T}\frac{p_{l,0}}{p_l}dx}\,||\,p_{l,0}\right)^{\frac{1}{2}}.$$

Since $p_{i,0}(x) = \sum_k \frac{w_{i,k}Z_{i,k}}{\sum_j w_{i,j}Z_{i,j}}\pi_{i,k}(x)$ the ratio $r$ in the context of Lemma J.4 is given by $r = \frac{\frac{w_{l+1,k}Z_{l+1,k}}{\sum_j w_{l+1,j}Z_{l+1,j}}}{\frac{w_{l,k}Z_{l,k}}{\sum_j w_{l,j}Z_{l,j}}}$. Using the inductive hypothesis (**H**1) this can be upper bounded by $r \leq C_1^2$.

Therefore by Lemma F.3 and Lemma J.4,

$$\leq C_1\chi^2\Big(\pi_{l+1,k}\,||\,\pi_{l,k}\Big)^{\frac{1}{2}}\cdot \left(\frac{\chi^2\big(\nu_0(x,i)||p(x,i)\big)\cdot C_{PI}\big(p_0(x,i)\big)}{\alpha_0\omega_0^l\cdot T}\right)^{\frac{1}{2}}.$$

Together this yields,

$$\mathbb{E}_{p_{l,0}}\left[\frac{p_{l+1,0}}{p_{l,0}}\cdot \frac{\hat{p}_{l,T}\frac{p_{l,0}}{\omega^l p_l}}{p_{l,0}}\right]$$

$$\geq \int_\Omega \hat{p}_{l,T}\frac{p_{l,0}}{\omega^l p_l}dx - C_1\chi^2\Big(\pi_{l+1,k}\,||\,\pi_{l,k}\Big)^{\frac{1}{2}}\cdot \left(\frac{\chi^2\big(\nu_0(x,i)||p(x,i)\big)\cdot C_{PI}\big(p_0(x,i)\big)}{\alpha_0\omega_0^l\cdot T}\right)^{\frac{1}{2}}.$$

Let $\Delta_{C_1} = \left(\frac{C_1^2\cdot \chi^2\big(\nu_0(x,i)||p(x,i)\big)\cdot C_{PI}\big(p_0(x,i)\big)}{\alpha_0(\omega_0^L)^2\cdot T}\right)^{\frac{1}{2}}$ and $\Delta = \left(\frac{\chi^2\big(\nu_0(x,i)||p(x,i)\big)\cdot C_{PI}\big(p_0(x,i)\big)}{\alpha_0(\omega_0^L)^2\cdot T}\right)^{\frac{1}{2}}$ then by Lemma F.4,

$$\geq \frac{1}{2||\frac{\nu_0}{\pi_0}||_\infty} - \Delta - \chi^2\Big(\pi_{l+1,k}\,||\,\pi_{l,k}\Big)^{\frac{1}{2}}\cdot \Delta_{C_1}$$

$$\geq \frac{1}{2||\frac{\nu_0}{\pi_0}||_\infty} - \left(1 + \chi^2\Big(\pi_{l+1,k}\,||\,\pi_{l,k}\Big)^{\frac{1}{2}}\right)\cdot \Delta_{C_1}.$$

$$\square$$

**Lemma G.9.** *Let Assumptions 1.1 hold. Then*

$$\frac{\mathbb{E}_{\hat{p}_T(x,i)}\big[\big(\frac{\bar{\pi}_{l+1,k}}{\tilde{p}(x,l)}\big)^2 I\{i=l\}\big]}{\mathbb{E}_{\hat{p}_T(x,i)}\big[\frac{\bar{\pi}_{l+1,k}}{\tilde{p}(x,i)}I\{i=l\}\big]^2} \le \frac{\big\|\frac{\nu_0(x,i)}{p(x,i)}\big\|_{L^\infty}}{C_B^2}\frac{MC_1}{c_{tilt}\omega^l}\Big(\chi^2\big(\bar{\pi}_{l+1,k}/\bar{Z}_{l+1,k} \,\|\, \bar{\pi}_{l,k}/\bar{Z}_{l,k}\big)-1\Big)$$

*with* $C_B = c_{tilt}\bigg(\frac{1}{2\|\frac{\nu_0(x,i)}{p_0(x,i)}\|_\infty} - \Big(1+\chi^2\Big(\pi_{l+1,k}\,\|\,\pi_{l,k}\Big)^{\frac{1}{2}}\Big)\cdot\Delta\bigg)$ *and*

$$\frac{\mathbb{E}_{\hat{p}_T(x,i)}\big[\big(\frac{\tilde{p}_{l+1}(x)}{\tilde{p}(x,l)}\big)^2 I\{i=l\}\big]}{\mathbb{E}_{\hat{p}_T(x,i)}\big[\frac{\tilde{p}_{l+1}(x)}{\tilde{p}(x,i)}I\{i=l\}\big]^2} \le \frac{\big\|\frac{\nu_0(x,i)}{p(x,i)}\big\|_{L^\infty}}{C_{B'}^2}\frac{C_1^2}{c_{tilt}^2\omega^l}\big(\chi^2\big(\bar{\pi}_{l+1,k}/\bar{Z}_{l+1,k} \,\|\, \bar{\pi}_{l,k}/\bar{Z}_{l,k}\big)-1\big)$$

*with* $C_{B'} = c_{tilt}\cdot\bigg(\frac{1}{2\|\frac{\nu_0(x,i)}{p_0(x,i)}\|_\infty} - \Big(1+\chi^2\Big(\pi_{l+1,k}\,\|\,\pi_{l,k}\Big)^{\frac{1}{2}}\Big)\cdot\Delta_{C_1}\bigg).$

*Proof.* For any function $f$,

$$\mathbb{E}_{\hat{p}_T}\Big[f(x,i)I\{i=l\}\Big] = \sum_i \int_\Omega \frac{\hat{p}_T(x,i)}{p(x,i)}p(x,i)f(x,i)I\{i=l\}dx$$

$$\le \Big\|\frac{\hat{p}_T(x,i)}{p(x,i)}\Big\|_{L^\infty}\mathbb{E}_p\Big[f(x,i)I\{i=l\}\Big] \le \Big\|\frac{\nu_0(x,i)}{p(x,i)}\Big\|_{L^\infty}\mathbb{E}_p\Big[f(x,i)I\{i=l\}\Big].$$

By Lemma G.6,

$$\mathbb{E}_{\hat{p}_T}\big[\frac{\bar{\pi}_{l+1,k}}{\tilde{p}(x,l)}I\{i=l\}\big]^2 \ge C_B^2\cdot\mathbb{E}_p\big[\frac{\bar{\pi}_{l+1,k}(x)}{\tilde{p}(x,l)}I\{i=l\}\big]^2.$$

Therefore,

$$\frac{\mathbb{E}_{\hat{p}_T(x,i)}\big[\big(\frac{\bar{\pi}_{l+1,k}}{\tilde{p}(x,l)}\big)^2 I\{i=l\}\big]}{\mathbb{E}_{\hat{p}_T(x,i)}\big[\frac{\bar{\pi}_{l+1,k}}{\tilde{p}(x,i)}I\{i=l\}\big]^2} \le \frac{\big\|\frac{\nu_0(x,i)}{p(x,i)}\big\|_{L^\infty}}{C_B^2}\cdot\frac{\mathbb{E}_p\big[\big(\frac{\bar{\pi}_{l+1,k}}{\tilde{p}(x,l)}\big)^2 I\{i=l\}\big]}{\mathbb{E}_p\big[\frac{\bar{\pi}_{l+1,k}}{\tilde{p}(x,i)}I\{i=l\}\big]^2}.$$

Further simplification yields

$$\frac{\mathbb{E}_p\big[\big(\frac{\bar{\pi}_{l+1,k}}{\tilde{p}(x,l)}\big)^2 I\{i=l\}\big]}{\mathbb{E}_p\big[\frac{\bar{\pi}_{l+1,k}}{\tilde{p}(x,i)}I\{i=l\}\big]^2} = \frac{\int_\Omega \omega^l p_l(x)\Big(\frac{\bar{\pi}_{l+1,k}}{r_l\tilde{p}_l(x)}\Big)^2 dx}{\Big(\int_\Omega \omega^l p_l(x)\frac{\bar{\pi}_{l+1,k}}{r_l\tilde{p}_l(x)}dx\Big)^2} = \frac{1}{\omega^l}\int_\Omega \frac{(\pi_{l+1,k}/\bar{Z}_{l+1,k})^2}{p_l(x)}dx.$$

Since $p_l(x) = \frac{1}{Z_l}\sum_k \pi(x)w_{l,k}q_l(x-x_k)$, by Lemma J.4 and Corollary G.2

$$\frac{1}{\omega^l}\int_\Omega \frac{(\pi_{l+1,k}/\bar{Z}_{l+1,k})^2}{p_l(x)}dx = \frac{1}{\omega^l}\big(\chi^2(\bar{\pi}_{l+1,k}/\bar{Z}_{l+1,k} \,\|\, p_l)-1\big)$$

$$\le \frac{1}{\omega^l}\cdot\frac{\sum_{k'} w_{l,k'}\bar{Z}_{l,k'}}{w_{l,k}\bar{Z}_{l,k}}\Big(\chi^2\big(\bar{\pi}_{l+1,k}/\bar{Z}_{l+1,k} \,\|\, \bar{\pi}_{l,k}/\bar{Z}_{l,k}\big)-1\Big)$$

$$\le \frac{MC_1}{c_{tilt}\omega^l}\Big(\chi^2\big(\bar{\pi}_{l+1,k}/\bar{Z}_{l+1,k} \,\|\, \bar{\pi}_{l,k}/\bar{Z}_{l,k}\big)-1\Big).$$

Similarly, by Lemma G.8,

$$\mathbb{E}_{\hat{p}_T}\big[\frac{\tilde{p}_{l+1}(x)}{\tilde{p}(x,l)}I\{i=l\}\big]^2 \ge C_{B'}^2\cdot\mathbb{E}_p\big[\frac{\tilde{p}_{l+1}(x)}{\tilde{p}(x,l)}I\{i=l\}\big]^2$$

where $C_{B'} = c_{tilt}\cdot\bigg(\frac{1}{2\|\frac{\nu_0(x,i)}{p_0(x,i)}\|_\infty} - \Big(1+\chi^2\Big(\pi_{l+1,k}\,\|\,\pi_{l,k}\Big)^{\frac{1}{2}}\Big)\cdot\Delta_{C_1}\bigg).$

Therefore,

$$\frac{\mathbb{E}_{\hat{p}_T(x,i)}\left[\left(\frac{\tilde{p}_{l+1}(x)}{\tilde{p}(x,l)}\right)^2 I\{i=l\}\right]}{\mathbb{E}_{\hat{p}_T(x,i)}\left[\frac{\tilde{p}_{l+1}(x)}{\tilde{p}(x,i)}I\{i=l\}\right]^2} \leq \frac{\left\|\frac{\nu_0(x,i)}{p(x,i)}\right\|_{L^\infty}}{C_{B'}^2} \cdot \frac{\mathbb{E}_p\left[\left(\frac{\tilde{p}_{l+1}(x)}{\tilde{p}(x,l)}\right)^2 I\{i=l\}\right]}{\mathbb{E}_p\left[\frac{\tilde{p}_{l+1}(x)}{\tilde{p}(x,i)}I\{i=l\}\right]^2}.$$

Further simplification yields

$$\frac{\mathbb{E}_p\left[\left(\frac{\tilde{p}_{l+1}(x)}{\tilde{p}(x,l)}\right)^2 I\{i=l\}\right]}{\mathbb{E}_p\left[\frac{\tilde{p}_{l+1}(x)}{\tilde{p}(x,i)}I\{i=l\}\right]^2} = \frac{\int_\Omega \omega^l p_l(x)\left(\frac{\tilde{p}_{l+1}(x)}{r_l \tilde{p}_l(x)}\right)^2 dx}{\left(\int_\Omega \omega^l p_l(x)\frac{\tilde{p}_{l+1}(x)}{r_l \tilde{p}_l(x)}dx\right)^2} = \frac{\frac{\omega^l}{r_l^2}\int_\Omega \frac{p_{l+1}(x)^2}{p_l(x)}dx}{\frac{(\omega^l)^2}{r_l^2}\left(\int_\Omega p_l(x)\frac{p_{l+1}(x)}{p_l(x)}dx\right)^2} = \frac{1}{\omega^l}\int_\Omega \frac{p_{l+1}(x)^2}{p_l(x)}dx.$$

Since $p_l(x) = \frac{1}{Z_l}\sum_k \pi(x)w_{l,k}q_l(x-x_k)$, by Lemma J.4 and Corollary G.2,

$$\frac{1}{\omega^l}\int_\Omega \frac{p_{l+1}(x)^2}{p_l(x)}dx \leq \frac{\frac{w_{l+1,k}\bar{Z}_{l+1,k}}{\sum_{k'} w_{l+1,k'}\bar{Z}_{l+1,k'}}}{\frac{w_{l,k}\bar{Z}_{l,k}}{\sum_{k'} w_{l,k'}\bar{Z}_{l,k'}}} \cdot \frac{1}{\omega^l}\left(\chi^2(\bar{\pi}_{l+1,k}/\bar{Z}_{l+1,k} \,\|\, \bar{\pi}_{l,k}/\bar{Z}_{l,k}) - 1\right)$$

$$\leq \frac{C_1^2}{c_{tilt}^2 \omega^l}\left(\chi^2(\bar{\pi}_{l+1,k}/\bar{Z}_{l+1,k} \,\|\, \bar{\pi}_{l,k}/\bar{Z}_{l,k}) - 1\right).$$

$\square$

## G.2 Mixing time bounds

**Lemma G.10.** *Given Assumptions 3.1, let* $\tilde{\pi}_0(x,i) = \sum_{j=1}^l \hat{r}_j^{(l)}\sum_{k=1}^M w_{jk}Z_{jk}\frac{\alpha_k \pi_k(x)q_j(x-x_k)}{Z_{jk}}I\{i = j\}$. *In the setting of Theorem E.1, we let* $\bar{\pi}_0$ *be the projected chain so that* $\bar{\pi}_0\big((i,j)\big) \propto \hat{r}_i^{(l)}w_{i,j}Z_{i,j}$. *Then*

$$C_{PI}\big(\pi_0(x,i)\big) = O\left(\frac{CMr \cdot l^2}{\gamma \cdot \lambda}\right)$$

*where* $r = \frac{\max_{i<i'\leq l}\bar{\pi}_0\big((i',j)\big)}{\min\left(\bar{\pi}_0\big((i,j)\big),\bar{\pi}_0\big((i-1,j)\big)\right)}$ *and* $C = \max_{ij} C_{ij}$. *When applying the inductive hypothesis*

$$C_{PI}\big(\pi_0(x,i)\big) = O\left(\frac{C_1^3 C_2 CM \cdot l^3}{c_{tilt}\gamma \cdot \lambda}\right).$$

*Proof.* First we note that by theorem E.1 we have that

$$C_{PI}(\pi_0(x,i)) \leq \max\left\{C(1 + (6M+12)\bar{C}), \frac{6M\bar{C}}{\gamma}, \frac{12\bar{C}}{\lambda}\right\}.$$

We are free to choose the constants $\gamma$ and $\lambda$ and $C$ is the maximum of the local Poincaré constants. Therefore it is left to bound $\bar{C}$, the Poincaré constant of the projected chain.

We show this using the canonical path method, Lemma J.7. Given the projected chain of the ST process there are two types of distinct edges. First consider the edge $e = \big((1,j),(1,k)\big)$, we have that

$$l(e) = \frac{1}{\bar{\pi}_0\big((i,j)\big)P\big((1,j),(1,k)\big)}\sum_{e\in\Gamma_{x,y}}\bar{\pi}_0(x)\bar{\pi}_0(y)|\gamma_{x\to y}|.$$

Given the definiton of $P\big((1,j),(1,k)\big)$ and our lower bounds in Lemma J.5 and Assumptions 3.1,

$$\leq \frac{O(1)}{\min\left(\bar{\pi}_0\big((1,j)\big),\bar{\pi}_0\big((1,k)\big)\right)}\sum_{e\in\Gamma_{x,y}}\bar{\pi}_0(x)\bar{\pi}_0(y)|\gamma_{x\to y}|.$$

Moreover, the longest path in our projected chain is from $x = (l, k)$ to $y = (l, j)$ and is of length $2l - 1$ therefore

$$\leq \frac{(2l-1)O(1)}{\min\left(\bar{\pi}_0\big((1,j)\big), \bar{\pi}_0\big((1,k)\big)\right) \cdot} \sum_{e \in \Gamma_{x,y}} \bar{\pi}_0(x)\bar{\pi}_0(y)$$

$$\leq \frac{(2l-1)O(1)}{\min\left(\bar{\pi}_0\big((1,j)\big), \bar{\pi}_0\big((1,k)\big)\right)} \sum_{i=1}^{l} \bar{\pi}_0\big((i,j)\big) \sum_{i'=1}^{l} \bar{\pi}_0\big((i',k)\big)$$

$$\leq \frac{\max_i \bar{\pi}_0\big((i,j)\big)}{\min\left(\bar{\pi}_0\big((1,j)\big), \bar{\pi}_0\big((1,k)\big)\right)} l(2l-1)O(1).$$

Now consider the second type of edge $e = \big((i,j), (i-1,j)\big)$, we have that

$$l(e) = \frac{1}{\bar{\pi}_0\big((i,j)\big) P\big((i,j), (i-1,j)\big)} \sum_{e \in \Gamma_{x,y}} \bar{\pi}_0(x)\bar{\pi}_0(y)|\gamma_{x \to y}|.$$

Given the definiton of $P\big((1,j), (1,k)\big)$ and our lower bounds in Lemma J.5 and Assumptions 3.1,

$$\leq \frac{O(1)}{\min\left(\bar{\pi}_0\big((i,j)\big), \bar{\pi}_0\big((i-1,j)\big)\right)} \sum_{e \in \Gamma_{x,y}} \bar{\pi}_0(x)\bar{\pi}_0(y)|\gamma_{x \to y}|.$$

Moreover, the longest path in our projected chain is from $x = (l, k)$ to $y = (l, j)$ and is of length $2l - 1$ therefore

$$\leq \frac{(2l-1)O(1)}{\min\left(\bar{\pi}_0\big((i,j)\big), \bar{\pi}_0\big((i-1,j)\big)\right) \cdot} \sum_{e \in \Gamma_{x,y}} \bar{\pi}_0(x)\bar{\pi}_0(y)$$

$$\leq \frac{(2l-1)O(1)}{\min\left(\bar{\pi}_0\big((i,j)\big), \bar{\pi}_0\big((i-1,j)\big)\right)} \sum_{i'=i+1}^{l} \bar{\pi}_0\big((i',j)\big) \left( \sum_{i'=1}^{i-1} \bar{\pi}_0\big((i',j)\big) + \sum_{i'=1}^{l} \bar{\pi}_0\big((i',k)\big) \right)$$

$$\leq \frac{\max_{i<i'\leq l} \bar{\pi}_0\big((i',j)\big)}{\min\left(\bar{\pi}_0\big((i,j)\big), \bar{\pi}_0\big((i-1,j)\big)\right)} l(2l-1)O(1).$$

In the case of induction, by letting $\bar{\pi}_0\big((i,j)\big) \propto \hat{r}_i^{(l)} w_{i,j} Z_{i,j}$ and applying Lemma G.4 we have that

$$\frac{\max_{i<i'\leq l} \bar{\pi}_0\big((i',j)\big)}{\min\left(\bar{\pi}_0\big((i,j)\big), \bar{\pi}_0\big((i-1,j)\big)\right)} \leq \frac{lC_1^3 C_2}{c_{tilt}}.$$

$\square$

**Lemma G.11.** *Given assumptions 1.1, 3.1, G.1 and* $\Delta = \left( \frac{\chi^2\left(\nu_0(x,i)||p(x,i)\right) \cdot C_{PI}\left(p_0(x,i)\right)}{\alpha_0(\omega_0^L)^2 \cdot T} \right)^{\frac{1}{2}}$ *then choosing*

$$T = \Omega\big(poly(l, M, C_1, C_2, C, \frac{1}{c_{tilt}}, \frac{1}{\lambda}, \frac{1}{\gamma}, \frac{1}{\delta_T})\big)$$

*yields* $\Delta \leq \delta_T$.

*Proof.* We have that,

$$\chi^2\big(\nu_0(x,i)||p(x,i)\big)^{\frac{1}{2}} \leq ||\frac{\nu_0(x,i)}{p(x,i)}||_\infty - 1 \leq ||\frac{\nu_0(x,i)}{p_0(x,i)}||_\infty.$$

By Assumptions 3.1

$$||\frac{\nu_0(x,i)}{p(x,i)}||_\infty \le U.$$

By Lemma G.10 with $\tilde{p}_0(x,i) = \hat{r}_l^{(l)} w_{l,k} \tilde{\pi}_{l,k}(x) I\{i = l\} + \sum_{j=1}^{l-1} \hat{r}_j^{(l)} \tilde{p}_{j0}(x) I\{j = i\}$ we have that,

$$C_{PI}(p_0(x,i)) = O(\frac{C_1^2 C_2 C M \cdot l^2}{c_{tilt} \gamma \cdot \lambda}).$$

For $\tilde{p}_0(x,i) = \hat{r}_l^{(l)} w_{l,k} \tilde{\pi}_{l,k}(x) I\{i = l\} + \sum_{j=1}^{l-1} \hat{r}_j^{(l)} \tilde{p}_{j0}(x) I\{j = i\}$ and $\tilde{p}(x,i) = \sum_{j=1}^{l} \hat{r}_j^{(l)} p_j(x) I\{j = i\}$ with $\alpha_0$ defined so that $p(x,i) = \alpha_0 p_0(x,i) + (1 - \alpha_0) p_1(x,i)$ we have that

$$\alpha_0 = \frac{\sum_{i=1}^{l} \int_\Omega \tilde{p}_0(x,i) dx}{\sum_{i=1}^{l} \int_\Omega \tilde{p}(x,i) dx}$$

$$\ge \frac{\hat{r}_l^{(l)} Z_{l0} + \sum_{i=1}^{l-1} \hat{r}_i^{(l)} Z_{i0} dx}{\sum_{i=1}^{l} \hat{r}_i^{(l)} Z_i dx}$$

$$\ge c_{tilt} \frac{\hat{r}_l^{(l)} Z_l + \sum_{i=1}^{l-1} \hat{r}_i^{(l)} Z_i dx}{\sum_{i=1}^{l} \hat{r}_i^{(l)} Z_i dx}$$

$$= c_{tilt}.$$

Lastly, for $\tilde{p}_0(x,i) = \hat{r}_l^{(l)} w_{l,k} \tilde{\pi}_{l,k}(x) I\{i = l\} + \sum_{j=1}^{l-1} \hat{r}_j^{(l)} \tilde{p}_{j0}(x) I\{j = i\}$ making use of Lemma G.4 and Lemma G.3 we have that

$$\omega_0^l = \frac{\hat{r}_l^{(l)} w_{l,k} \int_\Omega \tilde{\pi}_{l,k}(x) dx}{\sum_{i=1}^{l} \hat{r}_i^{(l)} \sum_{k=1}^{M} w_{i,k} \int_\Omega \tilde{\pi}_{i,k}(x) dx}$$

$$= \frac{\hat{r}_l^{(l)} w_{l,k} Z_{l,k}}{\sum_{i=1}^{l} \hat{r}_i^{(l)} \sum_{k=1}^{M} w_{i,k} Z_{i,k}}$$

$$\ge \frac{1}{\sum_{i=1}^{l} \sum_{k=1}^{M} \frac{\hat{r}_i^{(l)} w_{i,k} Z_{i,k}}{\hat{r}_l^{(l)} w_{l,k'} Z_{l,k'}}}$$

$$\ge \frac{c_{tilt}}{l \cdot Ml \cdot C_1^3 C_2}.$$

Putting all of the bounds together we have that

$$\Delta = \left( \frac{\chi^2(\nu_0(x,i) || p(x,i)) \cdot C_{PI}(p_0(x,i))}{\alpha_0 (\omega_0^L)^2 \cdot T} \right)^{\frac{1}{2}} \le \left( \frac{O(\frac{C_1^3 C_2 C M \cdot l^3}{c_{tilt} \gamma \cdot \lambda}) \cdot l^2 \cdot M^2 \cdot C_1^4 C_2^2}{c_{tilt}^3 \cdot T} \right).$$

Therefore choosing

$$T = \Omega\left( \frac{l^6 M^3 C C_1^7 C_2^3}{\delta_T \cdot \gamma \cdot \lambda \cdot c_{tilt}^6} \right)$$

with an appropriate constant yields $\Delta \le \delta_T$. $\qquad\square$

### G.3 PROOF OF INDUCTION STEP

In this subsection it is shown how estimating weights by Algorithm 3 maintains modal and level balance. Its important to note that Algorithm 3 has three separate Monte Carlo estimates, it'll be shown how each one is used to maintain level balance. First, Algorithm 3 estimates the modal weights at the next temperature level, the analysis of this component corresponds to Theorem G.13. Then, after estimating the modal weights, the level weight is estimated, which corresponds to Theorem G.15. Lastly, the algorithm is re-ran and samples are collected to re-adjust level weights, this corresponds to the analysis in Theorem G.17.

We show in the following lemma that the constant $R$ from Lemma G.5 is bounded.

**Lemma G.12.** *Let Assumptions 1.1 and Assumptions 3.1 hold. Then for $f = \frac{\bar{\pi}_{l,k}(x)}{\tilde{p}(x,l)}$ and $f = \frac{\tilde{p}_{l+1}(x)}{\tilde{p}(x,l)}$,*

$$\frac{\mathbb{E}_{\hat{p}_T}\left[f^2\right]}{\mathbb{E}_{\hat{p}_T}\left[f\right]^2} \leq R,$$

*with $R = \text{poly}(l, M, C_1, C_2, \frac{1}{c_{tilt}}, U)$.*

*Proof.* By Lemma G.9,

$$\frac{\mathbb{E}_{\hat{p}_T(x,i)}\left[\left(\frac{\bar{\pi}_{l+1,k}}{\tilde{p}(x,l)}\right)^2 I\{i=l\}\right]}{\mathbb{E}_{\hat{p}_T(x,i)}\left[\frac{\bar{\pi}_{l+1,k}}{\tilde{p}(x,i)} I\{i=l\}\right]^2} \leq \frac{\left\|\frac{\nu_0(x,i)}{p(x,i)}\right\|_{L^\infty}}{C_B^2} \frac{MC_1}{c_{tilt}\omega^l}\left(\chi^2\left(\bar{\pi}_{l+1,k}/\bar{Z}_{l+1,k} \,\|\, \bar{\pi}_{l,k}/\bar{Z}_{l,k}\right) - 1\right)$$

with $C_B = c_{tilt}\left(\frac{1}{2\|\frac{\nu_0(x,i)}{p_0(x,i)}\|_\infty} - \left(1 + \chi^2\left(\pi_{l+1,k} \,\|\, \pi_{l,k}\right)^{\frac{1}{2}}\right) \cdot \Delta\right)$ and

$$\frac{\mathbb{E}_{\hat{p}_T(x,i)}\left[\left(\frac{\tilde{p}_{l+1}(x)}{\tilde{p}(x,l)}\right)^2 I\{i=l\}\right]}{\mathbb{E}_{\hat{p}_T(x,i)}\left[\frac{\tilde{p}_{l+1}(x)}{\tilde{p}(x,i)} I\{i=l\}\right]^2} \leq \frac{\left\|\frac{\nu_0(x,i)}{p(x,i)}\right\|_{L^\infty}}{C_{B'}^2} \frac{C_1^2}{c_{tilt}^2\omega^l}\left(\chi^2(\bar{\pi}_{l+1,k}/\bar{Z}_{l+1,k} \,\|\, \bar{\pi}_{l,k}/\bar{Z}_{l,k}) - 1\right)$$

with $C_{B'} = c_{tilt} \cdot \left(\frac{1}{2\|\frac{\nu_0(x,i)}{p_0(x,i)}\|_\infty} - \left(1 + \chi^2\left(\pi_{l+1,k} \,\|\, \pi_{l,k}\right)^{\frac{1}{2}}\right) \cdot \Delta_{C_1}\right)$.

As made explicit in Lemma G.11, with $T = \Omega\left(poly(l, M, C_1, C_2, C, \frac{1}{c_{tilt}}, \frac{1}{\lambda}, \frac{1}{\gamma}, \frac{1}{\delta_T})\right)$ the $C_B$ and $C_{B'}$ terms can be lower bounded by $\frac{c_{tilt}}{\|\frac{\nu_0(x,i)}{p_0(x,i)}\|_\infty}$. Moreover, using that $\|\frac{\nu_0(x,i)}{p(x,i)}\|_\infty \leq \|\frac{\nu_0(x,i)}{p_0(x,i)}\|_\infty$ yields the final upper bound.

Since $\omega^l = \frac{\hat{r}_l^{(l)} Z_l}{\sum_k \hat{r}_k^{(l)} Z_k}$,

$$\frac{1}{\omega^l} = \frac{\sum_k \hat{r}_k^{(l)} Z_k}{\hat{r}_l^{(l)} Z_l} = \frac{\hat{r}_1^{(l)} Z_1 + \sum_{k=2}^l \hat{r}_k^{(l)} Z_k}{\hat{r}_l^{(l)} Z_l} = \frac{l \cdot C_2 r_1^{(l)} Z_1 + \sum_{k=2}^l r_k^{(l)} Z_k}{r_l^{(l)} Z_l} \leq 2 \cdot l \cdot C_2^2.$$

Using $\left\|\frac{\nu_0(x,i)}{p(x,i)}\right\|_{L^\infty} \leq U$ and $\chi^2\left(\bar{\pi}_{l+1,k}/\bar{Z}_{l+1,k} \,\|\, \bar{\pi}_{l,k}/\bar{Z}_{l,k}\right) = O(1)$ by Assumptions 3.1 then combining the bounds yields

$$\frac{\left\|\frac{\nu_0(x,i)}{p(x,i)}\right\|_{L^\infty}}{C_B^2} \frac{MC_1}{c_{tilt}\omega^l}\left(\chi^2\left(\bar{\pi}_{l+1,k}/\bar{Z}_{l+1,k} \,\|\, \bar{\pi}_{l,k}/\bar{Z}_{l,k}\right) - 1\right) \leq \frac{2lMC_1C_2^2U^3}{c_{tilt}^3} \cdot O(1)$$

$$\frac{\left\|\frac{\nu_0(x,i)}{p(x,i)}\right\|_{L^\infty}}{C_{B'}^2} \frac{C_1^2}{c_{tilt}^2\omega^l}\left(\chi^2(\bar{\pi}_{l+1,k}/\bar{Z}_{l+1,k} \,\|\, \bar{\pi}_{l,k}/\bar{Z}_{l,k}) - 1\right) \leq \frac{2lC_1^2C_2^2U^3}{c_{tilt}^3} \cdot O(1).$$

$\square$

**Theorem G.13.** $\boxed{H1(l+1)}$ *There is $C_1 = \text{poly}\left(\frac{U}{c_{tilt}}\right)$ and $R = \text{poly}(l, M, C_1, C_2, \frac{1}{c_{tilt}}, U)$ such that the following holds: Suppose that H1(l) holds with $C_1$. Then Algorithm 3, running the continuous process for time $T$ time and obtaining $N$ samples, with*

$$T = \Omega\left(\text{poly}(l, M, \tilde{C}, U, \frac{1}{c_{tilt}}, \frac{1}{\lambda}, \frac{1}{\gamma}, \frac{1}{\delta_T})\right)$$

$$N = \Omega\left(\frac{R}{\delta}\right),$$

*returns weights such that with probability $1 - \delta$, H1(l+1) holds with the same $C_1$. The key choice in Algorithm 3 is the weighting*

$$w_{l+1,k} = \frac{1}{\frac{1}{N} \sum_{j=1}^{N} \frac{\bar{\pi}_{l+1,k}(x_j)}{\hat{p}(x_j, i_j)} I\{i_j = l\}}.$$

*Proof.* We have with probability $1 - \delta$ that letting $c_A = 1 - \sqrt{\frac{R}{N\delta}}$, $c_B = c_{tilt}\left(\frac{1}{2\|\frac{\nu_0(x,i)}{p_0(x,i)}\|_\infty} - \left(1 + \chi^2\left(\pi_{l+1,k} \| \pi_{l,k}\right)^{\frac{1}{2}}\right) \cdot \Delta\right)$, $C_A = 1 + \sqrt{\frac{R}{N\delta}}$, $C_B = \left\|\frac{\nu_0(x,i)}{p(x,i)}\right\|_{L^\infty}$ and $c_C = \frac{1}{c_{tilt}}$,

$$\frac{\frac{1}{N} \sum_{j=1}^{N} \frac{\bar{\pi}_{l+1,k}(x_j)}{\hat{p}(x_j, i_j)} I\{i_j = l\}}{\frac{Z_{l+1,k}}{Z}} = \frac{\frac{1}{N} \sum_{j=1}^{N} \frac{\bar{\pi}_{l+1,k}(x_j)}{\hat{p}(x_j, i_j)} I\{i_j = l\}}{\mathbb{E}_{\hat{p}_t}\left[\frac{\bar{\pi}_{l+1,k}}{\hat{p}(x,i)} I\{i = l\}\right]} \cdot \frac{\mathbb{E}_{\hat{p}_t}\left[\frac{\bar{\pi}_{l+1,k}}{\hat{p}(x,i)} I\{i = l\}\right]}{\mathbb{E}_p\left[\frac{\bar{\pi}_{l+1,k}}{\hat{p}(x,i)} I\{i = l\}\right]} \cdot \frac{\mathbb{E}_p\left[\frac{\bar{\pi}_{l+1,k}}{\hat{p}(x,i)} I\{i = l\}\right]}{Z_{l+1,k}/Z}$$

$$\in [c_A c_B, C_A C_B C_C]$$

by applying Lemma G.5, Lemma G.6 and Lemma G.7 to the three terms respectively.

Choosing $N = \Omega\left(\frac{R}{\delta}\right)$ reduces $\sqrt{\frac{R}{N\delta}}$ to a constant. Lemma G.12 provides a bound for R. Let $\Delta = \left(\frac{\chi^2\left(\nu_0(x,i)\|p(x,i)\right) \cdot C_{PI}\left(p_0(x,i)\right)}{\alpha_0(\omega_0^L)^2 \cdot T}\right)^{\frac{1}{2}}$ then by taking the ratio of the two Monte Carlo estimates yields that we can take

$$C = \max\left\{C_A C_B, \frac{1}{c_A c_B}, c_C\right\} \leq \frac{\left\|\frac{\nu_0(x,i)}{\pi(x,i)}\right\|_{L^\infty} \cdot \frac{1}{c_{tilt}}}{c_{tilt}\left(\frac{1}{2\|\frac{\nu_0}{\pi_0}\|_\infty} - \left(1 + \chi^2\left(\pi_{l+1,k} \| \pi_{l,k}\right)^{\frac{1}{2}}\right) \cdot \Delta\right)}.$$

The rest follows from Assumptions 3.1 and Lemma G.11 which yield

$$\chi^2\left(\pi_{l+1,k} \| \pi_{l,k}\right)^{\frac{1}{2}} = O(1) \text{ and } \Delta = O(\delta),$$

respectively. This yields

$$C \leq \frac{\left\|\frac{\nu_0(x,i)}{\pi(x,i)}\right\|_{L^\infty} \cdot \frac{1}{c_{tilt}}}{c_{tilt}\left(\frac{1}{2\|\frac{\nu_0}{\pi_0}\|_\infty} - \left(1 + O(1)\right) \cdot \delta\right)}$$

By choosing $T$ so that $\delta$ is negligible

$$= \frac{2(1 + \delta)}{c_{tilt}^2(1 - \delta)} \left\|\frac{\nu_0(x,i)}{\pi(x,i)}\right\|_{L^\infty}^2$$

Which by Assumptions G.1,

$$C \leq \frac{4}{c_{tilt}^2} U^2 = \text{poly}\left(\frac{U}{c_{tilt}}\right).$$

$\square$

In order to simplify the proof of H2(l+1) we offload some of the work to the following lemma (G.14). This Lemma combines the bounds from the previous lemmas, Lemma G.5 and Lemma G.8, to find an upper and lower bound on $\frac{\hat{r}_k^{(l)} Z_k}{\hat{r}_{l+1}^{(l)} Z_{l+1}}$. Then the theorem that follows, Theorem G.15, details the run time $T$ and number of samples $N$ required to guarantee $\frac{\hat{r}_k^{(l)} Z_k}{\hat{r}_{l+1}^{(l)} Z_{l+1}} \in [\frac{1}{C_2}, C_2]$. Its important to note that Theorem G.15 only guarantees the exist of one $k \in [1, l]$ that maintains level balance.

**Lemma G.14.** *Given Assumptions 1.1 and Assumptions 3.1. Let* $\Delta_{C_1} = \left( \frac{C_1^2 \cdot \chi^2 \left( \nu_0(x,i) || \pi(x,i) \right) \cdot C_{PI} \left( \pi_0(x,i) \right)}{\alpha_0 (\omega_0^L)^2 \cdot T} \right)^{\frac{1}{2}}$ *and choose* $\hat{r}_{l+1}^{(l)} \propto \frac{1}{\frac{1}{N} \sum_{i=j}^{N} \frac{\tilde{p}_{l+1}(x_j)}{\tilde{p}(x_j, i_j)} I\{i_j = l\}}$, *with* $N = \Omega\left( \frac{R}{\delta} \right)$ *then with probability* $1 - \delta$

$$\frac{c_{tilt}}{l^2 C_2^2}(1-\delta)\left( \frac{1}{2||\frac{\nu_0(x,i)}{p_0(x,i)}||_\infty} - \left( 1 + \chi^2 \left( \pi_{l+1,k} \,||\, \pi_{l,k} \right) \right)^{\frac{1}{2}} \cdot \Delta_{C_1} \right) \leq \frac{\hat{r}_k^{(l)} Z_k}{\hat{r}_{l+1}^{(l)} Z_{l+1}} \leq (1+\delta) \cdot \left\| \frac{\nu_0(x,i)}{p(x,i)} \right\|_{L^\infty}$$

*for all* $k \in [1, l]$.

*Proof.* By Lemma G.5, with $R$ as in Lemma G.12, and Lemma G.8 we have that

$$\frac{\frac{1}{N} \sum_{i=j}^{N} \frac{\tilde{p}_{l+1}(x_j)}{\tilde{p}(x_j, i_j)} I\{i_j = l\}}{Z_{l+1} \Big/ Z} = \frac{\frac{1}{N} \sum_{i=j}^{N} \frac{\tilde{p}_{l+1}(x_j)}{\tilde{p}(x_j, i_j)} I\{i_j = l\}}{\mathbb{E}_{\hat{p}_t}\left[ \frac{\tilde{p}_{l+1}(x)}{\tilde{p}(x,i)} I\{i = l\} \right]} \cdot \frac{\mathbb{E}_{\hat{p}_t}\left[ \frac{\tilde{p}_{l+1}(x)}{\tilde{p}(x,i)} I\{i = l\} \right]}{\mathbb{E}_{p}\left[ \frac{\tilde{p}_{l+1}(x)}{\tilde{p}(x,i)} I\{i = l\} \right]}$$

$$\leq (1 + \delta) \cdot \left\| \frac{\nu_0(x,i)}{p(x,i)} \right\|_{L^\infty} = U.$$

Similarly, we have that

$$\frac{\frac{1}{N} \sum_{i=j}^{N} \frac{\tilde{p}_{l+1}(x_j)}{\tilde{p}(x_j, i_j)} I\{i_j = l\}}{Z_{l+1} \Big/ Z} = \frac{\frac{1}{N} \sum_{i=j}^{N} \frac{\tilde{p}_{l+1}(x_j)}{\tilde{p}(x_j, i_j)} I\{i_j = l\}}{\mathbb{E}_{\hat{p}_t}\left[ \frac{\tilde{p}_{l+1}(x)}{\tilde{p}(x,i)} I\{i = l\} \right]} \cdot \frac{\mathbb{E}_{\hat{p}_t}\left[ \frac{\tilde{p}_{l+1}(x)}{\tilde{p}(x,i)} I\{i = l\} \right]}{\mathbb{E}_{p}\left[ \frac{\tilde{p}_{l+1}(x)}{\tilde{p}(x,i)} I\{i = l\} \right]}$$

$$\geq c_{tilt}(1 - \delta)\left( \frac{1}{2||\frac{\nu_0(x,i)}{p_0(x,i)}||_\infty} - \left( 1 + \chi^2 \left( \pi_{l+1,k} \,||\, \pi_{l,k} \right) \right)^{\frac{1}{2}} \cdot \Delta_{C_1} \right) = L.$$

Lastly, given that $\frac{Z_{l+1}}{Z} = \frac{Z_{l+1}}{\sum_{i=1}^{l} \hat{r}_i^{(l)} Z_i}$, we consider

$$\frac{\frac{1}{N} \sum_{i=j}^{N} \frac{\tilde{p}_{l+1}(x_j)}{\tilde{p}(x_j, i_j)} I\{i_j = l\}}{Z_{l+1} \Big/ Z} = \frac{\sum_{i=1}^{l} \hat{r}_i^{(l)} Z_i}{\frac{1}{\frac{1}{N} \sum_{i=j}^{N} \frac{\tilde{p}_{l+1}(x_j)}{\tilde{p}(x_j, i_j)} I\{i_j = l\}} Z_{l+1}} \in [L, U].$$

$$\implies \frac{\hat{r}_k^{(l)} Z_k}{\frac{1}{\frac{1}{N} \sum_{i=j}^{N} \frac{\tilde{p}_{l+1}(x_j)}{\tilde{p}(x_j, i_j)} I\{i_j = l\}} Z_{l+1}} \leq U \text{ for all } k \in [1, l].$$

By the pigeonhole principle there exists $k \in [1, l]$ such that

$$\frac{L}{l} \leq \frac{\hat{r}_k^{(l)} Z_k}{\frac{1}{\frac{1}{N} \sum_{i=j}^{N} \frac{\tilde{p}_{l+1}(x_j)}{\tilde{p}(x_j, i_j)} I\{i_j = l\}} Z_{l+1}}.$$

Lemma G.3 yields for all $k \in [1, l]$

$$\frac{L}{l^2 C_2^2} \leq \frac{\hat{r}_k^{(l)} Z_k}{\frac{1}{\frac{1}{N} \sum_{i=j}^{N} \frac{\tilde{p}_{l+1}(x_j)}{\tilde{p}(x_j, i_j)} I\{i_j = l\}} Z_{l+1}}.$$

$\square$

**Theorem G.15.** *There is* $C = l^2 C_2^2 \cdot O\left( \frac{U}{c_{tilt}} \right)$ *and* $R = poly(l, M, C_1, C_2, \frac{1}{c_{tilt}}, U)$ *such that the following holds: Suppose that* **H1**(l) *holds with* $C_1$ *and* **H2**(l) *holds with* $C_2$. *By running Algorithm 3, the continuous time process for time* $T$ *and obtaining* $N$ *samples with*

$$T = \Omega\left( poly(l, M, \tilde{C}, C_1, C_2, \frac{1}{c_{tilt}}, \frac{1}{\lambda}, \frac{1}{\gamma}, \frac{1}{\delta_T}) \right)$$

$$N = \Omega\left( \frac{R}{\delta} \right).$$

*Then there exists $k \in [1, l]$ s.t. with probability $1 - \delta$*

$$\frac{1}{C} \leq \frac{\hat{r}_k^{(l)} Z_k}{\hat{r}_{l+1}^{(l)} Z_{l+1}} \leq C.$$

*The choice in Algorithm 3 is the weighting*

$$\hat{r}_{l+1}^{(l)} = \frac{1}{\frac{1}{N} \sum_{i=j}^{N} \frac{\tilde{p}_{l+1}(x_j)}{\tilde{p}(x_j, i_j)} I\{i_j = l\}}.$$

*Proof.* By Lemma G.14 this reduces to finding the maximum between the upper bounds on

$$\frac{1}{\frac{c_{tilt}}{l^2 C_2^2}(1 - \delta)\left(\frac{1}{2||\frac{\nu_0(x,i)}{p_0(x,i)}||_\infty} - \left(1 + \chi^2\left(\pi_{l+1,k} \,||\, \pi_{l,k}\right)^{\frac{1}{2}}\right) \cdot \Delta_{C_1}\right)}$$

and

$$(1 + \delta) \cdot \left\|\frac{\nu_0(x,i)}{p(x,i)}\right\|_{L^\infty}.$$

The first term can be bounded as follows

$$\frac{1}{\frac{c_{tilt}}{l^2 C_2^2}(1 - \delta)\left(\frac{1}{2||\frac{\nu_0(x,i)}{p_0(x,i)}||_\infty} - \left(1 + \chi^2\left(\pi_{l+1,k} \,||\, \pi_{l,k}\right)^{\frac{1}{2}}\right) \cdot \Delta_{C_1}\right)}$$

$$\leq \frac{l^2 C_2^2}{c_{tilt}(1 - \delta)\left(\frac{1}{2||\frac{\nu_0(x,i)}{p_0(x,i)}||_\infty} - \left(1 + \chi^2\left(\pi_{l+1,k} \,||\, \pi_{l,k}\right)^{\frac{1}{2}}\right) \cdot \Delta_{C_1}\right)}.$$

Where $\Delta_{C_1}$ is $\Delta$ with an additional $C_1^2$ term in the numerator. However, as shown in Lemma G.11 the time is already polynomial in $C_1$ therefore the run time remains unchanged. This puts us in the same settings as Theorem G.13. Taking the bound in Theorem G.13 as greedy upper bound we get the same constant $C = O(\frac{U}{c_{tilt}})$ with an additional $l \cdot C_2$ scaling factor.

$\square$

Theorem G.15 shows that there exists $k \in [1, l]$ such that we have good level balance for $\frac{\hat{r}_k^{(l)} Z_k}{\hat{r}_{l+1}^{(l)} Z_{l+1}}$. However, in order to guarantee that the projected chain mixes well we require that the ratio $\frac{\hat{r}_k^{(l)} Z_k}{\hat{r}_j^{(l)} Z_j}$ be within a constant for all $j, k \in [1, l + 1]$. In order to prevent the constant from becoming exponentially bad with respect to the number of levels, after estimating the level weight $r_l^{(l)}$, we re-run the chain and keep count of the number of samples at each level and adjust accordingly. The following Lemma G.16 shows that the level approximation acquired by sampling is close to the true level weights. This Lemma is then used to prove the bound for **H2(l+1)** in Theorem G.15.

**Lemma G.16.** *Let Assumptions 1.1 and Assumptions 3.1 hold. Then*

$$\frac{\alpha c_{tilt}}{2\left\|\frac{\nu_0(x,i)}{p_{good}(x,i)}\right\|_{L^\infty}} - \alpha\Delta \leq \frac{\mathbb{E}_{\hat{p}^T(x,i)}\left[I\{l_1 = i\}\right]}{\mathbb{E}_{p(x,i)}\left[I\{l_1 = i\}\right]} \leq \left\|\frac{\nu_0(x,i)}{p(x,i)}\right\|_{L^\infty}$$

*with $\Delta = \left(\frac{\chi^2\left(\nu_0(x,i)||p(x,i)\right) \cdot C_{PI}\left(p(x,i)\right)}{\alpha \cdot T}\right)^{\frac{1}{2}}$.*

*Proof.* An upperbound is given by,

$$\frac{\mathbb{E}_{\hat{p}^T(x,i)}\left[I\{l_1=i\}\right]}{\mathbb{E}_{p(x,i)}\left[I\{l_1=i\}\right]} = \frac{\sum_i \int_\Omega \frac{\hat{p}_T(x,i)}{p(x,i)}p(x,i)I\{l_1=i\}}{\mathbb{E}_{p(x,i)}\left[I\{l_1=i\}\right]}$$

$$\leq \frac{\left\|\frac{\hat{p}_{l,T}(x,i)}{p_l(x,i)}\right\|_{L^\infty} \sum_i \int_\Omega I\{l_1=i\}p(x,i)dx}{\mathbb{E}_{p(x,i)}\left[I\{l_1=i\}\right]}$$

$$= \left\|\frac{\hat{p}_{l,T}(x,i)}{p(x,i)}\right\|_{L^\infty}$$

By contraction,

$$\leq \left\|\frac{\nu_0(x,i)}{p(x,i)}\right\|_{L^\infty}$$

To find a lower bound first note that

$$\frac{\mathbb{E}_{\hat{p}^T(x,i)}\left[I\{l_1=i\}\right]}{\mathbb{E}_{p(x,i)}\left[I\{l_1=i\}\right]} = \frac{\sum_i \int_\Omega \left(\alpha\hat{p}^T(x,i)\frac{p_{good}(x,i)}{p(x,i)} + \left(\hat{p}^T(x,i) - \alpha\hat{p}^T(x,i)\frac{p_{good}(x,i)}{p(x,i)}\right)\right)I\{l_1=i\}dx}{\mathbb{E}_{p(x,i)}\left[I\{l_1=i\}\right]}$$

since $\alpha p_{good}(x,i) + (1-\alpha)p_{bad}(x,i) = p(x,i)$

$$\geq \alpha\frac{\sum_i \int_\Omega \hat{p}^T(x,i)\frac{p_{good}(x,i)}{p(x,i)}I\{l_1=i\}dx}{\mathbb{E}_{p(x,i)}\left[I\{l_1=i\}\right]}$$

let $p_{good}(x,i)$ be the good part of distribution on the extended state space. Then consider,

$$\left|\sum_i \int_\Omega \left(\frac{\hat{p}^T(x,i)\frac{p_{good}(x,i)}{p(x,i)}\Big/Z}{p_{good}(x,i)} - 1\right)I\{l_1=i\}p_{good}(x,i)dx\right|$$

$$\leq \sum_i \int_\Omega \left|\frac{\hat{p}^T(x,i)\frac{p_{good}(x,i)}{p(x,i)}\Big/Z}{p_{good}(x,i)} - 1\right|I\{l_1=i\}p_{good}(x,i)dx$$

$$\leq \chi^2\left(\frac{\hat{p}^T(x,i)\frac{p_{good}(x,i)}{p(x,i)}}{\sum_i \int_\Omega \hat{p}^T(x,i)\frac{p_{good}(x,i)}{p(x,i)}dx} \,\|\, p_{good}(x,i)\right)^{\frac{1}{2}}.$$

$$\implies \sum_i \int_\Omega \frac{\hat{p}^T(x,i)\frac{p_{good}(x,i)}{p(x,i)}}{\sum_i \int_\Omega \hat{p}^T(x,i)\frac{p_{good}(x,i)}{p(x,i)}dx}I\{l_1=i\}dx \geq$$

$$\mathbb{E}_{p_{good}(x,i)}\left[I\{l_1=i\}\right] - \chi^2\left(\frac{\hat{p}^T(x,i)\frac{p_{good}(x,i)}{p(x,i)}}{\sum_i \int_\Omega \hat{p}^T(x,i)\frac{p_{good}(x,i)}{p(x,i)}dx} \,\|\, p_{good}(x,i)\right)^{\frac{1}{2}}.$$

Combining the two lower bounds we have that,

$$\frac{\mathbb{E}_{\hat{p}^T(x,i)}\left[I\{l_1=i\}\right]}{\mathbb{E}_{p(x,i)}\left[I\{l_1=i\}\right]} \geq \frac{\alpha Z\left(\mathbb{E}_{p_{good}(x,i)}\left[I\{l_1=i\}\right] - \chi^2\left(\frac{\hat{p}^T(x,i)\frac{p_{good}(x,i)}{p(x,i)}}{\sum_i \int_\Omega \hat{p}^T(x,i)\frac{p_{good}(x,i)}{p(x,i)}dx} \,\|\, p_{good}(x,i)\right)^{\frac{1}{2}}\right)}{\mathbb{E}_{p(x,i)}\left[I\{l_1=i\}\right]}$$

By Lemma F.2 and using assumptions 1.1 we get,

$$\geq \frac{\alpha}{2\left\|\frac{\nu_0(x,i)}{p_{good}(x,i)}\right\|_{L^\infty}}\left(c_{tilt} - \chi^2\left(\frac{\hat{p}^T(x,i)\frac{p_{good}(x,i)}{p(x,i)}}{\sum_i \int_\Omega \hat{p}^T(x,i)\frac{p_{good}(x,i)}{p(x,i)}dx} \;\|\; p_{good}(x,i)\right)^{\frac{1}{2}}\right).$$

Lastly by Lemma F.1 with $\Delta = \left(\frac{\chi^2\left(\nu_0(x,i)\|p(x,i)\right)\cdot C_{PI}\left(p(x,i)\right)}{\alpha\cdot T}\right)^{\frac{1}{2}}$,

$$\geq \frac{\alpha c_{tilt}}{2\left\|\frac{\nu_0(x,i)}{p_{good}(x,i)}\right\|_{L^\infty}} - \alpha\Delta$$

$\square$

**Theorem G.17.** $\boxed{H2(l+1)}$ *There is* $C_2 = \text{poly}\left(\frac{U}{c_{tilt}}\right)$ *and* $R = poly(l, M, C_1, C_2, \frac{1}{c_{tilt}}, U)$ *such that the following holds: Suppose that **H1**(l) and **H2**(l) holds with* $C_2$. *Then Algorithm 3, running the continuous process for time* $T$ *time and obtaining* $N$ *samples, with*

$$T = \Omega\big(\text{poly}(l, M, \tilde{C}, C_1, C_2, U, \frac{1}{c_{tilt}}, \frac{1}{\lambda}, \frac{1}{\gamma}, \frac{1}{\delta_T})\big)$$

$$N = \Omega\big(\frac{R}{\delta}\big),$$

*returns weights such that with probability* $1 - \delta$, ***H2**(l + 1) holds with the same* $C_2$. *The key choice of weights here are*

$$r_i^{(l+1)} = \hat{r}_i^{(l)} \Big/ \frac{1}{N}\sum_{j=1}^N I\{i_j = i\}.$$

*Proof.* Applying the definition of $r_i^{(l)}$ the quotient can be rewritten as

$$\frac{\int_\Omega r_{l_1}^{(l+1)}\tilde{p}_{l_1}(x)dx}{\int_\Omega r_{l_2}^{(l+1)}\tilde{p}_{l_2}(x)dx} = \frac{\int_\Omega r_{l_1}^{(l)}\tilde{p}_{l_1}(x)dx}{\int_\Omega r_{l_2}^{(l)}\tilde{p}_{l_2}(x)dx} \cdot \frac{\frac{1}{N}\sum_{j=1}^N I\{i_j = l_2\}}{\frac{1}{N}\sum_{j=1}^N I\{i_j = l_1\}}$$

$$= \frac{\mathbb{E}_{p(x,i)}\Big[I\{l_1 = i\}\Big]}{\mathbb{E}_{p(x,i)}\Big[I\{l_2 = i\}\Big]} \cdot \frac{\frac{1}{N}\sum_{j=1}^N I\{i_j = l_2\}}{\frac{1}{N}\sum_{j=1}^N I\{i_j = l_1\}}$$

$$= \frac{\frac{\frac{1}{N}\sum_{j=1}^N I\{i_j=l_2\}}{\mathbb{E}_{\hat{p}^T(x,i)}\Big[I\{i=l_2\}\Big]} \cdot \frac{\mathbb{E}_{\hat{p}^T(x,i)}\Big[I\{i=l_2\}\Big]}{\mathbb{E}_{p(x,i)}\Big[I\{i=l_2\}\Big]}}{\frac{\frac{1}{N}\sum_{j=1}^N I\{i_j=l_1\}}{\mathbb{E}_{\hat{p}^T(x,i)}\Big[I\{i=l_1\}\Big]} \cdot \frac{\mathbb{E}_{\hat{p}^T(x,i)}\Big[I\{i=l_1\}\Big]}{\mathbb{E}_{p(x,i)}\Big[I\{i=l_1\}\Big]}}.$$

Therefore it is sufficient to upper and lower bound $\dfrac{\frac{1}{N}\sum_{j=1}^N I\{i_j=l_1\}}{\mathbb{E}_{\hat{p}^T(x,i)}\Big[I\{i=l_1\}\Big]} \cdot \dfrac{\mathbb{E}_{\hat{p}^T(x,i)}\Big[I\{i=l_1\}\Big]}{\mathbb{E}_{p(x,i)}\Big[I\{i=l_1\}\Big]}$ for all $1 \leq$

$l_1 \leq l$ which we get directly from Lemma G.5, with $R$ as in Lemma G.12, and Lemma G.16. This yields

$$C = \frac{1+\delta}{1-\delta} \cdot \frac{\frac{1}{\alpha}\left\|\frac{\nu_0(x,i)}{p(x,i)}\right\|_{L^\infty}}{\frac{c_{tilt}}{2\left\|\frac{\nu_0(x,i)}{p_{good}(x,i)}\right\|_{L^\infty}} - \Delta}$$

where $\alpha$ is the weight of the good component of joint distribution $p(x,i) = \alpha p_{good}(x,i) + (1 - \alpha)p_{bad}(x,i)$. First we note that by Theorem G.15 we have

$$\frac{1}{C'} \leq \frac{\hat{r}_k^{(l)}}{\hat{r}_{l+1}^{(l)}} \frac{Z_k}{Z_{l+1}} \leq C'$$

where $C' = l^2 \cdot C_2^2 \cdot O(\frac{U}{c_{tilt}})$. By the inductive hypothesis, we know that $\frac{1}{C_2} \leq \frac{\hat{r}_i Z_i}{\hat{r}_j Z_j} \leq C_2$ for $i, j \in [1, l]$. Together, this implies that $\frac{1}{\tilde{C}_2} \leq \frac{\hat{r}_i Z_i}{\hat{r}_j Z_j} \leq \tilde{C}_2$ for $i, j \in [1, l+1]$ with $\tilde{C}_2 = l^2 \cdot C_2^3 \cdot O(\frac{U}{c_{tilt}})$. Now, replacing $C_2$ with $\tilde{C}_2$ in the context of Lemma G.4 and then applied to Lemma G.10 yields

$$C_{PI}(p_{good}(x,i)) = O\left(\frac{C_1^2 \tilde{C}_2 CM l^2}{c_{tilt} \gamma \lambda}\right) = O\left(\frac{U C_1^2 C_2^3 CM \cdot l^4}{c_{tilt}^2 \gamma \cdot \lambda}\right).$$

Applying this to Lemma G.11 still yields a polynomial mixing time to bound $\Delta$. Therefore choosing appropriately $T = \Omega\left(\text{poly}(l, M, C_1, C_2, \frac{1}{c_{tilt}}, \frac{1}{\lambda}, \frac{1}{\gamma})\right)$, we have that $\Delta \leq \delta$. Since $\left\|\frac{\nu_0(x,i)}{p(x,i)}\right\|_{L^\infty} \leq \left\|\frac{\nu_0(x,i)}{p_{good}(x,i)}\right\|_{L^\infty}$ we get

$$C \leq \frac{2}{c_{tilt} \cdot \alpha} \left\|\frac{\nu_0(x,i)}{p_{good}(x,i)}\right\|_{L^\infty}^2 \cdot O(1).$$

By Assumptions 3.1,

$$C \leq \frac{1}{c_{tilt} \cdot \alpha} \cdot O(U^2).$$

Lastly, as shown in Lemma G.11, $\alpha \geq c_{tilt}$; therefore

$$C \leq O\left(\frac{U^2}{c_{tilt}^2}\right).$$

$\square$

## H   PROOF OF MAIN THEOREM

*Proof.* We will conclude the main theorem by applying Lemma F.3 to

$$\pi(x,i) \propto \sum_{l=1}^{L}\left(r_l \pi(x) \cdot \sum_{k=1}^{M} w_{l,k} q_l(x - x_k)\right) I\{i = l\},$$

with $L$ being the target level so that $q_L(x - x_k) = 1$. Rewrite $\pi(x,i)$ as

$$\pi(x,i) = \omega^L \pi(x) I\{i = L\} + \sum_{l=1}^{L-1} \omega^l \pi^l(x) I\{i = l\}.$$

To get an $\epsilon$ bound on TV-distance we note that by Cauchy-Schwarz,

$$TV(\hat{p}_T(x), \pi(x)) \leq \chi^2\left(\hat{p}_T(x) \,\|\, \pi(x)\right)^{\frac{1}{2}} = \left(\int_\Omega \left(\frac{\hat{p}_T(x,L)/\int_\Omega \hat{p}_T(x,L)dx}{\pi(x)} - 1\right)^2 \pi(x)dx\right)^{\frac{1}{2}}.$$

Lemma F.3 with $\pi_0^L = \pi^L = \pi(x)$ yields

$$\left(\int_\Omega \left(\frac{\hat{p}_T(x,L)/\int_\Omega \hat{p}_T(x,L)dx}{\pi(x)} - 1\right)^2 \pi(x)dx\right)^{\frac{1}{2}} \leq \frac{\Delta}{\int_\Omega \frac{\hat{p}_T(x,L)}{\omega^L} dx}. \tag{H.1}$$

where

$$\Delta := \left(\frac{\chi^2\left(\nu_0(x,i)\|\pi(x,i)\right) \cdot C_{PI}\left(\pi_0(x,i)\right)}{\alpha_0 \omega_0^L \cdot T}\right)^{\frac{1}{2}}.$$

It remains to show that (H.1) is $\leq \epsilon$. We first bound $\int_\Omega \frac{\hat{p}_T(x,L)}{\omega^L} dx$. By Lemma F.4, with $\pi_0 = \pi$,

$$\int_\Omega \frac{\hat{p}_T(x,L)}{\omega^L} dx \geq \frac{1}{2 \left\| \frac{\nu_0}{\pi_0} \right\|_\infty} - \frac{\Delta}{(\omega_0^L)^{\frac{1}{2}}}.$$

By Assumptions 3.1,

$$\chi^2\big(\nu_0(x,i)||\pi(x,i)\big) \leq \left\| \frac{\nu_0(x,i)}{\pi(x,i)} - 1 \right\|_{L^\infty} - 1 \leq \left\| \frac{\nu_0(x,i)}{\pi_0(x,i)} \right\|_{L^\infty} \leq U.$$

Then by Lemma G.10,

$$C_{PI}\big(\pi_0(x,i)\big) = poly\left( C_1, C_2, \tilde{C}, M, l, \frac{1}{c_{tilt}}, \frac{1}{\gamma}, \frac{1}{\lambda} \right).$$

Next we bound $\alpha_0$ and $\omega_0^L$ by using Assumption 1.1(2),

$$\alpha_0 = \frac{\sum_{i=1}^L \hat{r}_i^{(L)} Z_{i,0}}{\sum_{i=1}^L \hat{r}_i^{(L)} Z_i} \geq \frac{c_{tilt} \sum_{i=1}^L \hat{r}_i^{(L)} Z_i}{\sum_{i=1}^L \hat{r}_i^{(L)} Z_i} = c_{tilt}$$

$$\omega_0^L = \frac{\hat{r}_L^{(L)} w_{L,k} \int_\Omega \tilde{\pi}_{L,k}(x) dx}{\sum_{i=1}^L \hat{r}_i^{(L)} \sum_{k=1}^M w_{i,k} \int_\Omega \tilde{\pi}_{i,k}(x) dx}$$

$$= \frac{\hat{r}_L^{(L)} w_{L,k} Z_{L,k}}{\sum_{i=1}^L \hat{r}_i^{(L)} \sum_{k=1}^M w_{i,k} Z_{i,k}}$$

$$\geq \frac{1}{\sum_{i=1}^L \sum_{k=1}^M \frac{\hat{r}_i^{(L)} w_{i,k} Z_{i,k}}{\hat{r}_L^{(L)} w_{L,k'} Z_{L,k'}}}$$

$$\geq \frac{c_{tilt}}{L^2 \cdot M \cdot C_1^3 C_2},$$

where in the last step we use Lemma G.4. Therefore choosing $T = \Omega(\frac{1}{\varepsilon^2} \cdot poly(U, C_1, C_2, \tilde{C}, M, L, \frac{1}{c_{tilt}}, \frac{1}{\gamma}, \frac{1}{\lambda}))$ yields $\frac{\Delta}{(\omega_0^L)^{\frac{1}{2}}} \leq \frac{1}{4U}$, $\int_\Omega \frac{\hat{p}_T(x,L)}{\omega^L} dx \geq \frac{1}{4U}$, and $\frac{\Delta}{\int_\Omega \frac{\hat{p}_T(x,L)}{\omega^L} dx} \leq \varepsilon$. Noting that Theorem G.13 and Theorem G.17 yield $C_1 = poly(\frac{U}{c_{tilt}})$ and $C_2 = poly(\frac{U}{c_{tilt}})$, we have that $T = \Omega(\frac{1}{\varepsilon^2} poly(U, \tilde{C}, M, L, \frac{1}{c_{tilt}}, \frac{1}{\gamma}, \frac{1}{\lambda}))$ is sufficient. Moreover, the number of samples required at each level to run Algorithm 3 to ensure failure probability at most $\frac{\delta}{L}$ for each level (and hence total failure probability at most $\delta$) is given in Theorem G.13 and Theorem G.17 as $N = \Omega\left( \frac{RL}{\delta} \right)$. $R = poly(l, M, C_1, C_2, \frac{1}{c_{tilt}}, U)$ is given in Lemma G.12 and with $C_1 = poly(\frac{U}{c_{tilt}})$ and $C_2 = poly(\frac{U}{c_{tilt}})$ this reduces to $R = poly(L, M, \frac{1}{C_{tilt}}, U)$. Therefore, $N = \Omega(poly(L, M, U, \frac{1}{c_{tilt}}, \frac{1}{\delta}))$

$\square$

# I  GENERAL SETTING

## I.1  TEMPERING ON $\mathbb{R}^d$

In this subsection, we place reasonable assumptions on the tempering function $q_l(x)$ in $\mathbb{R}^d$ and show that Assumptions 3.1 hold. More specifically, we determine lower bounds on the probability flow between two modes of the projected chain. Lower bounding the probability flow between modes will provide us with a lower bound on the spectral gap, in turn, enabling us to upper bound the Poincaré constant $\bar{C}$ of the projected chain from section E. The following assumptions will be made for this subsection.

**Assumption I.1.**  *Let $\tilde{p}_i(x) = \sum_{k=1}^M \alpha_k p_k(x) \sum_{j=1}^M w_{i,j} q_i(x - x_j)$.*

  *1. The tempering function $q_i$ is defined as*

$$q_i(x) = e^{-\beta_i \frac{||x||^2}{2}}.$$

2. *We let the push forward measure $q_{jj'}^{\#}$ be defined as the translation*

$$q_{jj'}^{\#}(x) = x - x_j + x_{j'}.$$

3. *The function $\alpha_k p_k(x) = e^{-f_k(x)}$ where $f_k(x)$ is L-smooth.*

The following Lemma will allow us to find a suitable lower bound on the probability flows between modes by bounding the $\chi^2$-divergence between mixture components.

**Lemma I.2.** *Let $p_{\beta_i, good}$ be the probability distribution defined as in (B.3). Let the distribution $p_{ij} = \frac{\alpha_j p_j(x) e^{-\beta_i \frac{||x - x_j||^2}{2}}}{Z_j(\beta_i)}$ satisfy a Poincaré inequality with constant $C_{ij}$ and $||x_j - \mathbb{E}_{p_{ij}}(x)|| \leq \delta$ for some constant $\delta \geq 0$. Lastly, let $\Delta\beta = \beta_i - \beta_{i'}$ and $\Delta\beta \in [0, \frac{1}{2C_{LS}}]$. Then*

$$\chi^2(p_{i'j} || p_{ij}) \leq \frac{1}{\sqrt{1 - 2C_{LS}\Delta\beta}} \cdot \exp\left(\frac{2(dC_{LS} + \delta^2)\Delta\beta}{1 - 2C_{LS}\Delta\beta}\right) - 1.$$

*In particular, for $\Delta\beta = O(\frac{1}{C_{LS}d + \delta^2})$ this is $O(1)$.*

*Proof.* We have

$$\chi^2\left(\frac{\alpha_j p_j(x) q_{\beta_{i'}}(x - x_j)}{Z_{i'j}} \Big|\Big| \frac{\alpha_j p_j(x) q_{\beta_i}(x - x_j)}{Z_{ij}}\right) = \int_\Omega \left(\frac{e^{-\beta_{i'} \frac{||x - x_j||^2}{2}} Z_{ij}}{e^{-\beta_i \frac{||x||^2}{2}} Z_{i'j}}\right)^2 \alpha_j p_j(x) e^{-\beta_i \frac{||x - x_j||^2}{2}} / Z_{ij} dx - 1$$

$$= \left(\frac{Z_{i,j}}{Z_{i',j}}\right)^2 \int_\Omega e^{(\beta_i - \beta_{i'})||x - x_j||^2} \frac{\alpha_j p_j(x) e^{-\beta_i ||x - x_j||^2}}{Z_{ij}} dx - 1$$

Further note that with $\beta_{i'} < \beta_i$ and $e^{-\beta ||x - x_j||^2/2} \leq 1$, it follows that

$$Z_{i,j} = \int_\Omega \alpha_j p_j(x) e^{-\beta_i ||x - x_j||^2/2} dx \leq \int_\Omega \alpha_j p_j(x) e^{-\beta_{i'} ||x - x_j||^2/2} dx = Z_{i',j}.$$

Therefore, with $p_{ij} = \frac{\alpha_j p_j(x) e^{-\beta_i ||x - x_j||^2}}{Z_{ij}}$,

$$\left(\frac{Z_{i,j}}{Z_{i',j}}\right)^2 \int_\Omega e^{(\beta_i - \beta_{i'})||x - x_j||^2} \frac{\alpha_j p_j(x) e^{-\beta_i ||x - x_j||^2}}{Z_{ij}} dx - 1 \leq \int_\Omega e^{\Delta\beta ||x - x_j||^2} \frac{\alpha_j p_j(x) e^{-\beta_i ||x - x_j||^2}}{Z_{ij}} dx - 1$$

$$= \mathbb{E}_{p_{ij}}\left[e^{\Delta\beta ||x - x_j||^2}\right].$$

Applying Lemma J.6 yields

$$\mathbb{E}_{p_{ij}}\left[e^{\Delta\beta ||x - x_j||^2}\right] \leq \frac{1}{\sqrt{1 - 2C_{LS}\Delta\beta}} \cdot \exp\left(\frac{\Delta\beta}{1 - 2C_{LS}\Delta\beta} \mathbb{E}_{p_{ij}}\left[||x - x_j||\right]^2\right)$$

$$\leq \frac{1}{\sqrt{1 - 2C_{LS}\Delta\beta}} \cdot \exp\left(\frac{\Delta\beta}{1 - 2C_{LS}\Delta\beta} \mathbb{E}_{p_{ij}}\left[||x - \mathbb{E}_{p_{ij}}(x)|| + ||\mathbb{E}_{p_{ij}}(x) - x_j||\right]^2\right)$$

$$\leq \frac{1}{\sqrt{1 - 2C_{LS}\Delta\beta}} \cdot \exp\left(\frac{2\Delta\beta}{1 - 2C_{LS}\Delta\beta}\left(\sum_{k=1}^d \text{Var}_{p_{ij}}(x_k) + \delta^2\right)\right)$$

Lastly, applying the LSI inequality,

$$\leq \frac{1}{\sqrt{1 - 2C_{LS}\Delta\beta}} \cdot \exp\left(\frac{2(dC_{LS} + \delta^2)\Delta\beta}{1 - 2C_{LS}\Delta\beta}\right).$$

$\square$

**Lemma I.3.** *Let $p_{\beta_i, good}$ be the probability distribution defined as in (B.3). For all $1 \leq j \leq M$ let $\alpha_j p_j(x) = e^{-f_j(x)}$ for some L-smooth function $f_j(x)$ and let $||x_j - x_j^*|| \leq D$ for some constant $D \geq 0$. Then*

$$\chi^2\left(\frac{\alpha_{j'} q_{jj'}^{\#} p_{j'}(x) q_1(x - x_{j'})}{Z_{1j'}} \Big|\Big| \frac{\alpha_j p_j(x) q_1(x - x_j)}{Z_{1j}}\right) \leq \left(\frac{\beta_1 + L}{\beta_1 - 3L}\right)^d e^{5\frac{L^2 D^2}{\beta_1 - 3L}} - 1.$$

*In particular, for $\beta_1 = \Omega(L^2 D^2 d)$, this is $O(1)$.*

*Proof.* Consider the following,

$$\chi^2 \left( \frac{\alpha_{j'} q_{jj'}^{\#} p_{j'}(x) q_1(x - x_{j'})}{Z_{1j'}} || \frac{\alpha_j p_j(x) q_1(x - x_j)}{Z_{1j}} \right)$$

$$= \int_\Omega \left( \frac{\alpha_{j'} q_{jj'}^{\#} p_{j'}(x) q_1(x - x_{j'})}{Z_{1j'}} \Big/ \frac{\alpha_j p_j(x) q_1(x - x_j)}{Z_{1j}} - 1 \right)^2 \frac{\alpha_j p_j(x) q_1(x - x_j)}{Z_{1j}} dx$$

$$= \int_\Omega \left( \frac{\alpha_{j'} p_{j'}(x - x_j + x_{j'}) q_1(x - x_j)}{Z_{1j'}} \Big/ \frac{\alpha_j p_j(x) q_1(x - x_j)}{Z_{1j}} - 1 \right)^2 \frac{\alpha_j p_j(x) q_1(x - x_j)}{Z_{1j}} dx$$

$$= \frac{Z_{1j}}{Z_{1j'}^2} \int_\Omega \frac{\alpha_{j'}^2 p_{j'}(x - x_j + x_{j'})^2}{\alpha_j p_j(x)} e^{-\beta_i \frac{||x - x_j||^2}{2}} dx - 1$$

We continue by finding an upper bound on

$$\frac{Z_{1j}}{Z_{1j'}^2} \int_\Omega \frac{\alpha_{j'}^2 p_{j'}(x - x_j + x_{j'})^2}{\alpha_j p_j(x)} e^{-\beta_i \frac{||x - x_j||^2}{2}} dx = \frac{Z_{1j}}{Z_{1j'}^2} \int_\Omega \frac{e^{-2 f_{j'}(x - x_j + x_{j'})}}{e^{-f_j(x)}} e^{-\beta_i \frac{||x - x_j||^2}{2}} dx$$

By letting $a_j p_j(x) = e^{-f_j(x)}$ for some $L$-smooth $f_j(x)$,

$$\leq \frac{Z_{1j}}{Z_{1j'}^2} \frac{e^{-2 f_{j'}(x_{j'})}}{e^{-f_j(x_j)}} \int_\Omega e^{(2 \boldsymbol{\nabla} f_{j'}(x_{j'}) - \boldsymbol{\nabla} f_j(x_j))^T (x - x_j)} e^{\frac{3}{2} L ||x - x_j||^2} e^{-\beta_1 \frac{||x - x_j||^2}{2}} dx$$

Letting $v = 2 \boldsymbol{\nabla} f_{j'}(x_{j'}) - \boldsymbol{\nabla} f_j(x_j)$,

$$= \frac{Z_{1j}}{Z_{1j'}^2} \frac{e^{-2 f_{j'}(x_{j'})}}{e^{-f_j(x_j)}} \int_\Omega e^{-\frac{1}{2}(3L - \beta_1)(x - x_j + \frac{1}{3L - \beta_1} v)^T (x - x_j + \frac{1}{3L - \beta_1} v) + \frac{1/2}{3L - \beta_1} v^T v} dx$$

$$= \frac{Z_{1j}}{Z_{1j'}^2} \frac{e^{-2 f_{j'}(x_{j'})}}{e^{-f_j(x_j)}} e^{\frac{1/2}{\beta_1 - 3L} v^T v} \left( \frac{2\pi}{\beta_1 - 3L} \right)^{\frac{d}{2}}$$

We can bound $v^T v = ||2 \boldsymbol{\nabla} f_{j'}(x_{j'}) - \boldsymbol{\nabla} f_j(x_j)||^2 \leq (2L ||x_{j'} - x_{j'}^*|| + L ||x_j - x_j^*||)^2 \leq 9 L^2 D^2$

$$\leq \frac{Z_{1j}}{Z_{1j'}^2} \frac{e^{-2 f_{j'}(x_{j'})}}{e^{-f_j(x_j)}} e^{\frac{9 L^2 D^2}{2(\beta_1 - 3L)}} \left( \frac{2\pi}{\beta_1 - 3L} \right)^{\frac{d}{2}}$$

Now we can bound the following,

$$Z_{1j} = \int_\Omega e^{-\beta_i \frac{||x - x_j||^2}{2}} e^{-f_j(x)} dx$$

$$\leq \int_\Omega e^{-f_j(x_j) + \boldsymbol{\nabla} f_j(x_j)^T (x - x_j) + \frac{L}{2} ||x - x_j||^2} e^{-\beta_i \frac{||x - x_j||^2}{2}} dx$$

$$= e^{-f_j(x_j)} e^{\frac{1}{2(\beta_i - L)} \boldsymbol{\nabla} f_j(x_j)^T \boldsymbol{\nabla} f_j(x_j)} \int_\Omega e^{-\frac{\beta_1 - L}{2} ||x - x_j - \frac{1}{\beta_1 - L} \boldsymbol{\nabla} f_j(x_j)||^2} dx$$

$$\leq e^{-f_j(x_j)} e^{\frac{L^2 D^2}{2(\beta_i - L)}} \left( \frac{2\pi}{\beta_1 - L} \right)^{\frac{d}{2}}$$

$$Z_{1j'} = \int_\Omega e^{-\beta_i \frac{||x - x_{j'}||^2}{2}} e^{-f_{j'}(x)} dx$$

$$\geq \int_\Omega e^{-f_{j'}(x_{j'}) - \boldsymbol{\nabla} f_{j'}(x_{j'})^T (x_{j'} - x) - \frac{L}{2} ||x - x_{j'}||^2}$$

$$= e^{-f_{j'}(x_{j'})} e^{\frac{1}{2(\beta_1 + L)} \boldsymbol{\nabla} f_{j'}(x_{j'})^T \boldsymbol{\nabla} f_{j'}(x_{j'})} \int_\Omega e^{-\frac{\beta_1 + L}{2} ||x - x_{j'} + \frac{1}{\beta_1 + L} \boldsymbol{\nabla} f_{j'}(x_{j'})||^2} dx$$

$$= \left( \frac{2\pi}{\beta_1 + L} \right)^{\frac{d}{2}} e^{-f_{j'}(x_{j'})} e^{\frac{\boldsymbol{\nabla} f_{j'}(x_{j'})^T \boldsymbol{\nabla} f_{j'}(x_{j'})}{2(\beta_1 + L)}}.$$

Therefore we have that,

$$\frac{Z_{1j}}{Z_{1j'}^2} \frac{e^{-2f_{j'}(x_{j'})}}{e^{-f_j(x_j)}} e^{\frac{9L^2D^2}{2(\beta_1-3L)}} \Big(\frac{2\pi}{\beta_1-3L}\Big)^{\frac{d}{2}} \leq \frac{\big(\frac{2\pi}{\beta_1-L}\big)^{\frac{d}{2}} e^{-f_j(x_j)} e^{\frac{L^2D^2}{2(\beta_1-L)}}}{\big(\frac{2\pi}{\beta_1+L}\big)^d e^{-2f_{j'}(x_{j'})} e^{2\frac{\boldsymbol{\nabla}f_{j'}(x_{j'})^T\boldsymbol{\nabla}f_{j'}(x_{j'})}{2(\beta_1+L)}}} \frac{e^{-2f_{j'}(x_{j'})}}{e^{-f_j(x_j)}} e^{\frac{9L^2D^2}{2(\beta_1-3L)}} \Big(\frac{2\pi}{\beta_1-3L}\Big)^{\frac{d}{2}}$$

$$= \Big(\frac{2\pi}{\beta_1-L}\Big)^{\frac{d}{2}} e^{\frac{L^2D^2}{2(\beta_1-L)}} \Big(\frac{\beta_1+L}{2\pi}\Big)^d e^{-2\frac{\boldsymbol{\nabla}f_{j'}(x_{j'})^T\boldsymbol{\nabla}f_{j'}(x_{j'})}{2(\beta_1+L)}} e^{\frac{9L^2D^2}{2(\beta_1-3L)}} \Big(\frac{2\pi}{\beta_1-3L}\Big)^{\frac{d}{2}}$$

for $L > 0$, $\beta_1 - L > \beta_1 - 3L$ and since $\frac{\boldsymbol{\nabla}f_{j'}(x_{j'})^T\boldsymbol{\nabla}f_{j'}(x_{j'})}{2(\beta_1+L)} > 0$, we have $e^{-2\frac{\boldsymbol{\nabla}f_{j'}(x_{j'})^T\boldsymbol{\nabla}f_{j'}(x_{j'})}{2(\beta_1+L)}} < 1$ and hence,

$$\leq \Big(\frac{\beta_1+L}{\beta_1-3L}\Big)^d e^{5\frac{L^2D^2}{\beta_1-3L}}.$$

$\square$

We will show that the base case of **H1**(1) and **H2**(1) hold under Assumptions I.1. To do this we will reuse our previous analysis from this section. In Lemma I.3, we were able to show that if $\alpha_j p_j(x) = e^{-f_j(x)}$, with $L$-smooth $f_j(x)$ for all $j$, then

$$Z_{1j} \leq \Big(\frac{2\pi}{\beta_1-L}\Big)^{\frac{d}{2}} e^{-f_j(x_j)} e^{\frac{L^2D^2}{2(\beta_i-L)}}$$

and

$$Z_{1j} \geq \Big(\frac{2\pi}{\beta_1+L}\Big)^{\frac{d}{2}} e^{-f_j(x_j)}.$$

We first show that by choosing $\beta_1$ large enough the partition functions $Z_{1k}$ can be well approximated. With good enough approximates of $Z_{1k}$, we then show that we can estimate $Z_1$ up to a constant factor.

**Lemma I.4.** *Let Assumptions 1.1 hold and assume that $p(x_k) > \alpha_k p_k(x_k) > c_{tilt}p(x_k)$ (this is the limit as $\beta \to \infty$ of the tilting assumption in Assumptions 1.1). If $\beta_1 = \Omega(\frac{L^2D^2d}{\epsilon})$ with appropriate constants, then*

$$c_{tilt}(1-\epsilon) \cdot p(x_k) \leq Z_{1k} \leq (1+\epsilon) \cdot p(x_k).$$

*Proof.* By our previous bounds on $Z_{1k}$ and Assumptions 1.1, we choose $\beta_1$ such that

$$1 - \epsilon \leq \Big(\frac{\beta_1}{\beta_1+L}\Big)^{\frac{d}{2}} \iff \beta_1 \geq \frac{L}{\big(\frac{1}{1-\epsilon}\big)^{\frac{2}{d}}-1}.$$

Noting that $\big(\frac{1}{1-\epsilon}\big)^{\frac{2}{d}} = 1 + \Theta\big(\frac{\epsilon}{d}\big)$ gives that it suffices for $\beta_1 = \Omega\big(\frac{Ld}{\epsilon}\big)$. In similar fashion, for the upper bound, we choose $\beta_1$ such that

$$1 + \epsilon_1 \geq \Big(\frac{\beta_1}{\beta_1-L}\Big)^{\frac{d}{2}} \iff \beta_1 \geq \frac{L}{\big(\frac{1}{1+\epsilon}\big)^{\frac{d}{2}}+1}.$$

A similar analysis to the lower bound yields that this is satisfied when $\beta_1 = \Omega(\frac{Ld}{\epsilon_1})$. We also impose

$$1 + \epsilon_2 \geq e^{\frac{L^2D^2}{2(\beta_1-L)}} \iff \beta_1 = \frac{L^2D^2}{2\ln(1+\epsilon_2)} + L,$$

for which it suffices that $\beta_1 = \Omega(\frac{L^2D^2}{\ln(1+\epsilon_2)})$. By letting $1 + \epsilon = (1+\epsilon_1)(1+\epsilon_2)$ and $\epsilon_1 = \epsilon_2$, we require that $\beta_1 = \Omega\Big(\frac{L^2D^2+Ld}{\epsilon}\Big)$. $\square$

**Corollary I.5.** $\boxed{H1(1)}$ *Taking $w_{1,k} \propto \frac{1}{p(x_k)}$ and choosing $\beta_1 = \Omega(\frac{L^2D^2d}{\epsilon})$ yields*

$$\frac{c_{tilt}(1-\epsilon)}{1+\epsilon} \leq \frac{w_{1k'}Z_{1k'}}{w_{1k}Z_{1k}} \leq \frac{1+\epsilon}{c_{tilt}(1-\epsilon)},$$

*i.e.*

$$\frac{1}{C_1} \le \frac{w_{1k'}Z_{1k'}}{w_{1k}Z_{1k}} \le C_1,$$

*where $C_1 = O(\frac{1}{c_{tilt}})$.*

**Lemma I.6.** *Let $\tilde{p}_0(x,i) = \sum_{j=1}^{l} \hat{r}_j^{(l)} \tilde{p}_{j0}(x)I\{i=j\}$, with $\tilde{p}_{j0}(x) = \sum_{k=1}^{M} w_{j,k}\alpha_k\pi_k(x)q_j(x - x_k)$. Here, $\hat{r}_1^{(l)} = C_2 r_1^{(l)}$ and $\hat{r}_j^{(l)} = r_j^{(l)}$ for $j = 2, \dots, l$. Moreover, we define the normalized $p_0(x,i) = \sum_{j=1}^{l} \omega_0^j p_{j0}(x)I\{i=j\}$ with $p_{j0}(x) = \frac{1}{Z_{j0}}\tilde{p}_{j0}(x)$. Lastly, by choosing*

$$\nu_0(x,i) = \frac{1}{\hat{Z}_1} \sum_{k=1}^{M} c_{tilt}\pi(x_k)w_{1,k}\left(\frac{5L+\beta_1}{2\pi}\right)^{\frac{d}{2}} \exp\left(-(\frac{5L+\beta_1}{2})||x-x_k||^2\right)I\{i=1\},$$

*we have that*

$$\left\|\frac{\nu_0(x,i)}{p_0(x,i)}\right\|_\infty \le \frac{1+\epsilon}{c_{tilt}^2}\left(\frac{L+\beta_1+L^2D^2}{\beta_1}\right)^{\frac{d}{2}} \cdot O(1).$$

*Moreover, choosing $\beta_1 = \Omega(L^2D^2d)$,*

$$\left\|\frac{\nu_0(x,i)}{p_0(x,i)}\right\|_\infty = O\left(\frac{1}{c_{tilt}^2}\right).$$

*Proof.* Let $\nu_0(x,i) = \frac{1}{\hat{Z}_1} \sum_{k=1}^{M} \hat{w}_{1,k}\hat{q}_{1,k}(x)I\{i=k\}$ for some $\hat{q}_{1,k}$—to be defined later. Then we can consider

$$\left\|\frac{\nu_0(x,i)}{p_0(x,i)}\right\|_\infty = \left\|\frac{\frac{1}{\hat{Z}_1}\sum_{k=1}^{M}\hat{w}_{1,k}\hat{q}_{1,k}(x)}{\omega_0^1/Z_{10}\sum_{k=1}^{M}w_{1,k}\alpha_k\pi_k(x)q_j(x-x_k)}\right\|_\infty = \frac{1}{\hat{Z}_1}\frac{Z_{10}}{\omega_0^1}\left\|\frac{\sum_{k=1}^{M}\hat{w}_{1,k}\hat{q}_{1,k}(x)}{\sum_{k=1}^{M}w_{1,k}\alpha_k\pi_k(x)q_j(x-x_k)}\right\|_\infty.$$

To bound $\frac{A_1+\dots+A_M}{B_1+\dots+B_M} = \frac{B_1\frac{A_1}{B_1}+\dots+B_M\frac{A_M}{B_M}}{B_1+\dots+B_M}$ it's sufficient to bound $\frac{A_k}{B_k}$ for all $k$. Therefore, by using that $\alpha_k\pi_k(x) = e^{-f_k(x)}$ where $f_k(x)$ is $L$-smooth and $q_j(x-x_k)$ is a Gaussian centered at $x_k$ with variance $\beta_1$, we consider

$$\left\|\frac{\hat{w}_{1,k}\hat{q}_{1,k}(x)}{w_{1,k}\alpha_k\pi_k(x)q_j(x-x_k)}\right\|_\infty$$

$$\le \left\|\frac{\hat{w}_{1,k}\hat{q}_{1,k}(x)}{w_{1,k}\alpha_k\pi_k(x_k)\left(\frac{\beta_1}{2\pi}\right)^{\frac{d}{2}}\exp\left(-(x-x_k)^T\boldsymbol{\nabla}f_k(x_k)-\frac{L}{2}||x-x_k||^2-\frac{\beta_1}{2}||x-x_k||^2\right)}\right\|_\infty$$

$$= \frac{\hat{w}_{1,k}}{w_{1,k}\alpha_k\pi_k(x_k)\left(\frac{\beta_1}{2\pi}\right)^{\frac{d}{2}}}\left\|\hat{q}_{1,k}(x)\exp\left((x-x_k)^T\boldsymbol{\nabla}f_k(x_k)+\frac{L}{2}||x-x_k||^2+\frac{\beta_1}{2}||x-x_k||^2\right)\right\|_\infty$$

Now letting $\hat{q}_{1,k} = \left(\frac{\alpha}{2\pi}\right)^{\frac{d}{2}}\exp\left(-\frac{\alpha}{2}||x-x_k||^2\right)$,

$$= \frac{\hat{w}_{1,k}\left(\frac{\alpha}{2\pi}\right)^{\frac{d}{2}}}{w_{1,k}\alpha_k\pi_k(x_k)\left(\frac{\beta_1}{2\pi}\right)^{\frac{d}{2}}}\left\|\exp\left((x-x_k)^T\boldsymbol{\nabla}f_k(x_k)+\frac{L}{2}||x-x_k||^2+\frac{\beta_1}{2}||x-x_k||^2-\frac{\alpha}{2}||x-x_k||^2\right)\right\|_\infty$$

$$= \frac{\hat{w}_{1,k}\left(\frac{\alpha}{2\pi}\right)^{\frac{d}{2}}}{w_{1,k}\alpha_k\pi_k(x_k)\left(\frac{\beta_1}{2\pi}\right)^{\frac{d}{2}}}\left\|\exp\left(\frac{L+\beta_1-\alpha}{2}\left\|x-x_k+\frac{\boldsymbol{\nabla}f_k(x_k)}{L+\beta-\alpha}\right\|^2-\frac{\boldsymbol{\nabla}f_k(x_k)^T\boldsymbol{\nabla}f_k(x_k)}{2(L+\beta_1-\alpha)}\right)\right\|_\infty$$

Choose $\alpha = \beta_1 + L + L^2 D^2 > \beta_1 + L$; then

$$\leq \frac{\hat{w}_{1,k}\left(\frac{L+\beta_1+LD^2}{2\pi}\right)^{\frac{d}{2}}}{w_{1,k}\alpha_k\pi_k(x_k)\left(\frac{\beta_1}{2\pi}\right)^{\frac{d}{2}}}\left\|\exp\left(-\frac{LD^2}{2}\left\|x - x_k + \frac{\boldsymbol{\nabla}f_k(x_k)}{L+\beta-\alpha}\right\|^2 + \frac{\boldsymbol{\nabla}f_k(x_k)^T\boldsymbol{\nabla}f_k(x_k)}{2LD^2}\right)\right\|_\infty$$

$$\leq \frac{\hat{w}_{1,k}\left(\frac{L+\beta_1+LD^2}{2\pi}\right)^{\frac{d}{2}}}{w_{1,k}\alpha_k\pi_k(x_k)\left(\frac{\beta_1}{2\pi}\right)^{\frac{d}{2}}}\left\|1\cdot\exp\left(\frac{L^2\|x_k - x_k^*\|^2}{2LD^2}\right)\right\|_\infty$$

$$\leq \frac{\hat{w}_{1,k}}{w_{1,k}\alpha_k\pi_k(x_k)}\left(\frac{L+\beta_1+LD^2}{\beta_1}\right)^{\frac{d}{2}}\left\|1\cdot\exp\left(\frac{L^2D^2}{2L^2D^2}\right)\right\|_\infty$$

Lastly, by the assumption that $\alpha_k\pi_k(x_k) \geq c_{tilt}\pi(x_k)$ we have

$$\leq \frac{\hat{w}_{1,k}}{w_{1,k}c_{tilt}\pi(x_k)}\left(\frac{L+\beta_1+LD^2}{\beta_1}\right)^{\frac{d}{2}}\cdot O(1)$$

Therefore we have a bound given by

$$\left\|\frac{\nu_0(x,i)}{p_0(x,i)}\right\|_\infty \leq \frac{1}{\hat{Z}_1}\frac{Z_{10}}{\omega_0^1}\frac{\hat{w}_{1,k}}{w_{1,k}c_{tilt}\pi(x_k)}\left(\frac{L+\beta_1+L^2D^2}{\beta_1}\right)^{\frac{d}{2}}\cdot O(1).$$

Lastly by choosing $\hat{w}_{1,k} = c_{tilt}\pi(x_k)w_{1,k}$, the same estimate we use for the component measure weights, we get

$$= \frac{1}{\hat{Z}_1}\frac{Z_{10}}{\omega_0^1}\left(\frac{L+\beta_1+L^2D^2}{\beta_1}\right)^{\frac{d}{2}}\cdot O(1).$$

This yields a bound of

$$\left\|\frac{\nu_0(x,i)}{p_0(x,i)}\right\|_\infty \leq \frac{Z_{10}}{\hat{Z}_1}\frac{1}{\omega_0^1}\left(\frac{L+\beta_1+L^2D^2}{\beta_1}\right)^{\frac{d}{2}}\cdot O(1).$$

Since, by definition of $\nu_0(x,i)$, $\hat{Z}_1 = c_{tilt}\sum_{k=1}^M w_{1,k}\pi(x_k)$ we get that

$$\left\|\frac{\nu_0(x,i)}{p_0(x,i)}\right\|_\infty \leq \frac{\sum_{k=1}^M w_k Z_{1k}}{c_{tilt}\sum_{k=1}^M w_{1,k}\pi(x_k)}\frac{1}{\omega_0^1}\left(\frac{L+\beta_1+L^2D^2}{\beta_1}\right)^{\frac{d}{2}}\cdot O(1)$$

$$= \frac{\sum_{k=1}^M w_k\frac{Z_{1k}}{\pi(x_k)}\pi(x_k)}{c_{tilt}\sum_{k=1}^M w_{1,k}\pi(x_k)}\frac{1}{\omega_0^1}\left(\frac{L+\beta_1+L^2D^2}{\beta_1}\right)^{\frac{d}{2}}\cdot O(1).$$

By Lemma I.4

$$\leq \frac{1+\epsilon}{c_{tilt}}\frac{1}{\omega_0^1}\left(\frac{L+\beta_1+L^2D^2}{\beta_1}\right)^{\frac{d}{2}}\cdot O(1).$$

Lastly, we have that $\omega_0^1 = \frac{\hat{r}_1^{(l)}Z_{10}}{\sum_{i=1}^l \hat{r}_i^{(l)}Z_{i0}}$; therefore

$$\frac{1}{\omega_0^1} = \frac{\sum_{i=1}^l \hat{r}_i^{(l)}Z_{i0}}{\hat{r}_1^{(l)}Z_{10}} \leq \frac{1}{c_{tilt}}\frac{\sum_{i=1}^l \hat{r}_i^{(l)}Z_i}{\hat{r}_1^{(l)}Z_1} = \frac{1}{c_{tilt}}\frac{l\cdot C_2 r_1^{(l)}Z_1 + \sum_{i=2}^l r_i^{(l)}Z_i}{l\cdot C_2 r_1^{(l)}Z_1} \leq \frac{2}{c_{tilt}}.$$

$\square$

**Lemma I.7.** *Let Assumptions I.1 and 1.1 hold. Let* $\max_j \|\mathbb{E}_{\mu_{l,j,k}}[x] - x_k\| \leq \delta_c$ *with* $\mu_{l,j,k} = \frac{\alpha_j\pi_j(x)e^{\frac{-\beta_l\|x-x_k\|^2}{2}}}{Z_{l,j,k}}$. *Additionally, assume that each* $\mu_{j,k}$ *satisfies a log Sobolev inequality with*

*constant $C_{LS}^{(ij)}$ and let $C_{LS}^* = \max_{i,j} C_{LS}^{(ij)}$. Then*

$$\chi^2\left(\frac{\bar{\pi}_{l+1,k}}{\bar{Z}_{l+1,k}} \,\|\, \frac{\bar{\pi}_{l,k}}{\bar{Z}_{l,k}}\right) \le \frac{1}{\sqrt{1 - 2C_{LS}^*\Delta\beta}} \cdot \exp\left(\frac{\Delta\beta\left(C_{LS}^*d + \delta_c^2\right)}{1 - 2C_{LS}^*\Delta\beta}\right).$$

*In particular, if $\Delta\beta = O(\frac{1}{C_{LS}d + \delta_c^2})$ this is $O(1)$.*

*Proof.* By Lemma G.9 it's left to upper bound $\chi^2\left(\frac{\bar{\pi}_{l+1,k}}{\bar{Z}_{l+1,k}} \,\|\, \frac{\bar{\pi}_{l,k}}{\bar{Z}_{l,k}}\right)$. Let $\Delta\beta = \beta_l - \beta_{l+1}$, simplification of the $\chi^2$ term yields,

$$\chi^2\left(\frac{\bar{\pi}_{l+1,k}}{\bar{Z}_{l+1,k}} \,\|\, \frac{\bar{\pi}_{l,k}}{\bar{Z}_{l,k}}\right) = \frac{\bar{Z}_{l,k}^2}{\bar{Z}_{l+1,k}^2} \int_\Omega e^{\Delta\beta||x-x_k||^2} \frac{\pi(x)e^{-\beta_l\frac{||x-x_k||^2}{2}}}{\bar{Z}_{l,k}} dx$$

$\frac{\bar{Z}_{l,k}^2}{\bar{Z}_{l+1,k}^2} \le 1$ since $\beta_{l+1} < \beta_l$ therefore

$$\le \sum_j \frac{Z_{l,j,k}}{\bar{Z}_{l,k}} \int_\Omega e^{\Delta\beta||x-x_k||^2} \frac{\alpha_j\pi_j(x)e^{\frac{-\beta_l||x-x_k||^2}{2}}}{Z_{l,j,k}} dx.$$

Let $\mu_{l,j,k}(x) = \frac{\alpha_j\pi_j(x)e^{\frac{-\beta_l||x-x_k||^2}{2}}}{Z_{l,j,k}}$ then by Lemma J.6 with $\Delta\beta \in [0, \frac{1}{2C_{LS}^*}]$

$$\le \sum_j \frac{Z_{l,j,k}}{\bar{Z}_{l,k}} \frac{1}{\sqrt{1 - 2C_{LS}^*\Delta\beta}} \cdot \exp\left(\frac{\Delta\beta}{1 - 2C_{LS}^*\Delta\beta} \mathbb{E}_{\mu_{l,j,k}}\left[||x - x_k||\right]^2\right).$$

Lastly, by the triangle inequality and applying LSI with $\max_j ||\mathbb{E}_{\mu_{l,j,k}}[x] - x_k|| \le \delta_c$

$$\le \frac{1}{\sqrt{1 - 2C_{LS}^*\Delta\beta}} \cdot \exp\left(\frac{\Delta\beta\left(C_{LS}^*d + \delta_c^2\right)}{1 - 2C_{LS}^*\Delta\beta}\right).$$

Finally, we prove Proposition 3.2 as corollary of the previous Lemmas. □

*Proof.* (Proposition 3.2) Lemma I.2 with choice of $\Delta\beta = O(\frac{1}{C_{LS}d + r^2})$ yields $\chi^2\left(\pi_{l+1,k}||\pi_{l,k}\right) = O(1)$. Lemma I.3 with choice of $\beta_1 = \Omega(L^2D^2d)$ yields $\chi^2\left(\pi_{1,k}||\pi_{1,j}\right) = O(1)$. Lemma with choice of $\Delta\beta = O(\frac{1}{C_{LS}d + r^2})$ yields $\chi^2\left(\frac{\bar{\pi}_{l+1,k}}{\bar{Z}_{l+1,k}}||\frac{\bar{\pi}_{l,k}}{\bar{Z}_{l,k}}\right) = O(1)$.

Corollary I.5 guarantees level balance from definition 2.3 on level 1. Lastly, Lemma I.6 with an appropriate choice of $\nu_0(x,i)$ yields $U = O(\frac{1}{c_{tilt}^2})$

Since the warmest level is $\beta_L = 0$ and the coldest is $\beta_1 = \Omega(L^2D^2d)$ with choice of $\Delta\beta = O(\frac{1}{C_{LS}d + r^2})$ this yields $\frac{\beta_1 - \beta_L}{\Delta\beta} = \Omega(L^2D^2d^2C_{LS}r^2)$ levels. Therefore with respect to dimensionality we require $\Omega(d^2)$ levels.

□

## I.2 MIXTURE OF GAUSSIANS

In Section I.1 we showed that given exponential tempering functions, Assumptions 3.1 hold. This shows that the theory work in Section G.3 holds in a general setting. It is left to show that for a family of target functions, Assumptions 1.1 hold. In this subsection we will show that for a mixture of Gaussians with different variances, Assumptions 1.1 hold, which, in conjunction with Section I.1, gives a broad setting for Theorem 3.3 to be applied. For simplicity we consider the case of spherical Gaussians.

**Proposition I.8.** *Assume the setting of Section I.1 outlined in Assumptions I.1. Additionally, assume that the mixture components of the target measure are $\sum_k \alpha_k\pi_k(x) = \sum_k \alpha_k\left(\frac{a_k}{\pi}\right)^{\frac{d}{2}} e^{-a_k||x-\mu_k||^2}$. We make the following technical assumptions that quantify distance between modes let $\Delta_i = \frac{\beta_l}{1 + \frac{\beta_l}{a_i}}$*

*and $d_{ik} = \|\mu_i - x_k\|$ then we require $d_{kk}^2 \leq d_{jk}^2$, $\frac{d}{2\max\{\Delta_k, \Delta_j\}} \log\left(\frac{\Delta_j}{\Delta_k}\right) \leq d_{jk}^2 - d_{kk}^2$ and $\frac{\Delta_k}{\Delta_j} \leq \frac{d_{jk}^2}{d_{kk}^2}$.*

*Then Assumptions 1.1 hold with constants*

$$c_0 = \min_k \alpha_k$$

$$C_{\mathrm{P}}^{(\beta)} = \frac{1}{\min_k a_k + \frac{\beta}{2}}.$$

*Proof.* We will show that each of the three parts of Assumptions 1.1 hold.

1. Holds by definition.

2. At the target level $\beta_L = 0$ the inequality $\int_\Omega \alpha_k \pi_k(x) dx \geq c_0 \sum_j \int_\Omega \alpha_j \pi_j(x) dx$ is equivalent to $\alpha_k \geq c_0 \implies c_0 = \min_k \alpha_k$. To show this for any $\beta_l$ we want to find the maximum $c_0$ that satisfies

$$\frac{\int_\Omega \alpha_k \pi_k(x) q_l(x - x_k) dx}{\sum_j \int_\Omega \alpha_j \pi_j(x) q_l(x - x_k) dx} \geq c_0.$$

We show this by finding an upper bound on the quotient

$$\frac{\int_\Omega \alpha_j \pi_j(x) q_l(x - x_k) dx}{\int_\Omega \alpha_k \pi_k(x) q_l(x - x_k) dx} = \frac{\alpha_j \left(\frac{a_j}{\pi}\right)^{\frac{d}{2}} \int_\Omega e^{-a_j \|x - \mu_j\|^2 - \beta_l \|x - x_k\|^2} dx}{\alpha_k \left(\frac{a_k}{\pi}\right)^{\frac{d}{2}} \int_\Omega e^{-a_k \|x - \mu_k\|^2 - \beta_l \|x - x_k\|^2} dx}$$

$$= \frac{\alpha_j \left(\frac{a_j}{\pi}\right)^{\frac{d}{2}} \left(\frac{\pi}{a_j + \beta}\right)^{\frac{d}{2}} \exp\left(\frac{\|a_j \mu_j + \beta x_k\|^2}{a_j + \beta_l} - a_j \|\mu_j\|^2 - \beta_l \|x_k\|^2\right)}{\alpha_k \left(\frac{a_k}{\pi}\right)^{\frac{d}{2}} \left(\frac{\pi}{a_k + \beta}\right)^{\frac{d}{2}} \exp\left(\frac{\|a_k \mu_k + \beta x_k\|^2}{a_k + \beta_l} - a_k \|\mu_k\|^2 - \beta_l \|x_k\|^2\right)}.$$

Rewriting the exponential terms $\exp\left(\frac{\|a_i \mu_i + \beta x_i\|^2}{a_i + \beta_l} - a_i \|\mu_i\|^2 - \beta_l \|x_i\|^2\right) = \exp\left(\frac{-a_i \beta_l}{a_i + \beta_l} \|\mu_i - x_i\|^2\right)$ yields,

$$= \frac{\alpha_j}{\alpha_k} \left(\frac{1 + \frac{\beta_l}{a_k}}{1 + \frac{\beta_l}{a_j}}\right)^{\frac{d}{2}} \exp\left(\frac{a_k \beta_l}{a_k + \beta_l} \|\mu_k - x_k\|^2 - \frac{a_j \beta_l}{a_j + \beta_l} \|\mu_j - x_k\|^2\right).$$

To simplify the above expression let $\Delta_i = \frac{\beta_l}{1 + \frac{\beta_l}{a_i}}$ and $d_{ik} = \|\mu_i - x_k\|$. The above is at most $\frac{\alpha_j}{\alpha_k}$ iff

$$\frac{d}{2} \log\left(\frac{\Delta_j}{\Delta_k}\right) + \Delta_k d_{kk}^2 - \Delta_j d_{jk}^2 \leq 0. \tag{I.1}$$

Consider 3 cases.

Case 1 (equal variance): Assume $a_k = a_j$. In this case $\Delta_k = \Delta_j$ and (I.1) is satisfied when $d_{kk} \leq d_{jk}$.

Case 2: $a_k \leq a_j$ which is equivalent to $\Delta_k \leq \Delta_j$. In this case,

$$\frac{d}{2} \log\left(\frac{\Delta_j}{\Delta_k}\right) + \Delta_k d_{kk}^2 - \Delta_j d_{jk}^2 \leq \frac{d}{2} \log\left(\frac{\Delta_j}{\Delta_k}\right) + \max\{\Delta_k, \Delta_j\}(d_{kk}^2 - d_{jk}^2)$$

which is at most 0 when

$$\frac{d}{2\max\{\Delta_k, \Delta_j\}} \log\left(\frac{\Delta_j}{\Delta_k}\right) \leq d_{jk}^2 - d_{kk}^2.$$

Case 3: $a_j \leq a_k$ which is equivalent to $\Delta_j \leq \Delta_k$. In this case,

$$\frac{d}{2} \log\left(\frac{\Delta_j}{\Delta_k}\right) + \Delta_k d_{kk}^2 - \Delta_j d_{jk}^2 \leq \Delta_k d_{kk}^2 - \Delta_j d_{jk}^2,$$

which is at most 0 when

$$\frac{\Delta_k}{\Delta_j} \le \frac{d_{jk}^2}{d_{kk}^2}.$$

In all three cases we have that

$$\frac{\alpha_j}{\alpha_k}\left(\frac{1+\frac{\beta_l}{a_k}}{1+\frac{\beta_l}{a_j}}\right)^{\frac{d}{2}}\exp\left(\frac{a_k\beta_l}{a_k+\beta_l}\|\mu_k-x_k\|^2 - \frac{a_j\beta_l}{a_j+\beta_l}\|\mu_j-x_k\|^2\right) \le \frac{\alpha_j}{\alpha_k}.$$

Since this holds for all $j,k$ we have that

$$\frac{\int_\Omega \alpha_k\pi_k(x)q_l(x-x_k)dx}{\sum_j \int_\Omega \alpha_j\pi_j(x)q_l(x-x_k)dx} \ge \frac{1}{\sum_j \frac{\alpha_j}{\alpha_k}} = \alpha_k.$$

Therefore since this must hold for all $k$ we have that $c_{tilt} = \min_k \alpha_k$.

3. Let $V_j = -\log\pi_j$. We have $\boldsymbol{\nabla}^2 V_j(x) \succeq \left(a_j+\frac{\beta}{2}\right)I$. Since $V_j(x)$ is $\alpha$-strongly convex with $\alpha = a_j + \frac{\beta}{2}$, then $\alpha_j\pi_j(x)e^{-\beta_l\|x-x_k\|^2}$ satisfies an Poincaré inequality (and log-Sobolev inequality) with $C_{\mathrm{P}}^{(\beta)} = \frac{1}{a_j+\frac{\beta}{2}}$.

$\square$

We note that the technical conditions required in Proposition I.8 pose geometric implications on the target measure. In essence, they require that the modes of the Gaussian mixture are sufficiently spaced relative to their respective variances. This is used to ensure that the tilt towards a singular mode does not carry significant mass from a different component. This is a limitation in our current analysis, as showing mode separation for a general mixture of $L$-smooth distributions poses difficulties.

This also suggests an avenue for future work in developing adaptive tilting schemes. For instance, in our work we choose $q_\beta$ so be isotropic Gaussians, a simple improvement can be choosing $q_\beta$ adaptively to have a covariance structure that matches the local geometry of the target measure.

## J APPENDIX

**Lemma J.1.** *((Levin et al., 2017, Lemma 13.6)) For a reversible transition matrix $P$ with stationary distribution $\pi$, the associated Dirichlet form is*

$$\mathscr{E}(f,f) = \frac{1}{2}\sum_{x,y\in\Omega}\left(f(x)-f(y)\right)^2\pi(x)P(x,y).$$

**Lemma J.2.** *(HCR inequality (Lehmann, 1983)) Let $P,Q$ be measures with $P \ll Q$ and $f$ a measurable function. Then*

$$\left(\mathbb{E}_Q(f)-\mathbb{E}_P(f)\right)^2 \le \mathrm{Var}_P(f)\chi^2(Q\|P).$$

**Lemma J.3.** *((Lee & Santana-Gijzen, 2024, Lemma 20)) If the Markov semigroup $P_t$ is reversible with stationary distribution $\pi$, then*

$$\frac{d^2}{dt^2}\mathrm{Var}_\pi(P_t f) \ge 0.$$

*Hence, $-\frac{d}{dt}\mathrm{Var}_\pi(P_t f) = -2\langle P_t f, \mathscr{L}P_t f\rangle$ is strictly decreasing.*

*Proof.* We compute

$$\frac{d^2}{dt^2}\mathrm{Var}_\pi(P_t f) = \frac{d}{dt}\left(\frac{d}{dt}\mathrm{Var}_\pi(P_t f)\right)$$

$$= \frac{d}{dt}2\langle P_t f, \mathscr{L}P_t f\rangle$$

$$= 2(\langle\mathscr{L}P_t f, \mathscr{L}P_t f\rangle + \langle P_t f, \mathscr{L}^2 P_t f\rangle)$$

Since $\mathscr{L}$ is self-adjoint,

$$= 4 \langle \mathscr{L} P_t f, \mathscr{L} P_t f \rangle$$
$$= 4 \| \mathscr{L} P_t f \|^2_{L^2(\pi)} \geq 0.$$

$\square$

**Lemma J.4.** *Let $P = \sum_{k=1}^M w_k P_k$ and $Q = \sum_{k=1}^M w'_k Q_k$ be distributions where $w_k, w'_k \geq 0$ and $P_k, Q_k$ are distributions. Suppose $\frac{w_k}{w'_k} \leq r$ for all $k$. Then*

$$\chi^2(P||Q) \leq r \cdot \sum_k w_k \chi^2(P_k||Q_k).$$

*Proof.* Consider

$$\chi^2(P||Q) = \int_\Omega \frac{\left(P(x) - Q(x)\right)^2}{Q(x)} dx$$
$$= \int_\Omega \frac{\left(\sum_k w_k P_k(x) - \sum_k w'_k Q_k(x)\right)^2}{\sum_k w'_k Q_k(x)} dx$$

Cauchy-Schwarz yields $\left( \sum_k \frac{w_k P_k(x) - w'_k Q_k(x)}{\sqrt{w'_k Q_k(x)}} \sqrt{w'_k Q_k(x)} \right)^2 \leq$

$\sum_k \frac{\left(w_k P_k(x) - w'_k Q_k(x)\right)^2}{w'_k Q_k(x)} \sum_k w'_k Q_k(x)$; therefore

$$\leq \int_\Omega \sum_k \frac{\left(w_k P_k(x) - w'_k Q_k(x)\right)^2}{w'_k Q_k(x)} dx$$
$$= \int_\Omega \sum_k \frac{w_k^2 P_k(x)^2}{w'_k Q_k(x)} dx - 2 \int_\Omega \sum_k \frac{w_k w'_k P_k(x) Q_k(x)}{w'_k Q_k(x)} dx + \int_\Omega \sum_k \frac{{w'_k}^2 Q_k(x)^2}{w'_k Q_k(x)} dx$$
$$= \int_\Omega \sum_k \frac{w_k^2 P_k(x)^2}{w'_k Q_k(x)} dx - 2 \int_\Omega \sum_k w_k P_k(x) dx + \int_\Omega \sum_k w'_k Q_k(x) dx$$
$$= \sum_k \frac{w_k^2}{w'_k} \int_\Omega \frac{P_k(x)^2}{Q_k(x)} dx - 1$$
$$= \sum_k \frac{w_k^2}{w'_k} \chi^2(P_k||Q_k) \leq r \sum_k w_k \chi^2(P_k||Q_k).$$

$\square$

**Lemma J.5.** *(Ge et al., 2018c) Let $p, q$ be probability distribution functions. Then*

$$\int_\Omega \min\{p(x), q(x)\} dx \geq 1 - \sqrt{\chi^2(q||p)}.$$

*Proof.* We have that

$$\left( \int_\Omega q - \min\{p, q\} dx \right)^2 \leq \int_\Omega \frac{(q - \min\{p, q\})^2}{q} dx \leq \chi^2(q||p).$$

Therefore

$$\int_\Omega \min\{p, q\} dx \geq 1 - \sqrt{\chi^2(q||p)}.$$

$\square$

**Lemma J.6.** *((Bakry et al., 2014; Bobkov & Tetali, 2006)) Suppose that $\mu$ satisfies a log-Sobolev inequality with constant $C_{LS}$. Let $f$ be a 1-Lipschitz function. Then*

1. *(Sub-exponential concentration) For any $t \in \mathbb{R}$,*

$$E_\mu e^{tf} \leq e^{t\mathbb{E}_\mu f + \frac{C_{LS}t^2}{2}}.$$

   *This holds in both the continuous ([Bakry et al., 2014](#)) and discrete ([Bobkov & Tetali, 2006](#)) setting.*

2. *(Sub-gaussian concentration) For any $t \in [0, \frac{1}{C_{LS}})$,*

$$\mathbb{E}_\mu e^{\frac{tf^2}{2}} \leq \frac{1}{\sqrt{1 - C_{LS}t}} \exp\left( \frac{t}{2(1 - C_{LS}t)} (\mathbb{E}_\mu f)^2 \right).$$

**Lemma J.7** (Method of Canonical Paths (Corollary 13.21, ([Levin et al., 2017](#)))**.** *Let $P$ be a reversible and irreducible transition matrix with stationary distribution $\pi$. Define $Q(x, y) = \pi(x)P(x, y)$ and suppose $\Gamma_{xy}$ is a choice of E-path for each $x$ and $y$, and let*

$$B = \max_{e \in E} \frac{1}{Q(e)} \sum_{x,y,e \in \Gamma_{xy}} \pi(x)\pi(y)|\Gamma_{xy}|.$$

*The the spectral gap satisfies $\gamma \geq B^{-1}$.*

## K    LLM USAGE

LLMs used for light editing and coding aid.

