## A  ORGANIZATION OF PAPER

We provide a brief overview of each section. In Section 2, we define the simulated tempering and ST Teleporting algorithms, along with the main algorithm used in this paper. In Section 3, we present the main results of our paper.

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

$$\tilde{\pi}_{l,k}(x) = \pi_k(x) \cdot q_l(x - x_k).$$

The partition functions corresponding to these measures are given by $\bar{Z}_{l,k}$ and $Z_{l,k}$ respectively. In Section G.3, we also make use of the unnormalized joint distribution over the temperature levels this is defined to be

$$\tilde{p}(x, i) = \sum_{j=1}^{l} r_j^{(l)} \tilde{p}_j(x) I\{i = j\}. \tag{B.2}$$

Similarly, we define the normalized version to be

$$p(x, i) = \sum_{j=1}^{l} \omega^j p_j(x) I\{i = j\}$$

where $\omega^j = r_j^{(l)} Z_j$. In the context of Section F, we will occasionally refer to the "good" part of the distribution and will denote this unnormalized portion by

$$\tilde{p}_0(x, i) = \sum_{i=1}^{l} r_i^{(l)} \sum_{k=1}^{M} \alpha_k w_{i,k} \pi_k(x) q_i(x - x_k) I\{i = j\}.$$

The marginal over the levels of the good portion is then denoted by

$$p_{i0}(x) \propto \sum_{k=1}^{M} \alpha_k w_{i,k} \pi_k(x) q_i(x - x_k). \tag{B.3}$$

We can then express the normalized joint distribution as a mixture of the good and bad portions by

$$p(x, i) = \alpha_0 p_0(x, i) + (1 - \alpha_0) p_1(x, i)$$

where $\alpha_0 = \frac{\sum_i \int_\Omega \tilde{p}_0(x,i) dx}{\sum_i \int_\Omega \tilde{p}(x,i) dx}$ is the component weight of the good portion.

## B.2 MOTIVATING EXAMPLES

To motivate tempering to colder temperatures, corresponding to more peaked distributions, we show that in high dimensions, flat components of the mixture distribution can cause the teleport process to have low acceptance probabilities. This leads to poor mixing of the projected chain—mixing between modes—which in turn leads to long run times. In our algorithm, the projected chain at the coldest level is the probability flow between component measures after an affine translation which overlaps the warm start points.

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

} \Big( f_1 \circ g_{jj'}(x) - f_1(x) \Big)^2 \min\Big\{ P_1 \circ g_{jj'}(x), P_1(x) \Big\} dx$$

$$+ 12C \sum_{i=1}^{L} r_i \mathscr{E}_i(f_i, f_i) + 3 \sum_{\substack{1 \leq i \leq L \\ i'=i\pm1}} r_i w_{ij} \Big( \mathbb{E}_{P_{i'j}}(f_{i'}) - \mathbb{E}_{P_{ij}}(f_i) \Big)^2 \int_{\Omega} \min\Big\{ r'_i P_{i'}(x), r_i P_i(x) \Big\} dx \Bigg]$$

Grouping like terms and comparing to the Dirichlet form in Corollary D.7

$$\leq C(1 + \bar{C}(6M + 12)) \sum_{i=1}^{L} r_i \mathscr{E}_i(f_i, f_i)$$

$$+ \frac{6\bar{C}M}{\gamma} \frac{\gamma \cdot r_1}{2M} \sum_{j=1}^{M} \sum_{j'=1}^{M} \int_{\Omega} \min\Big\{ P_1 \circ g_{jj'}(x), P_1(x) \Big\} \Big( f_1 \circ g_{jj'}(x) - f_1(x) \Big)^2 dx$$

$$+ \frac{12\bar{C}}{\lambda} \frac{\lambda}{4} \sum_{\substack{1 \leq i \leq L \\ i'=i\pm1}} r_i w_{ij} \Big( f_{i'}(x) - f_i(x) \Big)^2 \int_{\Omega} \min\Big\{ r'_i P_{i'}(x), r_i P_i(x) \Big\} dx \Bigg]$$

By applying the dirichlet form in Corollary D.7

$$\leq \max\left\{ C(1 + (6M + 12)\bar{C}, \frac{6M\bar{C}}{\gamma}, \frac{12\bar{C}}{\lambda} \right\} \mathscr{E}(f, f)$$

$$\square$$

## F   LOCAL CONVERGENCE FOR A MARKOV PROCESS

In this section we show that for a Markov chain $P_t$, with stationary mixture distribution $\pi = \sum_k w_k \pi_k$, the weight adjusted distribution $p_{T,0} = p_T \frac{\pi_0}{\pi} / \int_{\Omega} p_T \frac{\pi_0}{\pi}$ converges to the component $\pi_0$ in chi-squared divergence, where $p_T = \nu_0 P_t$ for some initial probability measure $\nu_0$. This can be applied to the mixture measure $\tilde{p}_\beta(x) \propto \pi(x) \cdot \sum_{k=1}^{M} w_{\beta,k} q_\beta(x - x_k)$, on the extended state space $\Omega \times [L]$. Since on the target level $\beta_L = 0$, there is no bad component, the divergence $\chi^2(p_{T,good} \| p_{good})$ provides us with a good indication of how close we are at the target level. The following Lemma provides us with an upper bound on $\chi^2(p_{T,good} \| p_{good})$ in terms of the Poincaré constant on $p_{good}$.

**Lemma F.1.** *Suppose that for a Markov generator $\mathscr{L}$, with stationary distribution $\pi = \sum_{k=0}^{\ell} \alpha_k \pi_k$, $\langle f, \mathscr{L} f \rangle_\pi \leq \sum_k \alpha_k \langle f, \mathscr{L}_k f \rangle_{\pi_0}$ for all $f$, where $\mathscr{L}_0$ has stationary measure $\pi_0$ and Poincaré constant $C$. Let $\overline{p}_T$ be the distribution of $X_t$ where $t \sim \mathsf{Unif}(0, T)$, $K_0 = \chi^2(p_0 \| \pi)$, and $\overline{p}_{T,0} = \overline{p}_T \frac{\pi_0}{\pi} / \int_{\Omega} \overline{p}_T \frac{\pi_0}{\pi}$. Then*

$$\mathrm{Var}_{\pi_0}\left( \frac{\overline{p}_T}{\pi} \right) \leq \frac{K_0 C}{\alpha_0 T}$$

*or equivalently,*

$$\chi^2(\overline{p}_{T,0} \| \pi_0) \leq \frac{K_0 C}{\alpha_0 \left[ \mathbb{E}_{\overline{p}_T}\left( \frac{\pi_0}{\pi} \right) \right]^2 T}.$$

*Proof.* We have that

$$\frac{d}{dt}\chi^2(p_t\|\pi) = -\left\langle \frac{p_t}{\pi}, \mathscr{L}\frac{p_t}{\pi} \right\rangle,$$

so

$$K_0 = \chi^2(p_0\|\pi) \geq \int_0^T \left\langle \frac{p_s}{\pi}, \mathscr{L}\frac{p_s}{\pi} \right\rangle ds$$

$$\geq \int_0^T \sum_{k=0}^{\ell} \alpha_k \left\langle \frac{p_s}{\pi}, \mathscr{L}_k\frac{p_s}{\pi} \right\rangle ds$$

$$\geq \frac{1}{C} \int_0^T \sum_{k=0}^{\ell} \alpha_k \operatorname{Var}_{\pi_k}\left(\frac{p_s}{\pi}\right) ds$$

$$\geq \frac{\alpha_0}{C} \int_0^T \operatorname{Var}_{\pi_0}\left(\frac{p_s}{\pi}\right) ds$$

$$\geq \frac{\alpha_0 T}{C} \mathbb{E}_{s\sim\mathsf{Unif}(0,T)} \operatorname{Var}_{\pi_0}\left(\frac{p_s}{\pi}\right)$$

$$\geq \frac{\alpha_0 T}{C} \operatorname{Var}_{\pi_0}\left(\frac{\overline{p}_T}{\pi}\right). \tag{F.1}$$

Now

$$\operatorname{Var}_{\pi_0}\left(\frac{\overline{p}_T}{\pi}\right) = \operatorname{Var}_{\pi_0}\left(\frac{\overline{

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

 \le \left(\frac{\beta_1}{\beta_1 + L}\right)^{\frac{d}{2}} \iff \beta_1 \ge \frac{L}{\left(\frac{1}{1-\epsilon}\right)^{\frac{2}{d}} - 1}.$$

Noting that $\left(\frac{1}{1-\epsilon}\right)^{\frac{2}{d}} = 1 + \Theta\left(\frac{\epsilon}{d}\right)$ gives that it suffices for $\beta_1 = \Omega\left(\frac{Ld}{\epsilon}\right)$. In similar fashion, for the upper bound, we choose $\beta_1$ such that

$$1 + \epsilon_1 \ge \left(\frac{\beta_1}{\beta_1 - L}\right)^{\frac{d}{2}} \iff \beta_1 \ge \frac{L}{\left(\frac{1}{1+\epsilon}\right)^{\frac{d}{2}} + 1}.$$

A similar analysis to the lower bound yields that this is satisfied when $\beta_1 = \Omega\left(\frac{Ld}{\epsilon_1}\right)$. We also impose

$$1 + \epsilon_2 \ge e^{\frac{L^2 D^2}{2(\beta_1 - L)}} \iff \beta_1 = \frac{L^2 D^2}{2\