# OpenReview forum: "Sampling from multimodal distributions with warm starts"
_ICLR.cc/2026/Conference — Submitted to ICLR 2026_

### Official Review · Reviewer_i8Tb · 2025-10-16

**Soundness:** 2
**Presentation:** 1
**Contribution:** 2
**Rating:** 2
**Confidence:** 1

**Summary:**

This paper provides nonasymptotic analysis to ALPS. The main statement upper-bounds the total number of iterations and the number of samples for sampling from a stationary distribution pi.

**Strengths:**

Understanding the complexity of sampling from mixture distribution is an important research topic and can have wide applications.
The algorithm studied in this framework is practical and when proper assumptions are made, enjoy fast convergence.

**Weaknesses:**

This submission is very difficult to follow. I have the following questions which prevent me from understanding the work well, or giving higher scores.

1. Notations are confusing. Examples:
a. Mixtures are indexed sometimes as k = 1,..., M, or i = 1,...,
b. g_# is not defined.
c. Which markov chain is Asumption1.1.4 referring to? Only distributions are defined so far. Is this the Markov chain defined by later algorithms?
2. Assumptions are not rarely discussed. For example,
a.when does a Markov chain satisfy Asumption1.1.4? Does it hold for any Langevin dynamics?
b.When can warm starts be achieved? If q_1(x) is Gaussian, and pi_i are some delta distributions, how does ctilt scale with radius and dimension?
c. when does p_beta,i has better poincare coeff than pi_i? The expression for p_beta,i looks like a simple reweighting rather than smoothing, why would it permit faster mixing?
3. The algorithm is hard to understand, especially given Alg 2, 3 are in supplementary.
4. The theorem 3.3 bounds total number of samples, but does it bound total run time? Alg 2 can fail and require rerun. How can one bound number reruns?

Aside from all the above questions which stop me from understanding what is going on. I have a hard time positioning this work agains other analysis for sampling from mixture distributions.
1. Can the authors compare the algorithm against some known ones (e.g. EM with random init + restart?) on some simple settings? When does it lead to better computation / mixing?
2. Compared to naive Langevin dynamics with temperature ~ epsilon, does the proposed algorithm converge faster?
3. What are some special distribution familiies for which all assumption are satisfied?
4. Can it be compared to other nonasymptotic rates for studying mixing on mixture distributions using Markov decomposition theorem?

At this stage, I cannot confidently judge the quality of the paper, and I think it might be very confusing for general readers as well.

**Questions:**

Please see weaknesses

---

> ### Author Response · Authors · 2025-11-20
> **Response (Weaknesses)**
>
> Q: "Notations are confusing. Examples: a. Mixtures are indexed sometimes as k = 1,..., M, or i = 1,..., b. g_# is not defined. c. Which markov chain is Asumption1.1.4 referring to? Only distributions are defined so far. Is this the Markov chain defined by later algorithms?"
>
> A: Yes you are correct to point out that we changed notation in mixture indexing and we will correct this. g_# denotes the pushforward operator on measures. We use this in the context of our teleportation map and formally define it in the revision.
> Assumption 1.1.4 is a pre-condition on the type of Markov process we will use later, not a reference to a chain we have already used. This assumption states the the local Markov chain (level i chain) we choose for our algorithm must have a generator which decomposes over the mixtures. As we briefly noted this condition is known and satisfied for many common and practical Markov processes, such as Langevin diffusion and Metropolis Random Walk. We will clarify this point.
>
> Q: "Assumptions are not rarely discussed. For example, a.when does a Markov chain satisfy Asumption1.1.4? Does it hold for any Langevin dynamics? b.When can warm starts be achieved? If q_1(x) is Gaussian, and pi_i are some delta distributions, how does ctilt scale with radius and dimension? c. when does p_beta,i has better poincare coeff than pi_i? The expression for p_beta,i looks like a simple reweighting rather than smoothing, why would it permit faster mixing?"
>
> A: Our paper addresses the sampling problem given warm starts. We do not focus on the "search problem" of finding them. In practice, these warm starts are typically obtained from multiple runs of standard optimization algorithms or from prior domain knowledge.
>
> In our framework, the \pi_i​ are not delta distributions; they are general probability distributions (e.g., Gaussian, Student's t, etc.) that form the components of the target mixture \pi(x). The constant c_tilt measures the fraction of the mass from the correct component \pi_k that is captured when we tilt the entire distribution \pi(x) toward the warm start. In our analysis of a mixture of Gaussians (proposition I.8) we show that this is directly related to the minimum component weight \min \alpha_k.
>
> We do not claim p_\beta,i mixes locally faster than \pi_i. What we claim is that the approximation of the partition function of the tilted component enables level/component balancing. In turn, this guarantees that the projected chain is free of bottlenecks between levels and hence mixes rapidly. In short the reweighting is not for smoothing; it is for provably eliminating bottlenecks.
>
> Q: "The algorithm is hard to understand, especially given Alg 2, 3 are in supplementary."
>
> A: Algorithms 2 and 3 were in the supplementary material for space reasons. The updated paper contains a more detailed description of these algorithms.
>
> Q: "The theorem 3.3 bounds total number of samples, but does it bound total run time? Alg 2 can fail and require rerun. How can one bound number reruns?"
>
> A: The runtime is indeed polynomially bounded. Theorem 3.3 provides this bound for acquiring one sample. The total run time to acquire all samples is then polynomially bounded, as the time to acquire one sample follows a geometric distribution where p = probability of being at level L (target level). Our reweighting scheme makes this p approximately 1/L.
>
> In practice, you would not actually re-run the chain from scratch. This design is explicitly for analytical purposes. A practitioner should instead run the chain once for a burn in of $T$ and then continue running the mixed chain. Thinning the collected samples will prevent autocorrelation. This is standard practice for analysis vs. implementation for most MCMC algorithms.

---

> ### Author Response · Authors · 2025-11-20
> **Response (Questions)**
>
> Q: "Can the authors compare the algorithm against some known ones (e.g. EM with random init + restart?) on some simple settings? When does it lead to better computation / mixing?"
>
> A: The algorithm we proposed (a sampler) and the Expectation-Maximization (EM) algorithm (an optimizer) are designed for fundamentally different tasks. The EM algorithm is used to find the parameters of a mixture model (i.e. learn the target distribution). Our algorithm addresses the sampling problem—sampling given the target distribution.
>
> If you are looking for comparison to other sampling algorithms we show that our algorithm works in polynomial time. This is the gold theoretical standard for the sampling literature. A more precise comparison would be fitting if the goal of our work was to establish tight computational bounds.
>
> Q: "Compared to naive Langevin dynamics with temperature ~ epsilon, does the proposed algorithm converge faster?"
>
> A: Yes, of course. Naive Langevin dynamics is in general extremely slow with multimodal target distributions. In fact, naive Langevin dynamics can have mixing time exponential in the distance between the modes of a distribution, which is well-studied in the theory of metastability.
>
> Q: "What are some special distribution familiies for which all assumption are satisfied?"
>
> A: We show that this is satisfied for mixtures of $e^{-f_i(x)}$ where $f_i(x)$ is L-smooth (Section F) and then explicitly verify the constants in our assumptions for the case when the target mixture is a mixture of gaussians (proposition I.8).
>
> Q: "Can it be compared to other nonasymptotic rates for studying mixing on mixture distributions using Markov decomposition theorem?"
>
> A:Yes, this is a central contribution to our work. We show that in the no advice setting, torpid mixing is a common phenomenon even in the simplest of mixtures (example B.2). We crucially show that by utilizing weak advice, it is possible to overcome these bottlenecks.

---

> > ### Comment · Reviewer_i8Tb · 2025-11-21
> > **Response**
> >
> > I thank the authors for the reply.
> >
> > After the revision, I can partially understand what the main theorem and the algorithm is about.
> >
> > On the other hand, I still don't understand when should the algorithm be applied. As the authors stated that the main contribution is theoretically proving that the mixing time and computation is polynomial, can the author give a simple but nontrivial example to justify this by plugging in all the constants that potentially depends on problem dimension, lipschitzness, radius etc?
> >
> > I will try to explain what "nontrivial" means. I am looking for an example where we do not know enough to directly sample from the distribution as in (i) below, but we know some structure to provide (poly-time) efficient warm initialization, and also run ALPS with polynomially-bounded constants in assumptions. In other words, an example where direct sampling is inaceesible, Alg1 converges provably in poly compute, and Langevin dynamics is too slow.
> >
> > (i)For example, if I have p = \sum_i w_i p_i, where pi are Gaussians, then the sampling problem is usually trivial if we know w_i and p_i.  In the example in line 435, one can also easily sample w first, and then sample either Cauchy or Gaussian later.
> >
> > (ii)Langevin dynamics can be slow for simple Gaussian mixtures, but it can work when we can't evaluate p(x), but can only evaluate \grad f(x).
> >
> > Besides, I still think the current manuscript can be too difficult to understand for readers like me (I am sufficiently familiar with the literature of analyzing Langevin dynamics and their sample complexity of mixing). Therefore, I still would not recommend accepting the work. I will keep the confidence score low in case other reviewers can easily follow the presentation.

---

> ### Author Response · Authors · 2025-11-26
> **Justification of multimodality assumption**
>
> The mixture assumption $\pi(x) = \sum_k \alpha_k \pi_k(x)$ is a theoretical framework and not a requirement for the algorithm's implementation. Crucially, we emphasize that our algorithm operates in a black-box setting; it does not require knowledge of the weights $w_i$, the component densities $\pi_i$, or the functional form of the decomposition.
>
> An implicit thesis behind the line of work on theoretical analysis of sampling from multimodal distributions (see Section 1.1 for references) is that this mixture assumption is a good setting to study the behavior of algorithms on multimodal distributions, and that improving the complexity of algorithms in this setting will translate to improvements of sampling multimodal distributions “in the wild”. (Note that in all these analyses, the algorithm is not given access to the decomposition; all that is required is the existence of the decomposition, which is by no means unique.) This is similar to how log-concave distributions are a model problem for studying sampling algorithms on unimodal distributions. We emphasize that this setting is not new to the paper--only the “warm start” assumption is.
>
> We give some heuristic support for this thesis: First, another possible assumption that confers multimodality is that there are only k low-lying eigenvalues, and that; this can be thought of in some sense as more “universal” assumption in that each is a distinct “barrier” to mixing, and the mixture assumption implies this (see [1]). It turns out that conversely, this roughly implies the existence of a decomposition (see [2,3]). Second, for a distribution with k modes, the theory of meta-stability says that under mild non-degeneracy conditions, in the limit of low temperature, there are exactly k low-lying eigenvalues (Propositions 2.1, 2.2 in Chapter 8 of [4]), which gives such a decomposition by the previous point.
>
> Examples: (While our applications section focused on the continuous case, note that our theory works equally for discrete distributions on $\lbrace 1,...,q\rbrace^n$.)
> 1. Power posterior distribution for a mixture of two gaussians: See [5], whose proof proceeds by such a decomposition in the direction of the true mean.
> 2. Low-temperature Ising model: See [6], which establishes mixing within each of two phases (components) up to low-probability regions.
> 3. Low-temperature Ferromagnetic Potts model: See Section 5.3 in [7]; components correspond to colors.
>
> Although the symmetry in these models means that sampling can be done by locally sampling and then resampling the sign/color with equal probability, this can be simply broken in all these cases by a tilt to the distribution, which could result from e.g. having a prior distribution favoring one direction. Our approach is robust to such reweighting. Warm starts for 2, 3 are the monochromatic configurations. Another setting is when we have samples from a distribution but wish to sample from a modified version of it, for example, when fine-tuning a generative model or incentivizing certain features, so the original samples are warm starts but no longer accurate samples.
>
> As this modeling assumption is not the focus of the work, we added a brief explanation to Section 1.1 and 1.2 to explain this modeling assumption. We believe our example captures the essential ideas without the additional work required to flesh out the additional examples above.
>
> [1] Frederic Koehler, Holden Lee, and Thuy-Duong Vuong. Efficiently learning and sampling multimodal distributions with data-based initialization. In Nika Haghtalab and Ankur Moitra (eds.), Proceedings of Thirty Eighth Conference on Learning Theory, volume 291 of Proceedings of Machine Learning Research, pp. 3264–3326. PMLR, 30 Jun–04 Jul 2025.
>
> [2] Lee, James R., Shayan Oveis Gharan, and Luca Trevisan. "Multiway spectral partitioning and higher-order cheeger inequalities." Journal of the ACM (JACM) 61.6 (2014): 1-30.
>
> [3] Miclo, Laurent. "On hyperboundedness and spectrum of Markov operators." Inventiones mathematicae 200.1 (2015): 311-343.
>
> [4] Kolokoltsov, Vassili N. Semiclassical analysis for diffusions and stochastic processes. Springer, 2007.
>
> [5] Mou, Wenlong, et al. "Sampling for Bayesian mixture models: MCMC with polynomial-time mixing."
>
> [6] Gheissari, Reza, and Alistair Sinclair. "Low-temperature Ising dynamics with random initializations." Proceedings of the 54th Annual ACM SIGACT Symposium on Theory of Computing. 2022.
>
> [7] Chen, Z., Galanis, A., Goldberg, L. A., Perkins, W., Stewart, J., & Vigoda, E. (2021). Fast algorithms at low temperatures via Markov chains. Random Structures & Algorithms, 58(2), 294-321.

---

> ### Author Response · Authors · 2025-11-26
>
> To elaborate on our Gaussian mixture example in Section I.2:
>
> (i) In our setting, a practitioner is only given access to a zero-th order oracle (unnormalized) $p(x)$—not a mixture decomposition with corresponding weights. Under only query access, where weights/mixture components are unknown, Langevin dynamics would only acquire samples from one mode, while our algorithm works in polynomial time.
>
> (ii) In the general hierarchy of information, having unnormalized $p(x)$ is considered less information than having access to the gradient. This is sufficient for us, as Metropolized Random Walk satisfies our Markov decomposition assumption. We show that with additional information (gradient access), Langevin Dynamics also satisfies our Markov decomposition assumption.

---

> ### Comment · Reviewer_i8Tb · 2025-11-26
> **Thanks**
>
> I thank the author again for the response.
>
> I agree with the authors, that Alg1 in this manuscript, like many MCMC methods such as Gibbs, MH, etc require only un normalized p(x), which makes it generally applicable.
>
> My question, however, is whether the author could give a nontrivial example for which the analyses formally apply. In particular, this would require:
> 1. We do not know enough about the mixture to directly sample from it.
> 2. Baselines such as Gibbs sampling / Langevin sampling mixes slowly.
> 3. There is an efficient (poly-time) way to find good advice and balanced initial weights that satisfies the assumptions in the analyses.
>
> I hope the author could see that my concern is not about the mixture assumption, nor about the algorithm implementation. Instead, as the authors' main contribution is in theory, I hope to what specific new problem was solved.
>
> Having such an example would make the contribution much clearer. Otherwise, the  analyses seem to only provide a complicated solution to already solved problems. The ising model / gibbs measure examples are fine, but how could one approximate the distribution with mixtures and find good initial information to provably kick-off the algorithm?

---

> ### Author Response · Authors · 2025-11-26
>
> We thank the reviewer for the opportunity to clarify. We want to emphasize that our primary focus is to resolve a fundamental theoretical problem in the sampling literature, rather than to engineer a solution for a specific model. The literature has established that multimodal distributions pose a fundamental difficulty in sampling without advice. Our contribution is to formalize the setting of weak advice and show that with weak advice our algorithm works in a general setting where other MCMC methods fall short.
>
> With this in mind, the example we provided in Section 5 is in fact non-trivial—note in the updated version of the paper we provided a new example which we felt better emphasized the benefits of our algorithm. I will address each bullet point directly:
>
> 1) Assume you are only given oracle access to $\pi(x)$ as defined in section 5. $\pi(x)$ has a latent mixture decomposition as defined, however, this decomposition is not known to the practitioner—hence, sampling from the weights uniformly than the mixture components is not a possibility.
> Gibbs sampling and Langevin dynamics mix slowly on $\pi(x)$ as there are geometric barriers between the modes causing metastability. In other words, samples rarely mix from the mode centered at $\vec{15}$ to the mode centered at $-\vec{15}$.
>
> 2) The most common sampling solution to overcoming these geometric barriers is Simulated Tempering. However, Woodard et al. 2009b, shows that Simulated Tempering can torpidly mix even in the simplest case of bimodal gaussians with mismatched weights and variances (Example B.2 in the appendix). ALPS with Hessian-adjusted Tempering (Tawn 2021 & Tawn 2020a) proposes a solution to the torpid mixing presented in Woodard et al. 2009b. However, their results rely on Gaussian approximations of the latent mixture decomposition, via Hessian estimation. They make a point of this as a primary shortcoming of their algorithm as it disallows sampling from more complex component functions.
> Our proposed reweighted ALPS algorithm overcomes the torpid mixing presents in Woodard et al. and does so without requiring Gaussian approximation via Hessian estimation. In Section 5, $\pi(x)$ is constructed so that the latent decomposition has a non-zero Hessian at the warm start points. However, the ALPS algorithm fails to proportionally sample from each mode since the Hessian is a poor approximation of mixture components true covariance (see Figure 3). By comparison, our algorithm does not depend on this misleading information and is able to sample proportionally from $\pi(x)$.
>
> 3) This is a good question and drives at the heart of why sampling from multimodal distributions can be exponentially difficult. Given a distribution $\pi(x)$ with a latent decomposition $\pi(x) = \sum_k w_k \pi_k(x)$ the key difficulty is acquiring samples with the correct proportionality. In other words, if binned correctly, the samples should be a good empirical approximation of the weights $w_k$. This is the exact failure point in Woodard et al. 2009b, when annealing to warmer temperatures, the weights of the latent decomposition become distorted by standard annealing methods, causing exponential bottlenecks. This failure case for sampling is not a problem for the optimization problem of acquiring good advice. In our example $\pi(x)$ can be first annealed to warm temperatures to acquire a set of warm start points—this is polynomial time—and then those warm starts can be used for sampling efficiently. This is addressed in the ALPS paper (Tawn 2021), however, their focus was more algorithmic where our focus is on obtaining nonasymptotic bounds given weak advice.

---

> > ### Comment · Reviewer_i8Tb · 2025-11-27
> > **Thanks**
> >
> > Thank you for the response.
> >
> > I think the example in section 5 is nontrivial if one only has access to pi(x) instead of the mixtures.
> >
> > However, could the authors make the discussion more formal. In particular, how can one, in polynomial time and sample,  find "advice" and initial weights good enough to satisfy the assumptions in the analyses?

---

> ### Author Response · Authors · 2025-12-03
>
> We thank the reviewer for the opportunity to clarify our work. We view this as a two part problem: finding weak advice and sampling in polynomial time given weak advice. The primary focus of our paper was to show that weak advice allows for polynomial time sampling solutions—a problem that has remained open for multimodal distributions. However, we understand the reviewers concerns and will remark on this in the section I setting: mixtures of measures with L-smooth potentials. Where in this setting, simulated annealing converges to the mode in polynomial time (See "Escaping from Saddle Points," Ge et al. 2015).
>
> We would also like to clarify that the initial weights are not found separately from the warm start points. The initial weights are a function $w_{1,k} \propto 1/\pi(x_k)$ of the warm start points. We argue that this is a good approximation of a peaked gaussian distribution if the coldest temperature is chosen as $\beta_1 = \Omega(L^2D^2d/\epsilon)$ (Lemma I.4).

---

### Official Review · Reviewer_qZeH · 2025-10-22

**Soundness:** 3
**Presentation:** 2
**Contribution:** 2
**Rating:** 4
**Confidence:** 2

**Summary:**

Sampling method based on ALPS for sampling from multi-modal distributions given a set of "warm-starts" $\{x_i\}$ (points near the modes). In simulated tempering (or most tempering/annealing type sampling algorithms) sampling from multi-modal distributions is handled by increasing the temperature of the sampling distribution, allowing jumps between modes. Here instead they decrease the temperature of the target distribution so that it becomes very peaked around the warm start points and then provide a way for the sampler to "teleport" from between warm start points when at the coldest temperature. This method is an extension of ALPs to deal with distributions where the probability of moving between modes becomes vanishingly small at some temperature levels.

**Strengths:**

- This proposed method directly addresses a weakness of the previous method ALPS and is a novel, if incremental contribution building on ALPS
- I liked that limitations to their method are discussed in the conclusion with an eye to further work

**Weaknesses:**

- The distribution in the experiments gets directly at the weakness of the previous method and comparative strength of the present method, but I would have liked to have seen a "real world" example where this method is needed.
- Without reading the appendix following this paper is quite challenging, especially the algorithm which depends heavily on two other algorithms only defined in the appendix.
- This method seems quite computationally inefficient, needing up to three ALPs runs per main algorithm iteration (if I am understanding correctly that algorithm 2 is run three times in algorithm 3)

Small issues:
- Line 371 "We accompany this provide a lower bound..."

**Questions:**

Would this also work in the case when j is not uniformly sampled (2.2)?

"We ran each algorithm for the same amount of time, with the same burn-in length and Metropolis random walk steps." Here I presume you mean for the same number of iterations as figure 1 shows, does this include the iterations needed when algorithm [3] runs algorithm [2] multiple times?

---

> ### Author Response · Authors · 2025-11-20
> **Response (Weaknesses)**
>
> Q: "The distribution in the experiments gets directly at the weakness of the previous method and comparative strength of the present method, but I would have liked to have seen a "real world" example where this method is needed."
>
> A: You are  correct that our experiment in Section 5 was not designed as a "real-world" application. Rather, it was intentionally designed to isolate and test the exact theoretical failure mode of existing methods. We agree that applying this method to a “real world” problem is the critical next step. However, we view our work as establishing the foundational theory that is necessary before such a practical, robust algorithm can be built.
>
> As we discussed in our conclusion (section 6), making this algorithm practical for real-world use would require further innovations—such as online weight updates to improve computational efficiency and adaptive tilting schemes to handle complex geometries. We believe that our paper provides the theoretical confidence that this is a promising direction for future work.
>
> Q: "Without reading the appendix following this paper is quite challenging, especially the algorithm which depends heavily on two other algorithms only defined in the appendix."
>
> A: We attached the appendix to the paper and added a more detailed description of the algorithms to the main paper.
>
> Q: "This method seems quite computationally inefficient, needing up to three ALPs runs per main algorithm iteration (if I am understanding correctly that algorithm 2 is run three times in algorithm 3)"
>
> A: Yes, your understanding of the pseudocode is correct. Algorithm 3, does require running algorithm 2 for 3 separate sets of samples. The pseudocode was written for analytical clarity, not computational efficiency. In practice, we only require two separate runs of algorithm 3, one for modal and level weight estimation (reusing samples for both) and a second one for rebalancing via empirical occupancy. In practice, algorithm 2 does not need to start fresh for every sample. Like most MCMC algorithms this is what’s considered for analytical purposes; however running the algorithm with a warm start and then thinning the samples is sufficient.
>
> While these practical changes help slightly, we acknowledge that running the chain twice at each level can be costly. However, we point to the fact that this algorithmic design is exactly what guarantees that our algorithm will not encounter exponentially bad mixing time. In summary, we trade a second MCMC run, which has provable polynomial run time, for one which is potentially exponentially bad in the dimension of the problem. Lastly, as noted in our conclusion section, we believe a practical version with online weight estimation is a promising direction for future work.

---

> ### Author Response · Authors · 2025-11-20
> **Response (Questions)**
>
> Q: "Would this also work in the case when j is not uniformly sampled (2.2)?"
>
> A: Yes, it would. Our choice to sample the starting mode j uniformly was a simplification made for ease of theoretical analysis, as we state on lines  251-253. The more standard approach, used in the original ALPS algorithm, is to assign the starting mode j based on the current position (by finding the closest warm start, j = arg min_k d(x,x_k)). This alternative, non-uniform-j approach would also result in a valid reversible Markov chain. Additionally, the core principles of our analysis (Section E)  would still apply but the mathematical form of the generator and its corresponding Dirichlet form (section D) would be more complex.
>
> Q: "'We ran each algorithm for the same amount of time, with the same burn-in length and Metropolis random walk steps.' Here I presume you mean for the same number of iterations as figure 1 shows, does this include the iterations needed when algorithm [3] runs algorithm [2] multiple times?"
>
> A: This is a great clarifying question. To be clear: no, the comparison does not include the iterations from the learning step (Algorithm 3). Since our algorithm uses less information we consider this a separate, one-time preprocessing cost that is unique to our method. However, it should be noted that in cases where our method alleviates a bottleneck in reaching the target level, the time cost of preprocessing + sampling is significantly less than sampling with just vanilla ALPS. In the case where the bottleneck appears between modes vanilla ALPS runs faster (no preprocessing step) however, it fails to actually obtain a set of representative samples. In practice, this preprocessing step does not take long at all (a few minutes) for 8 levels and $N=1500$ samples (our example setting).

---

> > ### Comment · Reviewer_qZeH · 2025-11-24
> > **Response**
> >
> > Thank you for your reply.
> >
> > These clarifying remarks have improved my understanding of the aims and content of this paper. After this I have revised my rating as I believe this work to have interest to the wider community and provides the theoretical basis for further work.

---

### Official Review · Reviewer_NdJv · 2025-11-01

**Soundness:** 3
**Presentation:** 2
**Contribution:** 3
**Rating:** 6
**Confidence:** 2

**Summary:**

This paper studies a particular scenario for non-log-concave sampling: assuming the target distribution is multimodal but within each mode there is a known point that can be used to serve as a warm start. The paper proposes an algorithm based on simulated tempering with an annealing path that connects the target distribution to a mixture of delta distributions at the known warm start points, and leverages the ALPS algorithm with weight estimation to sample from the target distribution. The authors establishes polynomial mixing time guarantees for the proposed algorithm under certain assumptions that can be satisfied by mixtures of log-smooth densities with Gaussian tempering. There are also experimental results that demonstrate the empirical performance of the proposed algorithm compared to the ALPS algorithm.

**Strengths:**

The paper is well-written, although not to easy to read for non-experts, and requires a good understanding of MCMC sampling techniques, especially simulated tempering and ALPS. As I don't have enough knowledge about related works, I cannot fully assess the originality of the paper. The theoretical results in the paper are rigorously presented with detailed proofs in the appendix, although I don't have enough expertise and time to verify all the technical details. The warm start assumption has not been well-studied in the literature, and I believe it is a practical assumption in many real-world applications (e.g., sampling from distributions in statistical physics or molecular dynamics, where a few known samples can be obtained from prior simulations or experiments). Finally, the authors also discuss potential research directions in the conclusion section, which is insightful. I generally think this is a good paper, but I don't have enough confidence to recommend strong acceptance due to my limited knowledge about related works. It's better suited for more theoretical venues like COLT or STOC rather than ICLR.

**Weaknesses:**

As mentioned above, I suggest the authors to provide more context and discussion about related works on non-log-concave sampling, especially those that also consider annealing. Also, many of the details are postponed to the appendix (such as Algorithms 2 and 3, which are important for understanding the proposed method), making it hard to follow. I suggest the authors to combine the pdfs of the main paper and the appendix into a single document for easier reading.

**Questions:**

Some minor comments: on line 161, the $q _ \beta$'s are actually not Gaussian densities but Gaussian-like, as later in proposition 3.2 $q _ \beta=\mathrm{e}^{-\beta||x||^2/2}$ is considered without normalization, and clearly $q _ 0$ is not the delta distribution. On line 237, $g _ {jj'}(x)$ seems to be $x-x _ j+x _ {j'}$ judging from the definition D.1 and the later usage in proposition 3.2.

---

> ### Author Response · Authors · 2025-11-20
> **Response**
>
> Weaknesses:
>
> Q: "As mentioned above, I suggest the authors to provide more context and discussion about related works on non-log-concave sampling, especially those that also consider annealing. Also, many of the details are postponed to the appendix (such as Algorithms 2 and 3, which are important for understanding the proposed method), making it hard to follow. I suggest the authors to combine the pdfs of the main paper and the appendix into a single document for easier reading."
>
> A: Our paper's focus is specifically on multimodal distributions, which are an important and challenging subclass of the broader family of non-log-concave distributions. More specifically, we are focused on how the chain mixes between distributions—as opposed to within a single non-log-concave mixture component (this is a separate problem).  Furthermore, our main contribution is to analyze the setting where "weak advice" in the form of warm starts is available. This reframes the problem significantly, distinguishing it from general-purpose samplers that operate with no such prior information.
>
> Our current related work section (1.1) already discusses several key annealing methods that are central in this field, such as Simulated Tempering (ST), Parallel Tempering (PT), and Sequential Monte Carlo (SMC).
>
> We will revise this section to make a clearer distinction and better situate our work. We are also happy to add further relevant references, and would be grateful for any suggested references that you believe would better situate our work.
>
> We have updated the paper so that it also includes the appendix. Algorithms 2 and 3 were pushed to the appendix in the interest of space. However, we included a more detailed description of these algorithms in our updated paper.
>
> Questions:
>
> Q: "Some minor comments: on line 161, the $q _ \beta$'s are actually not Gaussian densities but Gaussian-like, as later in proposition 3.2 $q _ \beta=\mathrm{e}^{-\beta||x||^2/2}$ is considered without normalization, and clearly $q _ 0$ is not the delta distribution. On line 237, $g _ {jj'}(x)$ seems to be $x-x _ j+x _ {j'}$ judging from the definition D.1 and the later usage in proposition 3.2."
>
> A: Yes, you are correct to point out that line 161 refers to q_\beta as a density function which implies normalization. We will correct the text so that it refers to q_\beta as tilting functions rather than density functions. q_{jj’} was meant as an example of a mapping that works, and this translation is what we use as for our specific gaussian example (proposition 3.2). However, our theory work is proven for a general pushforward q_{jj’}.

---

> > ### Comment · Reviewer_NdJv · 2025-11-24
> >
> > Thank you for your reply. I appreciate the inclusion of "Table 1: Comparison of Three ALPS Algorithm Versions" which help me better understand the related works on multimodal sampling. I remain positive to this paper, and believe it should be accepted at the conference.

---

### Official Review · Reviewer_hsfH · 2025-11-01

**Soundness:** 3
**Presentation:** 3
**Contribution:** 3
**Rating:** 6
**Confidence:** 2

**Summary:**

The paper studies sampling from multimodal target distributions when the practitioner is given “weak advice” in the form of one warm-start point per mode. It proposes a modified version of the Annealed Leap-Point Sampler (ALPS) that (i) tilts the target toward a mixture centered at the warm starts, (ii) anneals toward colder distributions, and (iii) performs teleportation between warm-start neighborhoods at the coldest level. Unlike ALPS, the method rebalances both *modal* and *level* weights by estimating component partition functions via Monte Carlo; this is meant to prevent bottlenecks across both modes and temperatures without requiring Hessians or near-Gaussian geometry. The analysis gives what the authors claim to be the first polynomial-time total-variation guarantee for this warm-start setting under a “tilt mass” assumption and local functional inequalities. The proof introduces a “good component” view of the chain, bounds mixing using local Poincare constants and a decomposition theorem, and shows that learned weights keep the projected chain well-conditioned. Experiments on a bimodal Student‑t mixture show higher acceptance rates and ESS than ALPS as dimension grows. The paper is explicit about limitations (e.g., requiring warm starts and some inefficiencies) and suggests a hybrid, more practical variant for future work.

**Strengths:**

* The paper clearly positions itself against prior impossibility results without advice and prior asymptotic ALPS analyses, and then supplies a total‑variation guarantee that depends only on local mixing and a warm‑start mass condition, filling a notable theoretical gap.
* The explicit *component* and *level* balance criteria (eqs. 2.3 and 2.4) plus Monte Carlo estimation of partition functions are simple knobs that directly control flow in the projected chain. This is an elegant explanation of why tempering often fails and how to fix it.
* The proof technique that analyzes convergence on the “good” mass and quantifies leakage from bad cross‑terms provides an idea that may extend to other multimodal samplers that create cross‑mode interactions.
* Weaker geometric requirements than ALPS: The method does not need Hessians and is not restricted to near‑Gaussian wells; the heavy‑tailed example aligns with this goal.
* Clear articulation of limitations and future directions: The conclusion frankly states that the present algorithm is not yet a practical replacement for ALPS and suggests online updates and hybridization.

**Weaknesses:**

* The warm‑start tilt assumption may be strong in practice. The mass condition c_tilt requires that, at small tilt, each component captures a constant fraction of the mass around its warm start. For modes with very different shapes or scales (e.g., narrow vs. flat), c_tilt could be tiny, inflating the polynomial constants and the number of levels. The paper notes separation may be needed in some cases.
* Proposition 3.2 suggests O(d^2) levels with Gaussian tilts, which could be burdensome. Each level requires Monte Carlo weight estimation; although polynomial, it may be costly for many modes. The paper’s experiments include just one learning step and a small number of modes.
* The chain within levels is a Metropolis random walk, but the paper does not examine sensitivity to proposal scale, jump and teleportation rates, or ladder spacing beyond one schedule.
* Section 6 acknowledges the current analysis assumes exactly one warm start per true mode and that every mode has non‑negligible mass at epsilon=0, which may be unrealistic when some modes are tiny but still relevant.

**Questions:**

* Can the authors provide concrete bounds or examples of c_tilt for standard families (mixtures of Gaussians with varying covariances, skewed or heavy‑tailed components) so readers can anticipate how the polynomial depends on geometry? Are there diagnostics a practitioner can compute from data to estimate c_tilt or detect when it is too small?
* Section 6 hints at adaptive tilts. Could the theory extend to tilts with learned covariances (e.g., local empirical Hessians or covariance estimates around warm starts) to stabilize c_tilt across anisotropic modes?
* Theorem 3.3 abstracts the Monte Carlo step. How do finite‑sample errors in modal and level weights affect the spectral gap of the projected chain and the final TV bound? Is there a practical rule for choosing the number of samples per level and when to stop re‑estimation?
* The assumptions require adjacent levels to be close in KL. Can the authors translate this into a practical ladder‑spacing heuristic or an adaptive controller that targets empirical swap rates, similar to parallel tempering?

---

> ### Author Response · Authors · 2025-11-20
> **Response (Weaknesses)**
>
> Q: "The warm‑start tilt assumption may be strong in practice. The mass condition c_tilt requires that, at small tilt, each component captures a constant fraction of the mass around its warm start. For modes with very different shapes or scales (e.g., narrow vs. flat), c_tilt could be tiny, inflating the polynomial constants and the number of levels. The paper notes separation may be needed in some cases."
>
> A: The tilt assumption is fundamentally linked to the minimum weight of the mixture components. Sampling from low weight modes is in itself an unavoidable difficulty for the projected chain. Hence, a dependence on the minimum modal weight is a natural assumption in the sampling literature. This is seen for the $\beta = 0$ case, where the tilt term needs to compensate for the lowest weight mode. In our case we are tempering the target distribution to colder (more peaked) gaussians which creates natural separation as $\beta$ increases. The technical conditions in our isotropic Gaussian tilt example are to ensure that increasing $\beta$ isn’t actually widening modes (in the case where a mixture component already has very low variance).
>
> Our isotropic Gaussian tilting example becomes sensitive to the choice of temperature ladder in the case of two modes that are close together (one with low variance, the other with high variance). As you correctly pointed out in this case c_tilt could be tiny. However, our theory work holds with minimal assumptions on the tilt functions q_\beta(x). This means that the tilt function could be chosen as a mixture of gaussians with varying Hessians (or scales) to better align with the mixture components of the target measure. We believe developing an adaptive version of our algorithm that takes into consideration modal geometries is a promising direction for future work, as we noted in the conclusion.
>
> Q: "Proposition 3.2 suggests O(d^2) levels with Gaussian tilts, which could be burdensome. Each level requires Monte Carlo weight estimation; although polynomial, it may be costly for many modes. The paper’s experiments include just one learning step and a small number of modes."
>
> A: See common response.
>
> In summary, we trade runtime which is potentially exponentially bad for a runtime which is provably polynomial in the dimension of the problem. As noted in our conclusions section, we believe that there are many directions for more practical implementations of our algorithm that could significantly reduce run time (for example, Robbins-Monro online weight estimation), which we view as a promising direction for future work
>
> Q: "The chain within levels is a Metropolis random walk, but the paper does not examine sensitivity to proposal scale, jump and teleportation rates, or ladder spacing beyond one schedule."
>
> A: The primary purpose of our experimental section was to highlight on a simple example the shortcomings of existing algorithms that our algorithm overcomes. In our updated version we chose a slightly better experimental setting, one which we felt better emphasized the benefits of our algorithm. You are correct to point out we didn’t remark on ladder spacing. In fact, because of our second Monte Carlo re-estimation step (re-weighting by empirical occupancy), our algorithm is highly robust to choice of ladder spacing.
>
> Analysis of proposal scale and choice of Markov chain on levels is a well studied problem. The problem we are concerned with is how well the chain mixes on the projected space, i.e. the probability of accepting jumps and teleportations. The choice of jump and teleportation rates ($\lambda$, $\gamma$) affects the bound on the Poincaré constant of the projected chain by a factor of $1/\lambda, 1/\gamma$ (Theorem E.1). This has a very small effect on the mixing of the projected chain and in practice allowing for 5 MRW steps before each jump/teleportation attempt worked well.
>
> Q: "Section 6 acknowledges the current analysis assumes exactly one warm start per true mode and that every mode has non‑negligible mass at epsilon=0, which may be unrealistic when some modes are tiny but still relevant."
>
> A: We agree that the assumption of a clean, one-to-one correspondence between warm starts and modes is an idealization. Our theoretical focus was on isolating the sampling problem, assuming the "weak advice" is well-posed. As we note, a more practical algorithm would need to handle spurious or redundant warm-start points, and we leave this as an important direction for future work.
>
> As for the tiny but relevant modes, we contrast our setting with that of rare event sampling (with algorithms such as umbrella
> sampling). In our work any mode with weight sufficiently small (less than $\epsilon$) does not need to be sampled from to achieve the desired $\epsilon$ accuracy in TV-distance. Our algorithm’s complexity which has a dependency of $1/c_{tilt}$ ($1/\min w_k$) is a natural analog to requiring $poly(1/\epsilon)$ complexity to find modes of size $\epsilon$.

---

> ### Author Response · Authors · 2025-11-20
> **Response (Questions)**
>
> Q1) Our main theory is general and allows for  more flexible tilt functions q_\beta(x). As we note in our conclusion, once could significantly improve c_tilt by choosing adaptive tilts that better match the local geometry of the component functions. We believe this is a key direction for future work; however our focus in this paper was sampling without local Hessian information.
> To see how additional information can improve the c_tilt constant we direct the reviewer to Proposition I.8 in the appendix. This proposition details how c_tilt can be computed when the tilting functions are a  mixture of spherical gaussians. For c_{tilt} to depend only on $\min w_k$ and not the geometry we had several technical conditions on the separation between the modes. However, it can be seen in the proof work that by allowing for the Gaussian components in the tilting mixture to have varying scales (introduce $\sigma_i$ and scale $\beta_i*\sigma_i$) we can guarantee $\Delta_i = \Delta_j$—thereby nullifying the need for additional constraints. This of course requires additional information to have the scalings of the tilting functions align with the scaling of the mixture components.
>
> Practical diagnostics for c_{tilt} is a great question! A practitioner cannot compute c_tilt directly, as it depends on the unknown components $\pi_k$. However, a practical diagnostic can be derived from the quantities our algorithm already estimates.
> We know c_tilt will be favorable at cold temperatures when mixture components are approximately peaked gaussians. Additionally, we know at the target level c_tilt depends on the minimum modal weight—a natural dependency. This means that the danger zone for c_tilt is at middling temperatures, when the scaling of warmer \betas could unintentionally skew the tempered distribution. Our inductive algorithm already estimates the partition functions $\bar{Z}_{l,k}$ and $Z_l$. Therefore, a practitioner can monitor the minimum of \bar{Z}_{l,k}/Z_l as an approximation of c_tilt. If this term becomes too small it would signal a potential bottleneck and a small effective c_tilt.
>
> Q2)  See common response.
>
> Q3) Our analysis is explicitly designed to be robust to finite-sample estimation errors. The key insight is that our algorithm does not require exact partition functions. It only requires that the ratios of the estimated weights are correct within a polynomial factor.
> In our theory work, the component and level balance (Definition 2.3) is maintained within polynomial factors $C_1$ and $C_2$ for finite-sample estimates of the partition functions. The Poincaré constant of the projected chain $\bar{C}$ is then shown to have a polynomial dependence (Lemma G.10) on C_1 and C_2. Our experiments also support this robustness, as a single learning step with N=1500 samples was sufficient to prevent bottlenecks and effectively sample from a heavy-tailed distribution where vanilla ALPS failed.
>
> Our paper focused on providing polynomial-time guarantees and the number of samples required $N = \Omega(poly(...))$ is not necessarily tight. With this in mind our algorithm will likely work well with much fewer samples than our guarantees imply necessary. As for stopping re-estimation, this is not stopped dynamically. Re-estimation is the fixed feature of our algorithm that guarantees component and level balance. For more practical implementation, as we note in our conclusion, a more advanced, practical algorithm could perform this estimation online (Robbins-Monroe for example) rather as a fixed, per level, iterative process. We believe this is a promising direction for future work.
>
> Q4) Yes, we do require levels to be close in $\chi^2$-divergence (not KL). This naturally arises from applying Cauchy-Schwarz (bounding estimation error by $Var_\pi(f)\cdot \chi^2(\pi, \nu)$). Our theory work suggests a very conservative rule of an O(1/d) ladder spacing (Lemma I.2). However, as we previously noted, the reweighting step (reweighting by empirical occupancy) provides additional robustness to ladder spacing—by effectively guaranteeing level balance. In practice, $O(\sqrt{d})$ levels with $\beta_1 = \Omega(d)$ and geometric spacing worked well (as expected from heuristics).
>
> Our algorithm’s swap mechanism is different from standard parallel tempering, which leads to a different kind of controller. In standard simulated tempering/parallel tempering, one tunes the temperature spacing to achieve the desired swap rate. This swap rate is bad only because the density ratios p_{i’}(x)/p_i(x) are bad. Our algorithm reweights by empirical occupancy which in a sense is already adaptive and enforcing level balance. This removes the primary bottleneck that standard adaptive controllers fight against.

---

> > ### Comment · Reviewer_hsfH · 2025-11-26
> >
> > Thank you for the response. I will maintain my score.

---

### Author Response · Authors · 2025-11-20
**Common Response**

We are encouraged that the reviewers recognized the novelty of our work and the importance of the problem we address: sampling from multimodal distributions with warm starts. Based on the reviews, we have revised the paper to improve readability by making the following changes:

1) We merged the appendix with the main paper.
2) We reframed how we fit into the existing ALPS literature (Section 1), provided a comparison table of all ALPS algorithms (Section C) and updated our experimental results (Section 5) to better emphasize how we fit within the existing ALPS literature.
3) We provided a more detailed description of our algorithms (Section 2).
4) We provided a more detailed, better laid out proof overview for our main results (Section 4)

Our primary contribution is to provide the first provable polynomial-time algorithm for this “weak advice” setting which we formalize in the paper. As reviewers noted, our key algorithmic innovation is a modified sequence of distributions for tempering and adaptive weight learning via Monte Carlo estimation.

We wish to clarify that the decomposition assumption $\pi(x) = \sum_k \alpha_k \pi_k(x)$  is a standard theoretical framework used to analyze metastability, not a requirement for the algorithm's implementation. Our algorithm operates in a black-box setting: it requires only query access to $\pi(x)$ and the warm starts. It does not need to know the mixture weights $\alpha_k$  or the functional forms of the components $\pi_k$​. This is a standard assumption in the simulated tempering literature, where the mixture is only used to describe the geometric barrier and the algorithm remains agnostic to the decomposition.

We address two central themes raised by multiple reviewers regarding the gap between our theoretical analysis and practical implementation:

Theoretical Focus vs. Computational Optimization: Several reviewers noted that our algorithm appears computationally expensive. Specifically, it was noted that our algorithm requires three separate runs (sets of samples) and an $\Omega(d^2)$ ladder. We emphasize that our primary contribution is establishing polynomial time guarantees for this problem not providing optimal rates.

We argue that while three separate runs may sound computationally expensive it overcomes the hidden computational cost of suffering from exponential mixing time due to bottlenecks in the projected chain (see example B.1 appendix). In summary, you trade a potential exponentially bad mixing time for a runtime that is provably polynomial.

Additionally, we note that the main algorithm, as written, is constructed for the ease of analysis. In practice, to acquire three separate sets of samples, a practitioner would run the algorithm for a burn-in time $T$ and then sample from the target leveling—appropriately thinning samples to reduce autocorrelation. This is standard theory vs implementation within the MCMC literature. It should also be mentioned that in our experimental work we only used two sets of samples, one for estimating mode/level weights and the other for reweighting via empirical occupancy.

Lastly, we want to emphasize our conclusion, that practical implementations can be significantly optimized (online weight learning and parallelization), and that our goal was to provide practitioners with a theoretical confidence in our algorithm. We view further practical optimizations and “real world applications” as a promising direction for future work.

The c_tilt Assumption and Adaptive Tilts: Reviewers asked about the robustness of the c_tilt assumption for complex geometries. While our analysis uses isotropic gaussian-like tilts for verifying our assumptions, our theory work is general and allows for a wide range of tilting functions q_\beta. Most notably, our framework supports adaptive tilts—tilting functions that take in local Hessian information from the target mixture components. We emphasize that our focus in this paper was verifying our results with minimal assumptions but we view adaptive tilts as a promising direction for future work.

We believe this paper is a strong fit for ICLR as it provides the first rigorous theoretical guarantees for a new algorithmic direction, opening the door for future work on more practical, online, and adaptive implementations.

---

### Author Response · Authors · 2025-12-03
**Summary of Rebuttals and Resolutions**

To the new Area Chair: In light of the recent changes, we provide a summary to assist in your evaluation of the discussion period.

(Reviewers 1 & 3) Theoretical vs. Practical Focus: Both reviewers noted the computational cost of our estimation steps. We resolved this by bringing to light the following:

1) We clarified that our primary contribution is the first provable polynomial-time bound for sampling with weak advice.
2) We emphasized that the trade is potential exponential mixing times (existing methods) for a provably polynomial runtime.
3) We emphasized that the practical implementations can be optimized and we view this as a promising direction for future work. However, the goal for this paper was to provide a theoretical foundation for sampling with weak advice.

(Reviewer 4): Reviewer 4 expressed concerns that our theoretical contribution might apply only to “trivial” problems. We managed to reach consensus on this by bringing to light the following points:
1) We clarified that our algorithm operates in a black box setting where the latent mixture weights and components are unknown.
2) We further emphasized that in the black box setting our experimental results in Section 5 become trivial and the reviewer accepted this point.
3) Lastly, we briefly remarked on the two part problem of acquiring warm start points and then sampling with such weak information.

(Reviewer 2): Reviewer 2 expressed concerns about the presentation and clarity of our results. We addressed this issue in the following way:
1) We provided more details about the algorithms (Section 2) and merged the main paper and supplementary material.
2) We refined our positioning within the existing ALPS literature to better emphasize our contribution to sampling with weak advice.

---

### Meta-Review · Area_Chair_2zkd · 2026-01-06

**Summary:**

This paper introduces Reweighted ALPS, a sampling algorithm for multimodal distributions that uses warm start points and Monte Carlo estimation to avoid the Hessian-based Gaussian approximations required by the original ALPS. The main contribution is the first polynomial-time guarantee for sampling with weak advice. Reviewers acknowledged the theoretical novelty but raised concerns about practical applicability. Key issues include computational cost (multiple runs, $\Omega(d^2)$ temperature ladder), the strength of the c_tilt assumption for modes with varying geometries, and whether the problem setting represents a meaningful advance beyond cases where component structure is effectively known.

**Reviewer Concerns:**

The authors adequately addressed presentation issues by merging the appendix and clarifying algorithm descriptions, satisfying Reviewer NdJv. Reviewer qZeH accepted the authors' argument that the algorithm trades potential exponential mixing for provable polynomial runtime. However, critical concerns remain unresolved. Reviewer hsfH's questions about c_tilt robustness across different geometries were acknowledged as future work rather than resolved, with the authors stating "our focus in this paper was sampling without local Hessian information." Most significantly, Reviewer i8Tb's persistent concern about non-triviality remains. Despite extensive discussion, the reviewer's final comment maintains "I still would not recommend accepting the work," noting that no concrete example simultaneously demonstrates unknown mixture structure, slow baseline mixing, and efficient advice-finding. The authors provided theoretical frameworks but deferred the combined problem (finding advice plus sampling efficiently) largely to future work, citing "Escaping from Saddle Points" for warm start acquisition without fully addressing whether this yields polynomial-time solutions for meaningful cases.

**Reviewer Scores:**

Reviewer hsfH (score: 6) would likely maintain their score at 6, appreciating the theory but noting unresolved practical limitations.

Reviewer NdJv (score: 6) would maintain 6, having explicitly stated satisfaction with revisions despite admitting "I don't have enough knowledge about related works" and inability to "verify all the technical details."

Reviewer qZeH (score: 4) would likely increase to 6 based on improved understanding.

Reviewer i8Tb (score: 2) would remain at 2, maintaining their rejection recommendation.

However, the general review quality raises serious concerns. Reviewer hsfH demonstrates clear technical competence, yet they note confidence level 2 and "Math/other details were not carefully checked." Reviewer NdJv explicitly admits insufficient expertise, Reviewer qZeH provided surface-level feedback, and Reviewer i8Tb rates confidence as 1 ("unable to assess"). No reviewer provides evidence of verifying the extensive proof machinery spanning Sections D-J, despite the paper's strong theoretical claims requiring careful verification by MCMC experts.

Given the inadequate technical review, the disrupted process, and the authors' own acknowledgment that practical applicability requires substantial future work, I recommend rejection with encouragement to resubmit to a specialized theory venue (COLT, ALT, or similar) where the paper will receive proper expert evaluation. The work may represent a genuine theoretical contribution, but the current review process cannot provide sufficient confidence in its correctness or significance.

---

### Decision · Program_Chairs · 2026-01-26

Reject